# KAN: KOLMOGOROV–ARNOLD NETWORKS

**Ziming Liu**[1,4*]   **Yixuan Wang**[2]   **Sachin Vaidya**[1]   **Fabian Ruehle**[3,4]
**James Halverson**[3,4]   **Marin Soljačić**[1,4]   **Thomas Y. Hou**[2]   **Max Tegmark**[1,4]

[1] Massachusetts Institute of Technology
[2] California Institute of Technology
[3] Northeastern University
[4] The NSF Institute for Artificial Intelligence and Fundamental Interactions

## ABSTRACT

Although machine learning is a powerful tool for science, its black-box nature hinders the extraction of interpretable knowledge. In particular, although Multi-Layer Perceptrons (MLPs) are universal approximators, it is challenging to interpret what MLPs are doing under the hood. This paper, inspired by the Kolmogorov-Arnold representation theorem, proposes Kolmogorov-Arnold Networks (KANs) as promising alternatives to MLPs, especially when interpretability is desired. While MLPs have *fixed* activation functions on *nodes* ("neurons"), KANs have *learnable* activation functions on *edges* ("weights"). KANs learn interpretable 1D functions on their edges whose connection graph is also simple enough to be explained. Through two examples in mathematics and physics, KANs are shown to be useful "collaborators" helping scientists (re)discover mathematical and physical laws. Moreover, KANs are shown to be more accurate and have faster scaling laws than MLPs in function fitting and PDE solving, both theoretically and empirically. However, we admit that training KANs could be slower than MLPs, which should be addressed in the future to scale them up.

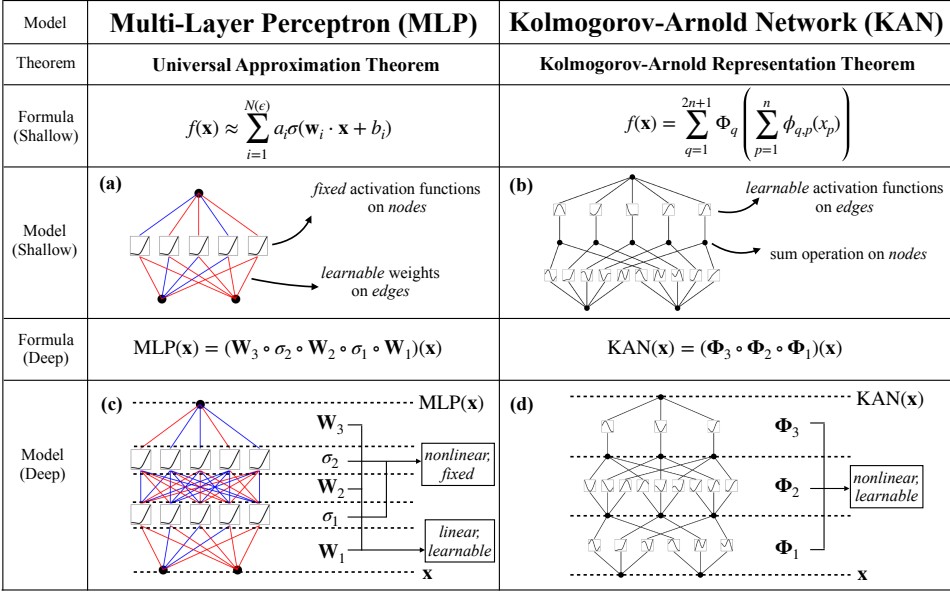

| Model | **Multi-Layer Perceptron (MLP)** | **Kolmogorov-Arnold Network (KAN)** |
|---|---|---|
| Theorem | **Universal Approximation Theorem** | **Kolmogorov-Arnold Representation Theorem** |
| Formula (Shallow) | $f(\mathbf{x}) \approx \sum_{i=1}^{N(\epsilon)} a_i \sigma(\mathbf{w}_i \cdot \mathbf{x} + b_i)$ | $f(\mathbf{x}) = \sum_{q=1}^{2n+1} \Phi_q\left(\sum_{p=1}^{n} \phi_{q,p}(x_p)\right)$ |
| Model (Shallow) | **(a)** *fixed* activation functions on *nodes* / *learnable* weights on *edges* | **(b)** *learnable* activation functions on *edges* / sum operation on *nodes* |
| Formula (Deep) | $\mathrm{MLP}(\mathbf{x}) = (\mathbf{W}_3 \circ \sigma_2 \circ \mathbf{W}_2 \circ \sigma_1 \circ \mathbf{W}_1)(\mathbf{x})$ | $\mathrm{KAN}(\mathbf{x}) = (\mathbf{\Phi}_3 \circ \mathbf{\Phi}_2 \circ \mathbf{\Phi}_1)(\mathbf{x})$ |
| Model (Deep) | **(c)** MLP(x) $\mathbf{W}_3$ / $\sigma_2$ *nonlinear, fixed* / $\mathbf{W}_2$ / $\sigma_1$ / $\mathbf{W}_1$ *linear, learnable* / x | **(d)** KAN(x) $\mathbf{\Phi}_3$ / $\mathbf{\Phi}_2$ *nonlinear, learnable* / $\mathbf{\Phi}_1$ / x |

Figure 1: Multi-Layer Perceptrons (MLPs) vs. Kolmogorov-Arnold Networks (KANs)

## 1   INTRODUCTION

Multi-layer perceptrons (MLPs) Haykin (1994); Cybenko (1989); Hornik et al. (1989), also known as fully-connected feedforward neural networks, are foundational building blocks of today's deep

---
*zmliu@mit.edu

learning models. The importance of MLPs can never be overstated, since they are the default models in machine learning for approximating nonlinear functions, due to their expressive power guaranteed by the universal approximation theorem Hornik et al. (1989). However, MLPs often lack interpretability, which makes them less useful for tasks when interpretability is key, e.g., when we want to extract symbolic formulas from datasets. In science, symbolic functions are prevalent, e.g., $E = mc^2$ (energy-mass relation), $r = \frac{a}{1+e\cos\theta}$ (ellipse), $p = e^{-\frac{E}{kT}}/Z$ (Boltzman distribution). Although MLPs can numerically approximate these functions to a reasonable accuracy, they cannot reveal symbolic structures of these equations.

Therefore, we need a representation theorem that is more aligned with symbolic representations than the universal approximation theorem. In our search, the good old Kolmogorov-Arnold representation theorem (KA theorem) came to our attention. Although the KA theorem has long been considered irrelevant for learning Girosi & Poggio (1989) because the theorem does not guarantee smoothness, we are more optimistic about the smoothness of deeper representations. For example, as we will show, $f(x_1, x_2, x_3, x_4) = \exp(\sin(x_1^2 + x_2^2) + \sin(x_3^2 + x_4^2))$ can be smoothly represented by a three-layer network, but a two-layer network that attempts to fit this function leads to pathological representations.

Unsurprisingly, the possibility of using Kolmogorov-Arnold representation theorem to build neural networks has been studied Sprecher & Draghici (2002); Köppen (2002); Lin & Unbehauen (1993); Lai & Shen (2021); Leni et al. (2013); Fakhoury et al. (2022); Montanelli & Yang (2020). However, most work has stuck with the original depth-2 width-$(2n+1)$ representation, and many did not have the chance to leverage more modern techniques (e.g., back propagation) to train the networks. Our contribution lies in generalizing the original Kolmogorov-Arnold representation to arbitrary widths and depths, revitalizing and contextualizing it in today's deep learning world, as well as using empirical experiments to highlight its potential for AI + Science due to its accuracy and interoperability.

Named after the two great Mathematicians, Andrey Kolmogorov and Vladimir Arnold, this new type of network is called the *Kolmogorov-Arnold Network* (KAN). Like MLPs, KANs have fully-connected structures. However, while MLPs place fixed activation functions on *nodes* ("neurons"), KANs place learnable activation functions on *edges* ("weights"), as illustrated in Figure 1. Each learnable weight parameter in an MLP is replaced by a learnable 1D function (parametrized as a spline) in a KAN. KANs' nodes simply sum incoming signals without applying any non-linearities.

Although interpretability is our initial motivation to develop KANs, KANs demonstrate impressive accuracy and fast scaling laws as well, both theoretically and empirically. Despite their elegant mathematical interpretation, KANs are nothing more than combinations of splines and MLPs, leveraging their respective strengths and avoiding their respective weaknesses. Splines are accurate for low-dimensional functions but suffer from curse of dimensionality (COD) problem. MLPs, On the other hand, suffer less from COD thanks to their ability to learn features and compositional structure, but are less accurate than splines in low dimensions. KANs have MLPs on the outside and splines on the inside, combining the best of two things into one.

The paper is organized as follows: In Section 2, we introduce the KAN architecture, analyze the network's approximation ability, and propose two training techniques to make KANs interpretable and accurate. In Section 3, we show that KANs are interpretable and can be used for scientific discoveries. We use two examples from mathematics (knot theory) and physics (Anderson localization) to demonstrate that KANs can be helpful "collaborators" for scientists to (re)discover math and physical laws. In Section 4, we show that KANs are more accurate than MLPs for data fitting and PDE solving with better scaling laws. We conclude in Section 5. Due to limited space, we defer related works to Appendix Y and discussion to Appendix Z.

## 2 KOLMOGOROV−ARNOLD NETWORKS (KAN)

Multi-Layer Perceptrons (MLPs) are inspired by the universal approximation theorem. We instead focus on the Kolmogorov-Arnold representation theorem, which can be realized by a new type of neural network called Kolmogorov-Arnold networks (KAN). We review the Kolmogorov-Arnold theorem in Section 2.1, to inspire the design of Kolmogorov-Arnold Networks in Section 2.2. Section 2.3 provides mathematical description of KANs' expressive power. Section 2.5 and Section 2.4 propose techniques to make KANs accurate and interpretable.

## 2.1 KOLMOGOROV-ARNOLD REPRESENTATION THEOREM

Vladimir Arnold and Andrey Kolmogorov established that if $f$ is a multivariate continuous function on a bounded domain, then $f$ can be written as a finite composition of continuous functions of a single variable and the binary operation of addition. More specifically, for a smooth $f : [0,1]^n \to \mathbb{R}$,

$$f(\mathbf{x}) = f(x_1, \cdots, x_n) = \sum_{q=1}^{2n+1} \Phi_q \left( \sum_{p=1}^{n} \phi_{q,p}(x_p) \right), \tag{1}$$

where $\phi_{q,p} : [0,1] \to \mathbb{R}$ and $\Phi_q : \mathbb{R} \to \mathbb{R}$. In a sense, they showed that the only true multivariate function is addition, since every other function can be written using univariate functions and sum. One might naively consider this great news for machine learning: learning a high-dimensional function boils down to learning a polynomial number of 1D functions. However, these 1D functions can be non-smooth and even fractal, so they may not be learnable in practice Poggio et al. (2020). Because of this pathological behavior, the Kolmogorov-Arnold representation theorem was regarded as theoretically sound but practically useless Poggio et al. (2020).

However, we are more optimistic about the usefulness of the Kolmogorov-Arnold theorem for machine learning. First of all, we need not stick to the original Eq. (1) which has only two-layer non-linearities and a small number of terms $(2n + 1)$ in the hidden layer: we will generalize the network to arbitrary widths and depths. Deeper and wider networks potentially have stronger expressive power with smooth functions. Moreover, most functions in science and daily life are often smooth and have sparse compositional structures Lin et al. (2017), potentially facilitating smooth Kolmogorov-Arnold representations.

## 2.2 KAN ARCHITECTURE

Suppose we have a supervised learning task consisting of input-output pairs $\{\mathbf{x}_i, y_i\}$, where we want to find $f$ such that $y_i \approx f(\mathbf{x}_i)$ for all data points. Eq. (1) implies that we are done if we can find appropriate univariate functions $\phi_{q,p}$ and $\Phi_q$. This inspires us to design a neural network which explicitly parametrizes Eq. (1). Since all functions to be learned are univariate functions, we can parametrize each 1D function as a B-spline curve, with learnable coefficients of local B-spline basis functions [1]. Now we have a prototype of KAN, whose computation graph is exactly specified by Eq. (1) and illustrated in Figure 1 (b) (with the input dimension $n = 2$), appearing as a two-layer neural network with activation functions placed on edges instead of nodes (simple summation is performed on nodes), and with width $2n + 1$ in the middle layer.

As mentioned, such a network is known to be too simple to approximate any function arbitrarily well in practice with smooth splines! We therefore generalize our KAN to be wider and deeper. The key insight comes from the analogy between MLPs and KANs. In MLPs, once we define a layer (which is composed of a linear transformation and nonlinearties), we can stack more layers to make the network deeper. To build deep KANs, we should first answer: "what is a KAN layer?" It turns out that a KAN layer with $n_{\text{in}}$-dimensional inputs and $n_{\text{out}}$-dimensional outputs can be defined as a matrix of 1D functions

$$\mathbf{\Phi} = \{\phi_{q,p}\}, \qquad p = 1, 2, \cdots, n_{\text{in}}, \qquad q = 1, 2 \cdots, n_{\text{out}}, \tag{2}$$

where the functions $\phi_{q,p}$ have trainable parameters (parameterized as B-splines, see Appendix I), as detaild below. In the Kolmogov-Arnold theorem, the inner functions form a KAN layer with $n_{\text{in}} = n$ and $n_{\text{out}} = 2n + 1$, and the outer functions form a KAN layer with $n_{\text{in}} = 2n + 1$ and $n_{\text{out}} = 1$. So the Kolmogorov-Arnold representations in Eq. (1) are simply compositions of two KAN layers. Now it becomes clear what it means to have deeper Kolmogorov-Arnold representations: simply stack more KAN layers! The shape of a general KAN is represented by an integer array

$$[n_0, n_1, \cdots, n_L], \tag{3}$$

where $n_i$ is the number of nodes in the $i^{\text{th}}$ layer of the computational graph. We denote the $i^{\text{th}}$ neuron in the $l^{\text{th}}$ layer by $(l, i)$, and the activation value of the $(l, i)$-neuron by $x_{l,i}$. Between layer $l$ and layer $l + 1$, there are $n_l n_{l+1}$ activation functions: the activation function that connects $(l, i)$ and $(l + 1, j)$ is denoted by

$$\phi_{l,j,i}, \quad l = 0, \cdots, L - 1, \quad i = 1, \cdots, n_l, \quad j = 1, \cdots, n_{l+1}. \tag{4}$$

---

[1]Details in Appendix I and illustrated in Figure 18 right.

The pre-activation of $\phi_{l,j,i}$ is simply $x_{l,i}$; the post-activation of $\phi_{l,j,i}$ is denoted by $\tilde{x}_{l,j,i} \equiv \phi_{l,j,i}(x_{l,i})$. The activation value of the $(l+1, j)$ neuron is simply the sum of all incoming post-activations:

$$x_{l+1,j} = \sum_{i=1}^{n_l} \tilde{x}_{l,j,i} = \sum_{i=1}^{n_l} \phi_{l,j,i}(x_{l,i}), \qquad j = 1, \cdots, n_{l+1}. \tag{5}$$

In matrix form, this reads

$$\mathbf{x}_{l+1} = \underbrace{\begin{pmatrix} \phi_{l,1,1}(\cdot) & \phi_{l,1,2}(\cdot) & \cdots & \phi_{l,1,n_l}(\cdot) \\ \phi_{l,2,1}(\cdot) & \phi_{l,2,2}(\cdot) & \cdots & \phi_{l,2,n_l}(\cdot) \\ \vdots & \vdots & & \vdots \\ \phi_{l,n_{l+1},1}(\cdot) & \phi_{l,n_{l+1},2}(\cdot) & \cdots & \phi_{l,n_{l+1},n_l}(\cdot) \end{pmatrix}}_{\boldsymbol{\Phi}_l} \mathbf{x}_l, \tag{6}$$

where $\boldsymbol{\Phi}_l$ is the function matrix corresponding to the $l^{\text{th}}$ KAN layer. A general KAN network is a composition of $L$ layers: given an input vector $\mathbf{x}_0 \in \mathbb{R}^{n_0}$, the output of KAN is

$$\text{KAN}(\mathbf{x}) = (\boldsymbol{\Phi}_{L-1} \circ \boldsymbol{\Phi}_{L-2} \circ \cdots \circ \boldsymbol{\Phi}_1 \circ \boldsymbol{\Phi}_0)\mathbf{x}. \tag{7}$$

We can also rewrite the above equation to make it more analogous to Eq. (1), assuming output dimension $n_L = 1$, and define $f(\mathbf{x}) \equiv \text{KAN}(\mathbf{x})$:

$$f(\mathbf{x}) = \sum_{i_{L-1}=1}^{n_{L-1}} \phi_{L-1,i_L,i_{L-1}} \left( \sum_{i_{L-2}=1}^{n_{L-2}} \cdots \left( \sum_{i_2=1}^{n_2} \phi_{2,i_3,i_2} \left( \sum_{i_1=1}^{n_1} \phi_{1,i_2,i_1} \left( \sum_{i_0=1}^{n_0} \phi_{0,i_1,i_0}(x_{i_0}) \right) \right) \right) \cdots \right), \tag{8}$$

which is quite cumbersome. In contrast, our abstraction of KAN layers and their visualizations are cleaner and intuitive. The original Kolmogorov-Arnold representation Eq. (1) corresponds to a 2-Layer KAN with shape $[n, 2n+1, 1]$. Notice that all the operations are differentiable, so we can train KANs with back propagation. For comparison, an MLP can be written as interleaving of affine transformations $\mathbf{W}$ and non-linearities $\sigma$:

$$\text{MLP}(\mathbf{x}) = (\mathbf{W}_{L-1} \circ \sigma \circ \mathbf{W}_{L-2} \circ \sigma \circ \cdots \circ \mathbf{W}_1 \circ \sigma \circ \mathbf{W}_0)\mathbf{x}. \tag{9}$$

It is clear that MLPs treat linear transformations and nonlinearities separately as $\mathbf{W}$ and $\sigma$, while KANs treat them all together in $\boldsymbol{\Phi}$. In Figure 1 (c) and (d), we visualize a three-layer MLP and a three-layer KAN, to clarify their differences. Implementation details of KANs are left in Appendix I.

**Remark: Complexities**. Assuming a KAN with depth $L$, width $N$, grid size $G$, spline order $k$. The model has $O(N^2 GL)$ parameters. Suppose a training batch has size $B$, memory usage is $O(2^k BN^2 GL)$, the number of operations is $O(2^k BN^2 GL)$ both for forward and backward runs. The $2^k$ factor is due to the recursive computation of order $k$ splines.

## 2.3 KAN's Approximation Abilities and Scaling Laws

Recall that in Eq. (1), the 2-Layer width-$(2n+1)$ representation may be non-smooth. However, deeper representations may bring the advantages of smoother activations. To facilitate an approximation analysis, we still assume smoothness of activations, but allow the representations to be arbitrarily wide and deep, as in Eq. (7). To emphasize the dependence of our KAN on the finite set of grid points, we use $\boldsymbol{\Phi}_l^G$ and $\Phi_{l,i,j}^G$ below to replace the notation $\boldsymbol{\Phi}_l$ and $\Phi_{l,i,j}$ used in Eq. (5) and (6).

**Theorem 2.1** (Approximation theory, KAN). *Let $\mathbf{x} = (x_1, x_2, \cdots, x_n)$. Suppose that a function $f(\mathbf{x})$ admits a representation*

$$f = (\boldsymbol{\Phi}_{L-1} \circ \boldsymbol{\Phi}_{L-2} \circ \cdots \circ \boldsymbol{\Phi}_1 \circ \boldsymbol{\Phi}_0)\mathbf{x}, \tag{10}$$

*as in Eq. (7), where each one of the $\Phi_{l,i,j}$ are $(k+1)$-times continuously differentiable. Then there exists a constant $C$ depending on $f$ and its representation, such that we have the following approximation bound in terms of the grid size $G$: there exist $k$-th order B-spline functions $\Phi_{l,i,j}^G$ such that for any $0 \le m \le k$, we have the bound*

$$\|f - (\boldsymbol{\Phi}_{L-1}^G \circ \boldsymbol{\Phi}_{L-2}^G \circ \cdots \circ \boldsymbol{\Phi}_1^G \circ \boldsymbol{\Phi}_0^G)\mathbf{x}\|_{C^m} \le CG^{-k-1+m}. \tag{11}$$

*Here we adopt the notation of $C^m$-norm measuring the magnitude of derivatives up to order $m$:*

$$\|g\|_{C^m} = \max_{|\beta| \le m} \sup_{x \in [0,1]^n} |D^\beta g(x)|.$$

We leave the proof and an in-depth discussion on the implication of the theorem in Appendix J. Asymptotically, provided that the assumption in Theorem 2.1 holds, KANs with finite grid size can approximate the function well with a residue rate independent of the dimension. This comes naturally since we only use splines to approximate 1D functions. In particular, for $m = 0$, we recover the accuracy in $L^\infty$ norm, which in turn provides a bound of RMSE on the finite domain, which gives a scaling exponent $k + 1$. Of course, the constant $C$ is dependent on the representation; hence it will depend on the dimension. Notice that if the assumption in the theorem holds for a shallow KAN, it automatically holds for a deeper KAN by setting the remaining layers to identity. A more general version of approximation theory in larger function class can be found in Wang et al. (2024b). More discussion on how our results are related to neural scaling laws is included in Appendix K. We also remark that: since the assumption in the theorem is a strong one, the neural scaling law should not be expected to be universally applicable to all machine learning applications. Now the basic architecture of KANs is in place, we propose a few techniques to make KANs accurate and interpretable.

### 2.4 Tricks For Interpretability: Pruning and Symbolifying KANs

How do we choose the KAN shape? If we know that the dataset is generated via the symbolic formula $f(x, y) = \exp(\sin(\pi x) + y^2)$, then we know that a $[2, 1, 1]$ KAN is able to express this function. However, in practice we do not know the shape a priori, so it would be nice to have approaches to determine this shape automatically. The idea is to start from a large enough KAN and train it with sparsity regularization followed by pruning. One may even symbolify activation functions into symbolic functions like exp, sine, etc, to make KANs a useful tool for symbolic regression. The idea is to match learned spline functions with candidates in a symbolic function library specified by human users and replace the spline functions with the best-fitting ones. Details of these simplification tricks are included in Appendix M.

### 2.5 A Trick For accuracy: Grid Update and Grid Extension

**Grid update** Since input data and (especially) hidden activations can have time-varying ranges in training, we update grids on the fly based on the statistics of input/activation ranges. The grid is initialized to be in [-1,1] (e.g., when $G = 5$, the grid points are [-1, -0.6, -0.2, 0.2, 0.6, 1.0]), but once it receives input/activations, say, in the range [-3,3] (the maximum and minimum values are 3 and -3, respectively), the grid will be updated to [-3,3] (correspondingly, grid points become [-3,-1.8,-0.6,0.6,1.8,3.0]) to accommodate the whole range.

**Grid extension** A spline can be made arbitrarily accurate to a target function as the grid can be made arbitrarily fine-grained. This good feature can be inherited by KANs. By contrast, MLPs do not have the notion of "fine-graining". For KANs, one can first train a KAN with fewer parameters and then extend it to a KAN with more parameters by simply making its spline grids finer, without the need to retrain the larger model from scratch. The main idea of grid extension is: for each 1D function defined on a coarse grid, we determine the coefficient of a finer grid using least squares that minimize the difference between the two curves evaluated on data samples. Details of how to perform grid extension are included in Appendix L and in Figure 18.

### 2.6 Benefits of deep KANs

It is one of our major contributions to generalize the 2-layer KA representations to multiple layers. Although it is challenging to prove the benefits of deeper KANs theoretically, we want to present a concrete example where 3-layer KANs admit smooth representations while 2-layer KANs do not. We consider fitting a function $f(x_1, x_2, x_3, x_4) = \exp(\frac{1}{2}(\sin(\pi(x_1^2 + x_2^2)) + \sin(\pi(x_3^2 + x_4^2))))$ where we draw samples (3000 training, 1000 training) uniformly from $[-1, 1]^4$. We train a 3L KAN ([4,2,1,1]) and a 2L KAN ([4,9,1]) with the LBFGS optimizer for 250 steps, with increasing $G = 3, 5, 10, 20, 50$ (50 steps for each $G$). As shown in Figure 2, we see that the 3-layer KAN has smooth representations (as expected, since the parse tree of the symbolic formula has depth 3), while the 2-layer KAN learns highly oscillatory functions on some edges. The 3-layer KAN also achieves lower losses than the 2-layer KAN. While the 3-layer KAN has a small train-test gap, the 2-layer KAN starts to overfit at large grid sizes.

## 3 KANs are interpretable

In this section, we show that KANs can be interpretable on synthetic toy tasks and realistic research questions in math and physics.

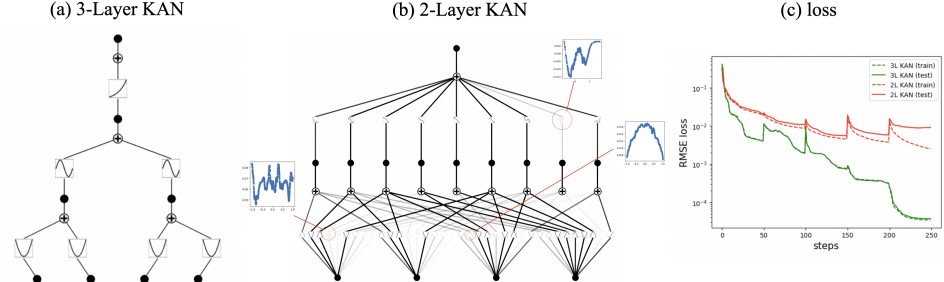

Figure 2: Fitting the function $f(x_1, x_2, x_3, x_4) = \exp(\frac{1}{2}(\sin(\pi(x_1^2 + x_2^2)) + \sin(\pi(x_3^2 + x_4^2))))$. (a) 3-Layer KAN admits smooth representations. (b) The 2-Layer KAN learns highly oscillatory representations. (c) The 3-layer KAN achieves lower losses and has a smaller train-test gap than the 2-layer KAN.

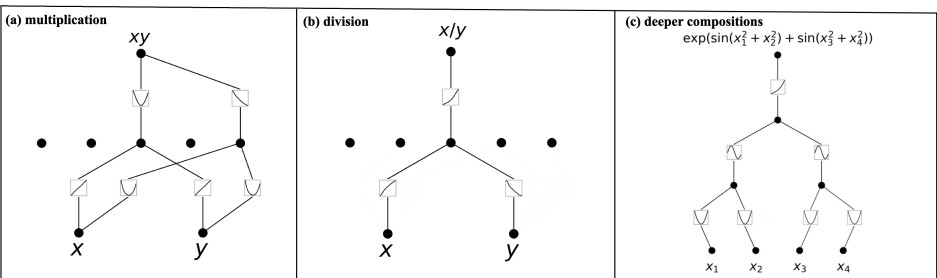

Figure 3: KANs are interpretable for simple symbolic tasks.

**Synthetic toy datasets** We first examine KANs' ability to reveal the compositional structures in symbolic formulas. Three examples are presented in Figure 3. KANs are able to reveal the compositional structures present in these formulas, as well as learn the correct univariate functions. (1) Multiplication $f(x, y) = xy$. KAN computes it via the equation $2xy = (x + y)^2 - (x^2 + y^2)$. (2) Division of positive numbers $f(x, y) = x/y$. KAN computes it via $\exp(\log x - \log y)$. (3) Deeper compositions $f(x_1, x_2, x_3, x_4) = \exp(\sin(x_1^2 + x_2^2) + \sin(x_3^2 + x_4^2))$. Discussion about the implications of these examples is left in Appendix R. We also discussed an unsupervised learning paradigm and how we can convert unsupervised learning to supervised learning by borrowing ideas from contrastive learning, detailed in Appendix S.

**Application to Mathematics: Knot Theory** Knot theory is a subject in low-dimensional topology that sheds light on topological aspects of three-manifolds and four-manifolds and has a variety of applications, including in biology and topological quantum computing. In Davies et al. (2021), supervised learning and human domain experts were utilized to arrive at a new theorem relating algebraic and geometric knot invariants. They use network attribution methods to find that the signature $\sigma$ is mostly dependent on meridinal distance $\mu$ (real $\mu_r$, imag $\mu_i$) and longitudinal distance $\lambda$. We show that KANs can not only identify these important variables with much smaller networks and much more automation, but also present some interesting new results and insights.

We treat 17 knot invariants as inputs and signature as outputs. Similar to the setup in Davies et al. (2021), signatures (which are even numbers) are encoded as one-hot vectors and networks are trained with cross-entropy loss. We find that an extremely small $[17, 1, 14]$ KAN is able to achieve $81.6\%$ test accuracy (while DeepMind's 4-layer width-300 MLP achieves $78\%$ test accuracy). The $[17, 1, 14]$ KAN ($G = 3$, $k = 3$) has $\approx 200$ parameters, while the MLP has $\approx 3 \times 10^5$ parameters. It is remarkable that KANs can be both more accurate and much more parameter efficient than MLPs at the same time. In terms of interpretability, we scale the transparency of each activation according to its magnitude, so it becomes immediately clear which input variables are important without the need for feature attribution (see Figure 4 left top): signature is mostly dependent on $\mu_r$, and slightly

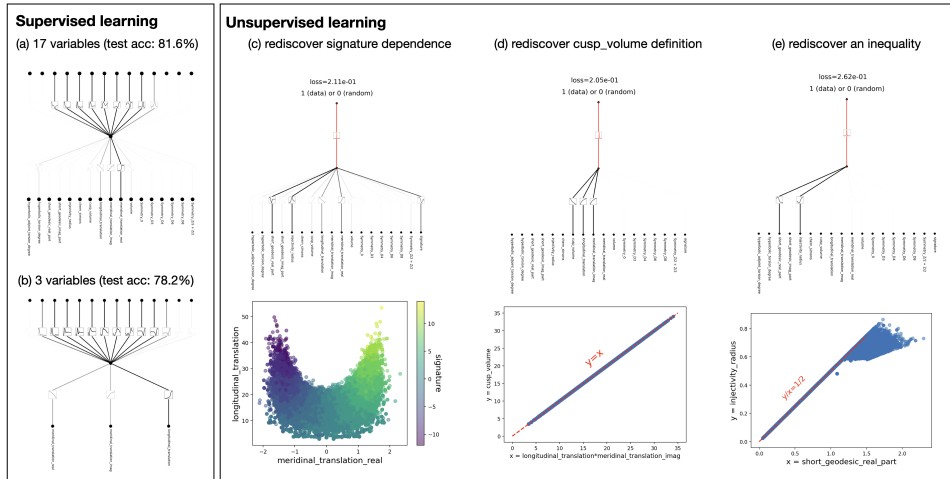

Figure 4: Knot dataset. Supervised mode (left): we rediscover DeepMind's three important variables. Unsupervised mode (right): we discover three "new" relations without supervision.

dependent on $\mu_i$ and $\lambda$, while dependence on other variables is small. We then train a $[3, 1, 14]$ KAN on the three important variables, obtaining test accuracy 78.2% (Figure 4 left bottom). More ablation results and symbolic formula results are included in Appendix T.

We attempt to make discoveries beyond DeepMind's in the unsupervised learning mode, where we treat all 18 variables (including signature) as inputs. We train 200 networks with different random seeds. They can be grouped into three clusters, with representative KANs displayed in Figure 4. These three groups of dependent variables are (1) rediscovering DeepMind's relation in unsupervised learning. (2) cusp volume is by definition of the multiplication of two translations. (3) short geodesic $g_r$ is upper bounded by two times of injecitivy radius Petersen (2006). It is interesting that KANs' unsupervised mode can rediscover several known mathematical relations. The good news is that the results discovered by KANs are probably reliable; the bad news is that we have not discovered anything new yet. It is worth noting that we have chosen a shallow KAN for simple visualization, but deeper KANs can probably find more relations if they exist. We would like to investigate how to discover more complicated relations with deeper KANs in future work.

**Remark: symbolic regression benchmarks** We have presented KANs' interpretability as an interactive tool with human users. However, as a network-based method, its strong capability (in fitting even non-symbolic functions) makes it unfavorable for standard symbolic regression benchmarks. For example, KAN ranks second-to-last in GEOBENCH (Anonymous, 2024), whereas the last-ranked one EQL is also a network-based model, which has been shown to be useful at least for certain problems (Martius & Lampert, 2016; Dugan et al., 2020) despite its inability to do well on benchmarks. On the one hand, we would like to explore ways to restrict KANs' hypothesis space so that KANs can achieve good performance on symbolic regression benchmarks. On the other hand, we want to point out that KANs have good features that are not reflected by existing benchmarks: (1) interactivity. It is relatively easier to visualize the training dynamics of KANs, which gives human users intuition on what could go wrong hence facilitating debugging. (2) The ability to "discover" new functions. If the ground truth formula contains a special function but is not given in the symbolic library, SR methods will fail. However, KANs can discover the need for a new function whose numerical behavior suggests maybe it is a Bessel function; see Figure 23 (d) for an example.

## 4 KANs ARE ACCURATE

In this section, we demonstrate that KANs are more accurate at representing functions than MLPs in various tasks (regression and PDE solving). When comparing two families of models, it is fair to compare both their accuracy (loss) and their complexity (number of parameters). Moreover, in Appendix Q, we show that KANs can naturally work in continual learning without catastrophic

forgetting. All experiments reported in the paper are reproducible on CPUs usually within minutes, at most in a day. Codes are built based on pytorch Paszke et al. (2019).

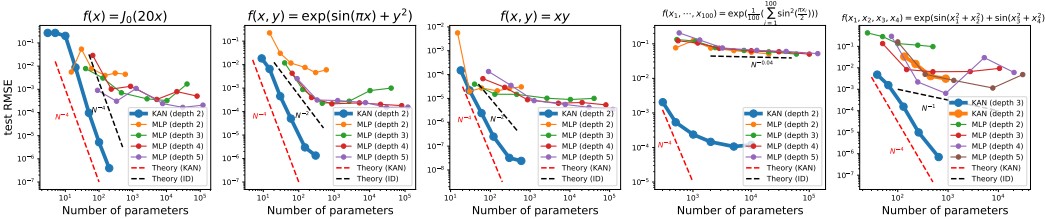

Figure 5: Compare KANs to MLPs on five toy examples. KANs can almost saturate the fastest scaling law predicted by our theory ($\alpha = 4$), while MLPs scales slowly and plateau quickly.

**Toy datasets** In Section 2.3, our theory suggested that test RMSE loss $\ell$ scales as $\ell \propto N^{-(k+1)} = N^{-4}(k = 3)$ with model parameters $N$. However, this relies on the existence of a smooth Kolmogorov-Arnold representation. As a sanity check, we construct five examples we know have smooth KA representations: (1) $f(x) = J_0(20x)$, which is the Bessel function. Since it is a univariate function, it can be represented by a spline, which is a $[1, 1]$ KAN. (2) $f(x, y) = \exp(\sin(\pi x) + y^2)$. We know that it can be exactly represented by a $[2, 1, 1]$ KAN. (3) $f(x, y) = xy$. We know from Figure 3 that it can be exactly represented by a $[2, 2, 1]$ KAN. (4) A high-dimensional example $f(x_1, \cdots, x_{100}) = \exp(\frac{1}{100} \sum_{i=1}^{100} \sin^2(\frac{\pi x_i}{2}))$ which can be represented by a $[100, 1, 1]$ KAN. (5) A four-dimensional example $f(x_1, x_2, x_3, x_4) = \exp(\frac{1}{2}(\sin(\pi(x_1^2 + x_2^2)) + \sin(\pi(x_3^2 + x_4^2))))$ which can be represented by a $[4, 4, 2, 1]$ KAN. The empirical scaling for KANs is quite aligned with theory and outperforms MLPs. Details of training are included in Appendix N.

**Special functions** In practice, we may not know the existence of KA representations. Special functions of more than one variables are such cases, e.g., a Bessel function $f(\nu, x) = J_\nu(x)$. We collect 15 special functions common in math and physics, summarized in Table 9 in Appendix O. We find that: (1) KANs are more efficient and accurate in representing special functions than MLPs, as shown in Figure 6. In all cases, KANs have better pareto frontiers than MLPs. (2) Finding (approximate) compact KA representations of special functions is possible, revealing novel mathematical properties of special functions from the perspective of Kolmogorov-Arnold representations. Details are included in Appendix O.

**Fitting Images** We task KANs with three images: (1) The Cameraman picture is the standard picture for the image fitting task. (2) The turbulence profile is taken from PDEBench Takamoto et al. (2022), demonstrating high-frequency and fractal behavior typical in scientific computing. (3) Van Gogh's *The Starry Night* is quite challenging because it contains fine-grained details as well. In addition to MLPs, We compare KANs with these stronger baselines: (A) MLP with random Fourier features (MLP_RFF). Before feeding input coordinates $\mathbf{x} \equiv (x, y)$ to the MLP, we first augment them into a higher-dimensional feature space $\Phi(\mathbf{x}) = (\mathbf{x}, \Phi_1(\mathbf{x}), \cdots, \Phi_{N_f}(\mathbf{x}))$, where $\Phi_i(\mathbf{x}) = (\cos(\mathbf{s}_i \cdot \mathbf{x}), \sin(\mathbf{s}_i \cdot \mathbf{x})), i = 1, \cdots, N_f$, and $\mathbf{s}_i \sim \mathcal{N}(0, s^2)$ ($s$ controls the frequency bias). We choose $N_f = 50$ and $s = 3, 30$. (B) SIREN (Sitzmann et al., 2020) uses sines as activation functions in MLPs and uses large initialization for the first layer (effectively creating high-frequency features). To compare KANs and baselines as fairly as possible, we try two control strategies (same shape or as,e number of parameters) and report both performance (measured by PSNR) and efficiency (wall time). All methods are listed below in Table 1. For all baseline models, 1 means their width is the same as KAN 1, while 2 means their number of parameters is (approximately) the same as KAN 1 ($\sqrt{G}$ times wider, where $G = 10$ is the grid size used in KAN 1). We also explore KAN 2, which uses a finer grid ($G = 100$ instead of $G = 10$) for the first layer only (inspired by the idea of random Fourier features in the input layer). The whole image is treated as the training set and there is no test set. All models are trained with the Adam Optimizer for 15000 steps with learning rate decay (5000 steps for learning rate $10^{-3}$, $10^{-4}$ and $10^{-5}$), with batch size 1024, on a V100 GPU.

We list PSNR and training wall time in Table 1, and fitted images in Figure 9. We have a few observations from the results: (1) KANs are comparable to or even outperform baseline methods (including SIREN) in terms of PSNR, however with more training time. (2) having random features in the inputs is useful for MLPs, especially high-frequency random features ($s = 30$ outperforms

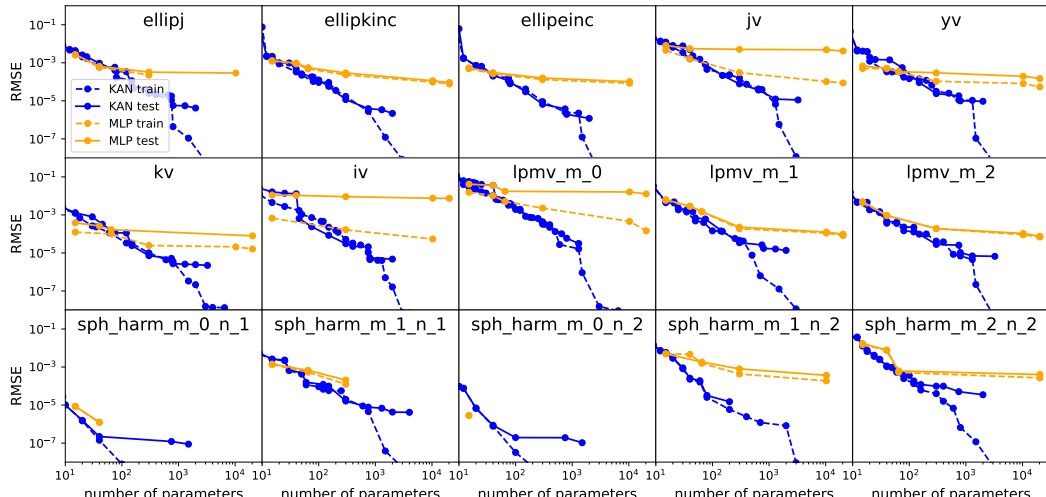

Figure 6: Fitting special functions. We show the Pareto Frontier of KANs and MLPs in the plane spanned by the number of model parameters and RMSE loss. Consistently accross all special functions, KANs have better Pareto Frontiers than MLPs. The definitions of these special functions are in Table 9.

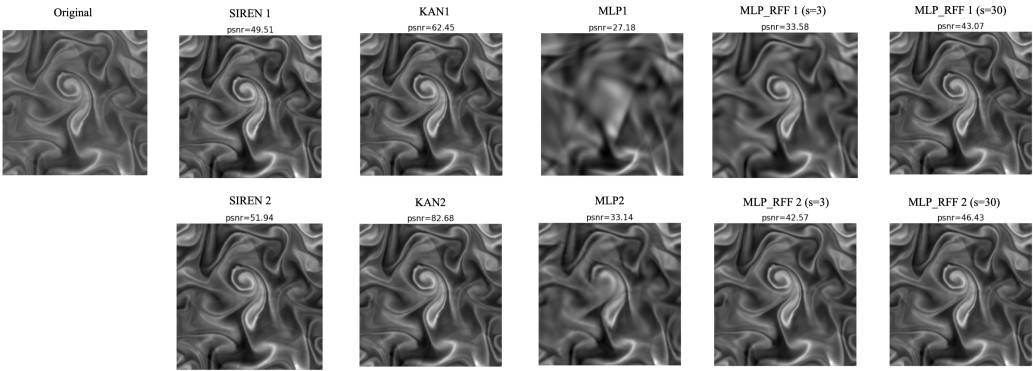

Figure 7: Image fitting task (a PDE solution from PDEBench Takamoto et al. (2022)). KAN outperforms baseline methods in terms of PSNR.

$s = 3$). We may also understand KANs' superior performance as being good at generating random features in early layers. By changing the grid size in the first layer from $G = 10$ to $G = 100$ (KAN 2), PSNR significantly increases with little additional overhead in training time. We show the turbulence profile in Figure 7. Results of the other two images can be found in Appendix A.

**Solving partial differential equations (PDEs** We consider a Poisson equation with zero Dirichlet boundary data. For $\Omega = [-1, 1]^2$, consider the PDE $u_{xx} + u_{yy} = f$ with zero boundary condition. We consider the data $f = -\pi^2(1 + 4y^2)\sin(\pi x)\sin(\pi y^2) + 2\pi \sin(\pi x)\cos(\pi y^2)$ for which $u = \sin(\pi x)\sin(\pi y^2)$ is the true solution. We use the framework of physics-informed neural networks (PINNs) Raissi et al. (2019); Karniadakis et al. (2021) to solve this PDE, with the loss function given by $\text{loss}_{\text{pde}} = \alpha\text{loss}_i + \text{loss}_b := \alpha\frac{1}{n_i}\sum_{i=1}^{n_i}|u_{xx}(z_i) + u_{yy}(z_i) - f(z_i)|^2 + \frac{1}{n_b}\sum_{i=1}^{n_b}u^2$, where we use $\text{loss}_i$ to denote the interior loss, discretized and evaluated by a uniform sampling of $n_i$ points $z_i = (x_i, y_i)$ inside the domain, and similarly we use $\text{loss}_b$ to denote the boundary loss, discretized and evaluated by a uniform sampling of $n_b$ points on the boundary. $\alpha = 0.01$ is the hyperparameter balancing the effect of the two terms. KANs are shown to have Pareto Frontiers than MLPs for this simple example.

Table 1: Image Fitting: Comparing various methods

| Method | width | PSNR ↑ | | | Training Wall Time (s) ↓ | | |
|---|---|---|---|---|---|---|---|
| | | Cam | Turb | Star | Cam | Turb | Star |
| KAN 1 | [2,128,128,128,128,1] $G =$[10,10,10,10,10] | 32.06 | 62.45 | 50.55 | 1800 | 1721 | 1715 |
| KAN 2 | [2,128,128,128,128,1] $G =$[100,10,10,10,10] | **45.76** | **82.68** | **71.82** | 1809 | 1734 | 1727 |
| MLP 1 | [2,128,128,128,128,1] | 20.76 | 27.18 | 18.28 | **162** | **92** | **91** |
| MLP 2 | [2,404,404,404,404,1] | 22.09 | 33.14 | 18.96 | 182 | 110 | 110 |
| SIREN 1 | [2,128,128,128,128,1] | 27.34 | 49.51 | 29.88 | 254 | 226 | 232 |
| SIREN 2 | [2,404,404,404,404,1] | 30.79 | 51.94 | 53.05 | 407 | 400 | 404 |
| MLP_RFF 1 ($s = 3$) | [2,128,128,128,128,1] | 22.17 | 33.58 | 19.19 | 176 | 96 | 96 |
| MLP_RFF 2 ($s = 3$) | [2,404,404,404,404,1] | 24.73 | 42.57 | 22.00 | 192 | 117 | 118 |
| MLP_RFF 1 ($s = 30$) | [2,128,128,128,128,1] | 23.92 | 43.07 | 22.68 | 174 | 96 | 97 |
| MLP_RFF 2 ($s = 30$) | [2,404,404,404,404,1] | 26.26 | 46.43 | 28.62 | 195 | 117 | 121 |

Figure 8: The PDE example. We plot L2 squared and H1 squared losses between the predicted solution and ground truth solution. First and second: training dynamics of losses. Third and fourth: scaling laws of losses against the number of parameters. KANs converge faster, achieve lower losses, and have steeper scaling laws than MLPs.

**More complicated PDEs**. We test more PDE examples in Appendix B, showing that KANs can achieve reasonable performance for more complicated PDEs. However, we want to note that KANs are slightly slower than MLPs to train in terms of wall time (reported in Appendix B), despite their smaller number of parameters. The point we want to make with Figure 8 is that KANs can achieve the theoretical scaling law in this PDE example (beyond function fitting), but this result should not be interpreted as an immediate real-world improvement. Also, both the LBFGS optimizer and the grid extension technique are required to achieve the theoretical scaling law, but in practice, people use the Adam optimizer and do not need grid extension for MLPs, which we explore in Appendix B.

## 5 CONCLUSIONS

Inspired by the Kolmogorov-Arnold representation theorem, we propose the Kolmogorov-Arnold Networks (KANs) as promising alternatives to MLPsOur contributions are three-fold: (1) we put the KA theorem in the perspective of modern machine learning, relating to MLPs, and generalize the representation from two-layer to multiple layers via the KAN layers introduced, greatly enhancing expressive power. (2) we show that KANs are interpretable, serving as a useful tool for scientific discoveries. (3) we show that KANs are accurate and have nice scaling laws via theory and experiments. The major limitation of this work, however, is that our numerical examples focus on various aspects of science and are relatively small-scale. The scalability and extensibility of KANs for large-scale machine-learning tasks are left as future work. We also acknowledge that the similarities and differences between MLPs and KANs require more study, both theoretically and empirically. For example, a reasonable criticism of KANs is that they can be rewritten as MLPs or the other way around since the notion of "edge" vs "node" is somewhat dual. Future work should aim to better clarify similarities and differences from the perspective of optimization, generalization, etc. For example, a recent preprint Wang et al. (2024b) shows that although KANs and MLPs are both universal approximators, KANs have fewer spectral biases than MLPs.

**Acknowledgement** We would like to thank Mikail Khona, Tomaso Poggio, Pingchuan Ma, Rui Wang, Di Luo, Sara Beery, Catherine Liang, Yiping Lu, Nicholas H. Nelsen, Nikola Kovachki, Jonathan W. Siegel, Hongkai Zhao, Juncai He, Shi Lab (Humphrey Shi, Steven Walton, Chuanhao Yan) and Matthieu Darcy for fruitful discussion and constructive suggestions. Z.L., F.R., J.H., M.S. and M.T. are supported by IAIFI through NSF grant PHY-2019786. The work of FR is in addition supported by the NSF grant PHY-2210333 and by startup funding from Northeastern University. Y.W and T.H are supported by the NSF Grant DMS-2205590 and the Choi Family Gift Fund. S. V. and M. S. acknowledge support from the U.S. Office of Naval Research (ONR) Multidisciplinary University Research Initiative (MURI) under Grant No. N00014-20-1-2325 on Robust Photonic Materials with Higher-Order Topological Protection.

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

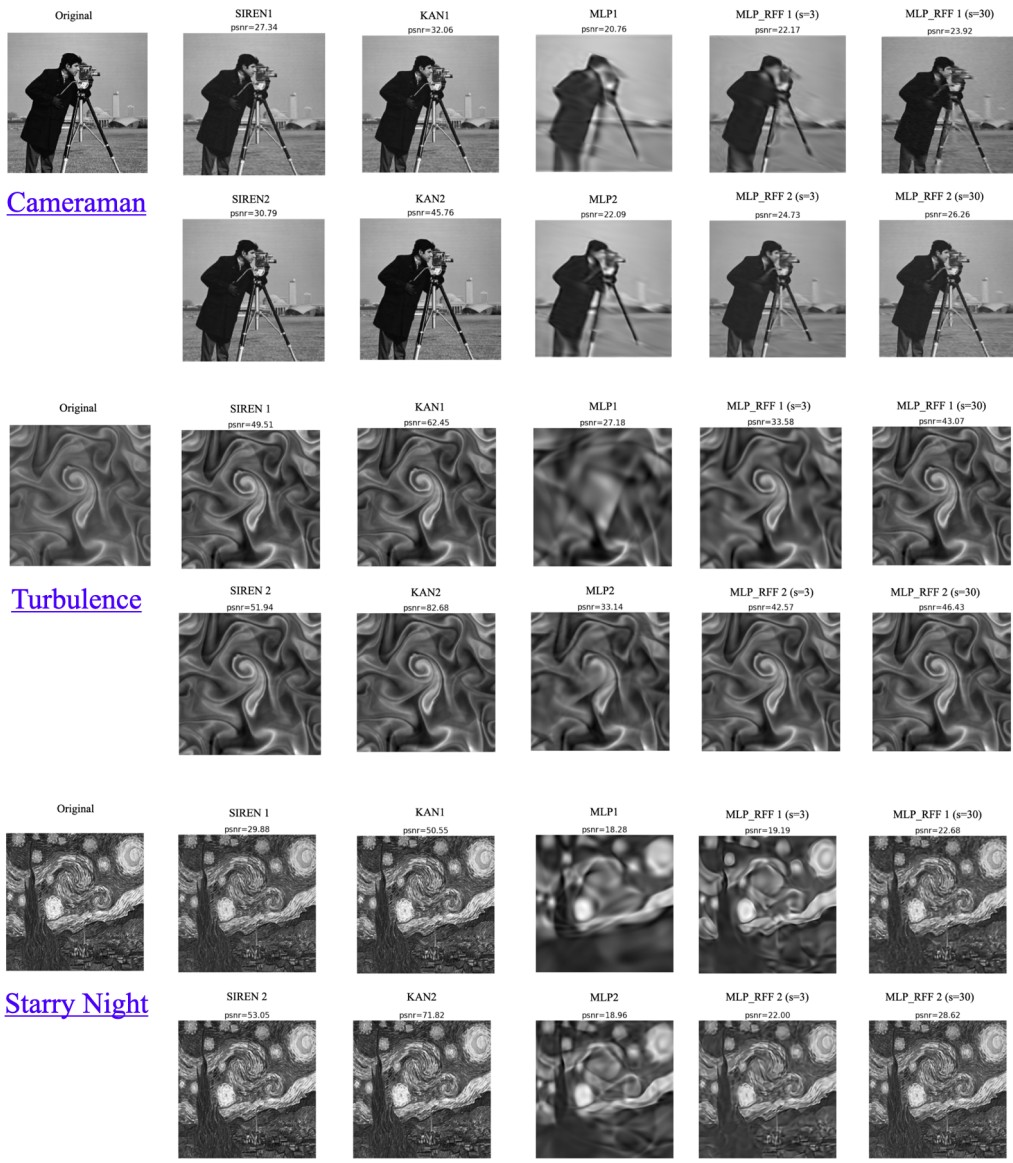

Figure 9: Comparing various methods for fitting the picture of Cameraman (top), turbulent flow (middle,from PDEBench Takamoto et al. (2022)), and Van Gogh's *the starry night* (bottom).

## A ADDITIONAL IMAGE FITTING EXAMPLES

## B ADDITIONAL PDE EXAMPLES

In Section 4, we showed KANs' superior performance over MLPs for solving a 2D Poisson equation with a smooth solution. To really understand the capabilities and limitations of KANs, we test KANs by taking three (relatively more) challenging PDEs, as suggested by reviewers: (1) Poisson equations with high-frequency solutions, to test KANs' ability to model high-frequency modes. (2) Allen-Cahn equation, to test KANs' ability to model temporal phenomenon and capture sharp transitions. (3) Darcy flow, to test KANs' ability to model random structures (e.g., Darcy flow can be used to model porous media). The goal of this section is to show that KANs (as they are) can achieve reasonable performance for these challenging PDEs, rather than attempting to establish SOTA performance. Indeed, PDE modeling with neural networks is a huge field and many techniques (e.g.,

adaptive weights (McClenny & Braga-Neto, 2020), causality training (Wang et al., 2022), the gating mechanism in PirateNet (Wang et al., 2024a)) have been developed that can be very useful to improve KANs in the future.

**Remark on the problem setup**: In Figure 8, we used the LBFGS optimizer and the grid extension technique to achieve the theoretical scaling law. However, it is more common to use the Adam optimizer and the grid extension technique is too specific to KANs. To make fair comparisons in typical user cases, below we use the Adam optimizer and do not use grid extension (the grid size is fixed once chosen), to avoid the possibility of any of our algorithmic choices favoring KANs. Said that this does not mean one should avoid using these tricks in practice when the goal is to optimize results rather than make fair comparisons. We note that KAN's training wall time is much reduced from the last version because we now disable the symbolic front (which takes up most of the training time but is unnecessary for PDE cases).

### B.1 POISSON EQUATION WITH HIGH-FREQUENCY SOLUTIONS

To test KANs' ability to approximate high-frequency PDE solutions, we revisit the Poisson equation in the main text but impose high-frequency solutions. To be specific, we consider the Poisson equation

$$u_{xx} + u_{yy} = f, \Omega \in [-1, 1]^2 \qquad (12)$$

with zero boundary condition ($u(x, -1) = u(x, 1) = u(-1, y) = u(1, y) = 0$) and $f = -2n^2\pi^2\sin(n\pi x)\sin(n\pi y)$, which has the solution $u(x, y) = \sin(n\pi x)\sin(n\pi y)$. We train our models (listed in Table 2) using Adam optimizers with a learning rate $10^{-3}$ for 1000 steps except for 10000 steps for MLP (10x training). The training loss is the PINN loss $\text{loss}_{\text{pde}} = \alpha\text{loss}_i + \text{loss}_b :=$ $\alpha\frac{1}{n_i}\sum_{i=1}^{n_i}|u_{xx}(z_i) + u_{yy}(z_i) - f(z_i)|^2 + \frac{1}{n_b}\sum_{i=1}^{n_b}u^2$, where we use $\text{loss}_i$ to denote the interior loss, discretized and evaluated by a uniform sampling of $n_i = 51^2$ points $z_i = (x_i, y_i)$ inside the domain, and similarly we use $\text{loss}_b$ to denote the boundary loss, discretized and evaluated by a uniform sampling of $n_b = 51$ points on the boundary. $\alpha = 0.01$ is the hyperparameter balancing the effect of the two terms. It is clear that larger $n$ means the solution is more high-frequency and hence more challenging. We visualize predictions by models in Figure 10, and list their relative $\ell_2$ error and training wall time in Table 2. We have a few observations: (1) All the models, i.e., KANs, MLPs, MLP_RFFs (MLP with random Fourier features) can achieve qualitatively good predictions given proper hyperparameters. (2) high-frequency Fourier features in MLP_RFFs can be harmful to training. Similarly, KANs become unstable with large grid sizes and/or depths. This is in contrast to the results in image fitting in Appendix A. Our explanation is that PINN losses have quite sharp loss landscapes, and adding high-frequency features will only make things worse. (3) KANs have slightly higher training time than MLPs, which is due to the recursive evaluations of splines, and derivatives in the PINN objective makes this problem even more severe. As we will see below, the observations drawn in this example apply to the two equations below as well.

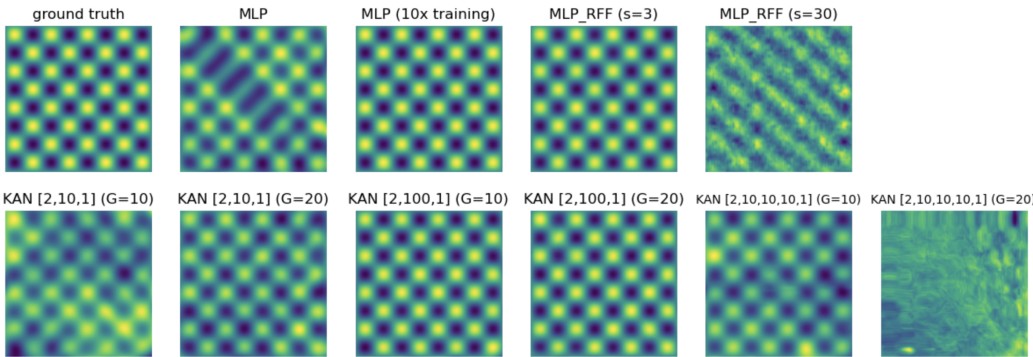

Figure 10: Comparing various methods on solving 2D Poisson equation with a high-frequency solution ($n = 4$).

| Method | $n = 1$ | | $n = 2$ | | $n = 4$ | |
|---|---|---|---|---|---|---|
| | $l_2$ error↓ | time (s) ↓ | $l_2$ error↓ | time (s) ↓ | $l_2$ error↓ | time (s) ↓ |
| MLP | 0.003 | **28** | 0.027 | **19** | 0.553 | **19** |
| MLP (10x training) | **0.001** | 184 | 0.328 | 184 | **0.022** | 202 |
| MLP_RFF ($s = 3$) | **0.001** | 254 | **0.001** | 233 | 0.084 | 232 |
| MLP_RFF ($s = 30$) | 1.000 | 250 | 0.999 | 242 | 0.934 | 249 |
| KAN [2,10,1] $G = 10$ | 0.006 | **28** | 0.135 | 29 | 0.729 | 24 |
| KAN [2,10,1] $G = 20$ | 0.221 | 33 | 0.082 | 33 | 0.295 | 34 |
| KAN [2,100,1] $G = 10$ | **0.001** | 51 | 0.006 | 52 | 0.099 | 51 |
| KAN [2,100,1] $G = 20$ | 0.326 | 72 | 0.135 | 71 | 0.090 | 74 |
| KAN [2,10,10,10,1] $G = 10$ | 0.012 | 89 | 0.117 | 95 | 0.576 | 92 |
| KAN [2,10,10,10,1] $G = 20$ | 0.995 | 127 | 0.993 | 130 | 0.982 | 125 |

Table 2: Comparing various method on solving 2D Poisson equations. All MLPs (including MLP_RFFs) have shapes [2,128,128,128,1].

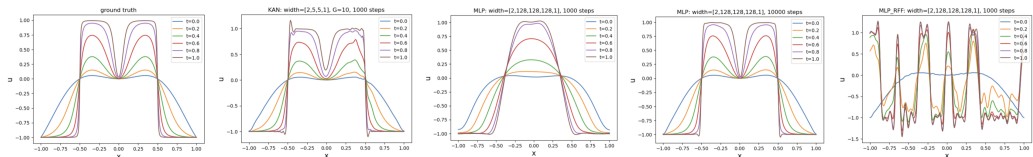

Figure 11: Solving the 1D Allen-Cahn equation.

## B.2 ALLEN-CAHN EQUATION

To test KANs' ability to solve temporal PDEs and model phase transitions, we consider the one-dimensional Allen-Cahn equation with the periodic boundary conditions and the quartic double-well potential energy, formulated as below

$$
\begin{aligned}
u_t - 0.0001u_{xx} + 5(u^3 - u) &= 0, x \in [-1, 1], t \in [0, 1], \\
u(x, 0) &= x^2 \cos(\pi x) \equiv u_0(x), \\
u(-1, t) &= u(1, t), \\
u_x(-1, t) &= u_x(1, t).
\end{aligned}
\tag{13}
$$

Since we do not have an exact solution, a reference solution is obtained via the direct Euler forward method (1000 mesh points in space and in time). Although this temporal equation seems innocuous, using standard PINN training (using MLPs) can lead to a problem – the solution would collapse to a zero solution very quickly in time! To solve this issue, Wang et al. (2022) proposes causal training where the temporal domain is divided into several blocks. Each block corresponds to a separate PINN and these PINNs are trained sequentially in time, where the previous block is used to initialize the next block. Each block covers $\Delta t = 0.1$, so there are 10 blocks in total. The training loss is the PINN loss $\text{loss}_{\text{pde}} = \alpha_i \text{loss}_i + \alpha_b \text{loss}_b + \alpha_t \text{loss}_t := \alpha_i \frac{1}{n_i} \sum_{i=1}^{n_i} |u_t - 0.0001u_{xx} + 5(u^3 - u)|^2 + \alpha_b \frac{1}{n_b} \sum_{i=1}^{n_b} (u(1, t_i) - u(-1, t_i))^2 + (u_x(1, t_i) - u_x(-1, t_i))^2 + \alpha_t \frac{1}{n_t} \sum_{i=1}^{n_t} (u(x_i, 0) - u_0(x_i))^2$, where we use $\text{loss}_i$ to denote the interior loss, discretized and evaluated by a uniform sampling of $n_i = 51^2$ points $z_i = (x_i, y_i)$ inside the domain, and similarly we use $\text{loss}_b$ to denote the boundary loss, discretized and evaluated by a uniform sampling of $n_b = 51$ points on the boundary. $\text{loss}_t$ to denote the initial profile, discretized and evaluated by a uniform sampling of $n_t = 51$ points on the boundary. We choose $\alpha_i = 1, \alpha_b = 1, \alpha_t = 100$. We train each temporal block with the Adam optimizer with a learning rate $10^{-3}$ for 1000 steps. We show in Figure 11 their prediction profiles. With 1000 training steps, KANs have already learned good qualitative evolution (although with some imperfections). Training KANs for 10000 steps probably helps, but that will take about 10h to train so we did not try this given the limited time during rebuttal. With 1000 training steps, MLPs do not learn the correct qualitative evolution, but adding training steps to 10000 makes MLPs learn

| Model | $l_2^2$ error $\downarrow$ | Training wall time (s) $\downarrow$ |
|---|---|---|
| KAN [2,5,5,1] $G = 5$ | $3.4 \times 10^{-3}$ | 2801 |
| KAN [2,5,5,1] $G = 10$ | $3.9 \times 10^{-3}$ | 2831 |
| MLP | $1.5 \times 10^{-1}$ | **478** |
| MLP (10x training) | $\mathbf{3.9 \times 10^{-4}}$ | 4766 |
| MLP_RFF ($s = 30$) | $8.0 \times 10^{-1}$ | 599 |

Table 3: Comparing various models on the 1D Allen-Cahn equation. All MLPs including MLP_RFFs have the shape [2,128,128,128,1].

the evolution quite accurately. For MLP with high-frequency random features ($s = 30$), the training curve fails to decrease, which is similar to the observation in the Poisson case in Appendix B.1.

## B.3 DARCY FLOW

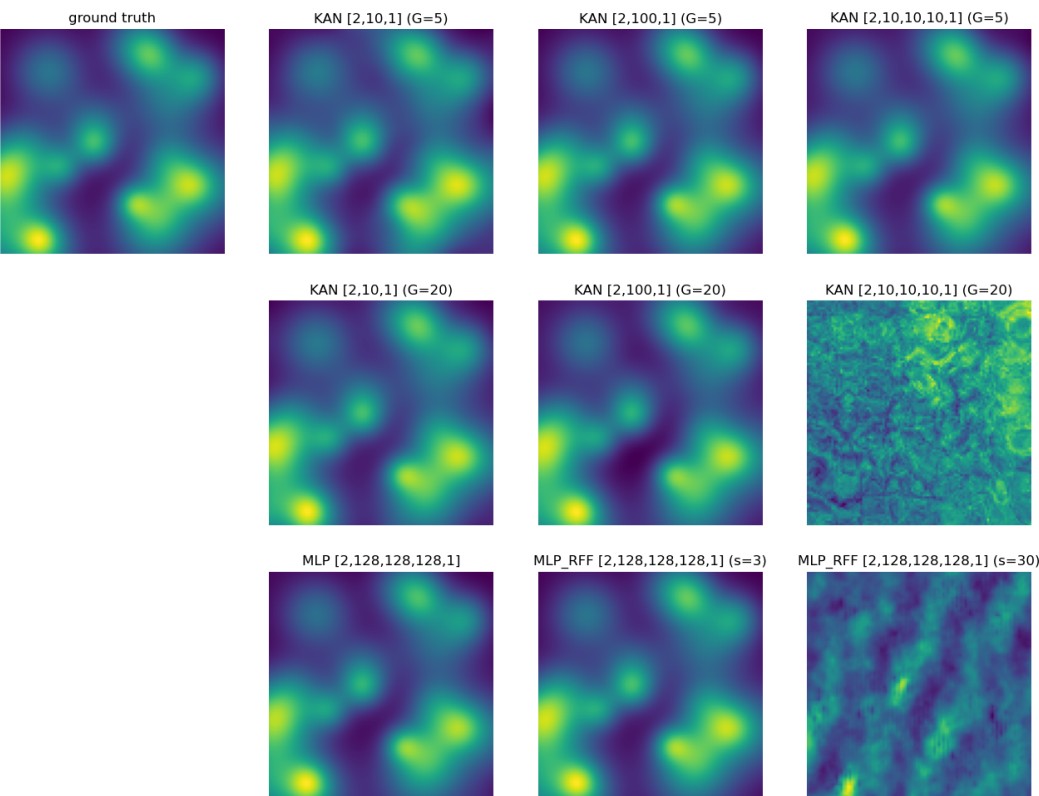

Figure 12: Prediction of various models on Darcy flow

We use Darcy flow to test KANs' ability to model random media (modeled as Gaussian mixtures). The equation is similar to the Poisson equation, but the permeability $a(x)$ can be spatially dependent ($\mathbf{x} \equiv (x, y)$):

| Model | $l_2^2$ error $\downarrow$ | Training wall time (s) $\downarrow$ |
|---|---|---|
| KAN [2,10,1] $G = 5$ | $3.9 \times 10^{-3}$ | 71 |
| KAN [2,10,1] $G = 10$ | $1.3 \times 10^{-3}$ | 66 |
| KAN [2,10,1] $G = 20$ | $3.9 \times 10^{-4}$ | 66 |
| KAN [2,100,1] $G = 5$ | $1.7 \times 10^{-5}$ | 81 |
| KAN [2,100,1] $G = 10$ | $\mathbf{4.3 \times 10^{-6}}$ | 107 |
| KAN [2,100,1] $G = 20$ | $1.9 \times 10^{-2}$ | 136 |
| KAN [2,10,10,10,10,1] $G = 5$ | $8.5 \times 10^{-5}$ | 123 |
| KAN [2,10,10,10,10,1] $G = 10$ | 1.2 | 123 |
| KAN [2,10,10,10,10,1] $G = 20$ | 1.3 | 125 |
| MLP | $3.0 \times 10^{-5}$ | $\mathbf{30}$ |
| MLP (10x training) | $4.5 \times 10^{-6}$ | 277 |
| MLP_RFF $(s = 3)$ | $5.9 \times 10^{-6}$ | 31 |
| MLP_RFF $(s = 30)$ | $4.0 \times 10^{-1}$ | 31 |

Table 4: Comparing various models on darcy flow. All MLPs including MLP_RFFs have the shape [2,128,128,128,1].

$$\nabla \cdot (a(\mathbf{x}) \cdot \nabla u(\mathbf{x})) = f(\mathbf{x}), \mathbf{x} \in \Omega = [-1, 1]^2,$$
$$u(\mathbf{x}) = u_t(\mathbf{x}), \mathbf{x} \in \partial\Omega,$$
$$a(\mathbf{x}) = 1 + \sum_{i=1}^{N_a} \exp\left(-\frac{(x - x_{a,i}^2) + (y - y_{a,i}^2)}{2\sigma_{a,i}^2}\right), x_{a,i}, y_{a,i} \sim U[-1, 1], \sigma_{a,i} \sim U[0.1, 0.3],$$
$$u_t(\mathbf{x}) = \sum_{i=1}^{N_u} \exp\left(-\frac{(x - x_{u,i}^2) + (y - y_{u,i}^2)}{2\sigma_{u,i}^2}\right), x_{u,i}, y_{u,i} \sim U[-1, 1], \sigma_{u,i} \sim U[0.1, 0.3].$$

$$(14)$$

Our setup is exactly the same as the Poisson equation in B.1, except that the ground truth solution is different, and the left differential operator is slightly different from simple Laplacian. We visualize prediction solutions in Figure 12 and report errors and training time in Table 12.

### B.4 DISCUSSION

From the three examples above, we conclude that KANs can produce reasonable performance for PDE solving, but face a few challenges that should be addressed to make them competitive with SOTA PDE methods: (1) Slow training. In the image fitting task, we find that KANs typically have $2^k$ more wall time than MLPs of same sizes, due to the recursive computation of splines. However, the slowdown factor is even worse for PDE solving. Potential solutions include more efficient computations of splines (e.g., pre-computing spline coefficients), or using other activation functions (e.g., Fourier bases or radial basis functions) to avoid recursive evaluations. (2) Stability at large depths and large grid sizes. In image fitting, we find that larger depths and larger grids lead to better performance. However for PDEs, shallow but wide KANs typically perform better than deep KANs. When grid size is small, increasing it can gain more accuracy; however, when grid size reaches, say, 20, training can be totally messed up. Potential solutions include leveraging gating mechanisms as in PirateNet (Wang et al., 2024a), adding residual connections (He et al., 2016), and trying other regimes (e.g., deep Ritz method) instead of the PINN loss.

## C   MNIST

To test KANs' scalability for high-dimensional datasets, we train KANs on MNIST. We normalize the pixel values into [0,1], and flatten the 28x28 image into a 784-dimensional vector. We train models (listed in Table 5) with the Adam optimizer ($10^{-2}$ learning rate) for 2000 steps on the cross-entropy loss, with batch size 1024. The whole training dataset (60000) and test dataset (10000) are used to evaluate train/test loss/acc. We report these metrics in Table 5.

There are a few observations: (1) the shape [784,100,10] (1 hidden layer of size 100) is optimal both for MLP and for KAN, which is an interesting observation. This seems to imply there is something

universal across different architectures. (2) the effect of grid size: increasing grid size decreases training loss (since it enhances fitting capability), however, the test metrics may get worse (e.g., for [784,10]) or may get better (e.g., for [784,100,10]). Combined with (1), this seems to imply that increasing grid size is beneficial when the shape of the network is correct but might be harmful otherwise. (3) KANs and MLPs have comparable performance in terms of loss and accuracy. This is probably because the MNIST dataset is too simple. KANs consume much more training time than MLPs - Besides the $2^k$ factor slowdown due to the recursive computation of order-$k$ splines, grid updates are also quite expensive due to the high-dimensional inputs. We expect these slowdown factors to have straightforward solutions, and combining KANs with Convolutional neural networks is a promising direction to incorporate symmetry inductive biases into architectures.

| Model | Train loss ↓ | Test loss ↓ | Train Acc ↑ | Test Acc ↑ | Time (s) ↓ |
|---|---|---|---|---|---|
| KAN [784,10] $G = 3$ | $1.5 \times 10^{-1}$ | $2.8 \times 10^{-1}$ | 95.7% | 93.0% | 83.8 |
| KAN [784,10] $G = 5$ | $9.9 \times 10^{-2}$ | $3.3 \times 10^{-1}$ | 97.0% | 92.4% | 96.8 |
| KAN [784,10] $G = 10$ | $3.4 \times 10^{-2}$ | $4.4 \times 10^{-1}$ | 99.1% | 91.7% | 155.4 |
| MLP [784,10] | $2.3 \times 10^{-1}$ | $2.8 \times 10^{-1}$ | 93.7% | 92.5% | 5.8 |
| KAN [784,10,10] $G = 3$ | $8.3 \times 10^{-2}$ | $2.2 \times 10^{-1}$ | 97.5% | 94.5% | 106.8 |
| KAN [784,10,10] $G = 5$ | $4.0 \times 10^{-2}$ | $3.1 \times 10^{-1}$ | 98.7% | 94.2% | 121.8 |
| KAN [784,10,10] $G = 10$ | $1.8 \times 10^{-2}$ | $3.7 \times 10^{-1}$ | 99.4% | 94.2% | 168.0 |
| MLP [784,10,10] | $1.6 \times 10^{-1}$ | $2.3 \times 10^{-1}$ | 95.1% | 93.7% | 6.3 |
| KAN [784,100,10] $G = 3$ | $4.0 \times 10^{-2}$ | $2.0 \times 10^{-1}$ | 99.0% | 97.4% | 419.3 |
| KAN [784,100,10] $G = 5$ | $5.6 \times 10^{-5}$ | $9.6 \times 10^{-2}$ | **100.0%** | **98.2%** | 435.9 |
| KAN [784,100,10] $G = 10$ | $3.8 \times 10^{-5}$ | $9.2 \times 10^{-2}$ | **100.0%** | **98.2%** | 531.8 |
| MLP [784,100,10] | $3.5 \times 10^{-4}$ | $9.7 \times 10^{-2}$ | **100.0%** | **97.9%** | 8.3 |
| KAN [784,100,100,10] $G = 3$ | $1.3 \times 10^{-2}$ | $1.8 \times 10^{-1}$ | 99.6% | 97.6% | 498.6 |
| KAN [784,100,100,10] $G = 5$ | $1.6 \times 10^{-2}$ | $1.9 \times 10^{-1}$ | 99.5% | 97.6% | 551.1 |
| KAN [784,100,100,10] $G = 10$ | $1.1 \times 10^{-2}$ | $2.0 \times 10^{-1}$ | 99.7% | 97.5% | 655.3 |
| MLP [784,100,100,10] | $1.0 \times 10^{-2}$ | $1.5 \times 10^{-1}$ | 99.7% | 97.7% | 9.6 |
| KAN [784,100,100,100,10] $G = 3$ | $1.3 \times 10^{-2}$ | $1.8 \times 10^{-1}$ | 99.6% | 97.8% | 631.5 |
| KAN [784,100,100,100,10] $G = 5$ | $1.9 \times 10^{-2}$ | $1.6 \times 10^{-1}$ | 99.4% | 97.3% | 643.6 |
| KAN [784,100,100,100,10] $G = 10$ | $1.5 \times 10^{-2}$ | $1.6 \times 10^{-1}$ | 99.6% | 97.4% | 813.3 |
| MLP [784,100,100,100,10] | $1.8 \times 10^{-2}$ | $1.4 \times 10^{-1}$ | 99.6% | 97.7% | 11.0 |

Table 5: Comparing KANs and MLPs on MNIST

## D  HYPERPARAMETER SEARCH OF MLPS AND KANS

To help understand how hyper-parameters affect the comparison of KAN vs MLP performance, we conduct a hyperparameter sweeping for the function fitting task $f(x, y) = \exp(\sin(\pi x) + y^2)$ (randomly generated 1000 training and test samples from $U[-1, 1]^2$). We sweep a few things below:

- Optimizer: Adam or LBFGS.

- Learning rate: For Adam, we choose learning rates from $\{10^{-4}, 3 \times 10^{-4}, 10^{-3}, 3 \times 10^{-3}, 10^{-2}\}$. For LBFGS, we choose learning rates from $\{10^{-2}, 3 \times 10^{-2}, 10^{-1}, 3 \times 10^{-1}, 1\}$.

- Network width: 10 or 100.

- Network Depth: 2, 3, or 4.

When we use Adam, we train MLPs for 60000 steps and train KANs with 10000*6 steps (10000 steps for each grid size $\{3,5,10,20,50,100\}$). In Figure 13, we show the train/test losses for MLPs (red) and KANs (green) under different conditions. We have a few observations: (1) learning rate does not seem to play a big effect (except that large learning rates for Adam lead to more oscillations). (2) In terms of test performance, KANs outperform MLPs on shallow models (2 or 3 layers), but are comparable for 4-layer models. (3) In terms of training performance, KANs can fit to much lower losses than MLPs, both with LBFGS and Adam (despite wild oscillations for Adam, which can probably be mitigated by learning rate decay). KANs are prone to "overfit", which might be good or bad depending on the context, i.e., we expect KANs to improve test performance with more

training data, but we also expect KANs to have more performance degradation when data contains noises.

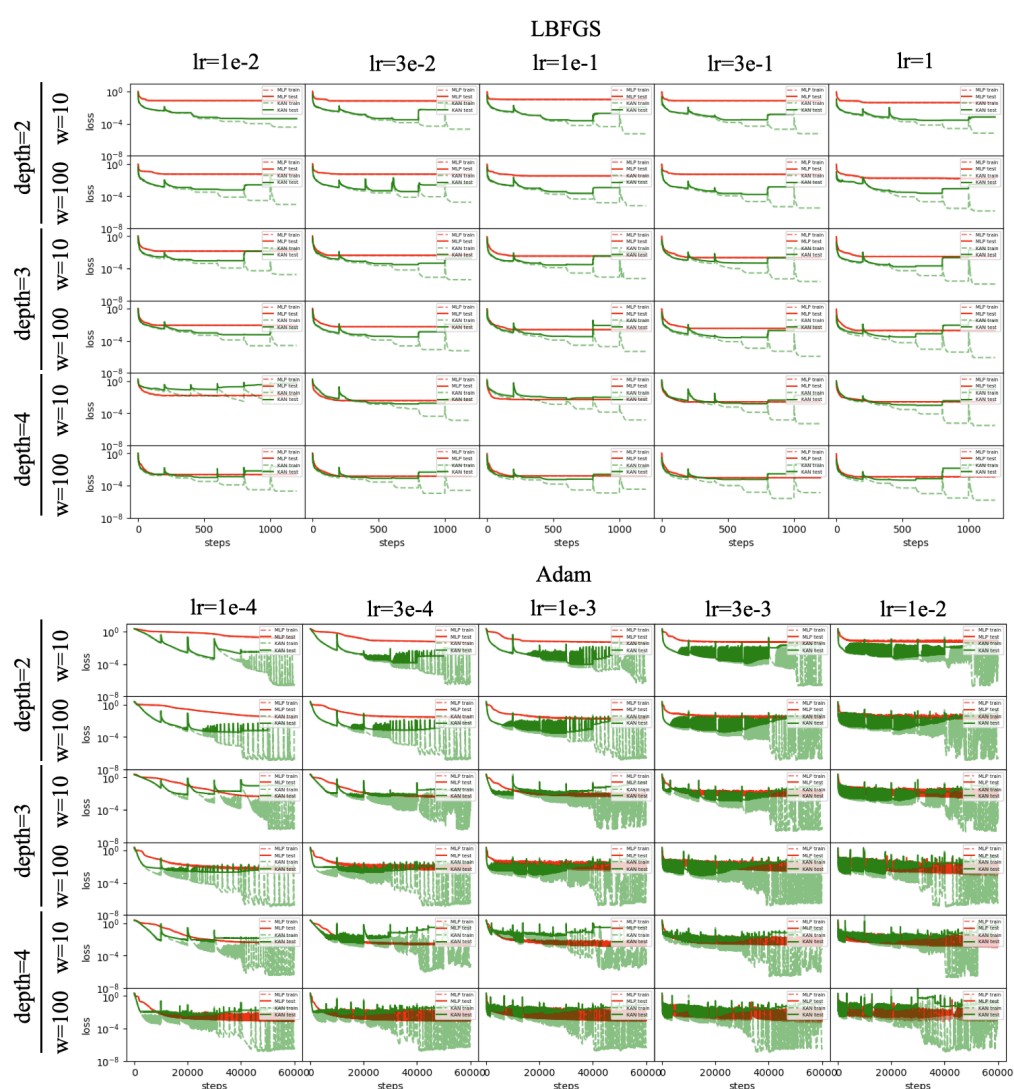

Figure 13: Hyperparameter search for KANs and MLPs. Hyperparameters include depths, widths, optimization methods, and learning rates.

# E   WHAT IF THE NETWORK HAS MORE LAYERS THAN NEEDED?

In Figure 2, we have demonstrated that when the KAN network has a depth smaller than needed, the learned activation functions can be highly oscillatory, appearing to fit some non-smooth functions. We are also curious about what happens if the network is deeper than needed. We consider fitting a 2D function $f(x, y) = \exp(\sin(\pi x) + y^2)$. We know that a 2L KAN can smoothly represent the function (Figure 14 left). When we attempt to fit the function with more layers (with sparsity penalty), some edges would become (nearly) identities, shown for 3L (middle) and 4L (right). The identity shortcuts are easy to form with linear residuals, while SiLU residuals can lead to a complex network structure even under sparsity penalty. We suspect this is a pathology when depth becomes larger, which we want to investigate in the future.

| Method | lr = 1e-2 | | lr = 1e-1 | | lr = 1 | |
|---|---|---|---|---|---|---|
| | train loss↓ | test loss↓ | train loss↓ | test loss↓ | train loss↓ | test loss↓ |
| MLP D=2, W=10 | $7.2 \times 10^{-2}$ | $7.7 \times 10^{-2}$ | $6.6 \times 10^{-2}$ | $7.1 \times 10^{-2}$ | $1.0 \times 10{-1}$ | $1.0 \times 10^{-1}$ |
| MLP D=2, W=100 | $5.1 \times 10^{-2}$ | $5.4 \times 10^{-2}$ | $4.9 \times 10^{-2}$ | $5.3 \times 10^{-2}$ | $3.1 \times 10^{-2}$ | $3.3 \times 10^{-2}$ |
| MLP D=3, W=10 | $1.4 \times 10^{-2}$ | $1.4 \times 10^{-2}$ | $4.2 \times 10^{-3}$ | $4.5 \times 10^{-3}$ | $3.5 \times 10^{-3}$ | $3.6 \times 10^{-3}$ |
| MLP D=3, W=100 | $8.1 \times 10^{-3}$ | $8.8 \times 10^{-3}$ | $5.5 \times 10^{-3}$ | $5.7 \times 10^{-3}$ | $2.5 \times 10^{-3}$ | $2.5 \times 10^{-3}$ |
| MLP D=4, W=10 | $1.5 \times 10^{-2}$ | $1.7 \times 10^{-2}$ | $3.6 \times 10^{-3}$ | $3.8 \times 10^{-3}$ | $5.0 \times 10^{-3}$ | $5.2 \times 10^{-3}$ |
| MLP D=4, W=100 | $2.4 \times 10^{-3}$ | $2.5 \times 10^{-3}$ | $1.4 \times 10^{-3}$ | $1.5 \times 10^{-3}$ | $1.6 \times 10^{-3}$ | $1.7 \times 10^{-3}$ |
| KAN D=2, W=10 | $1.2 \times 10^{-5}$ | $\mathbf{2.0 \times 10^{-4}}$ | $4.5 \times 10^{-6}$ | $2.5 \times 10^{-4}$ | $4.2 \times 10^{-6}$ | $2.7 \times 10^{-4}$ |
| KAN D=2, W=100 | $7.3 \times 10^{-6}$ | $2.9 \times 10^{-4}$ | $4.2 \times 10^{-6}$ | $\mathbf{1.9 \times 10^{-4}}$ | $\mathbf{1.3 \times 10^{-6}}$ | $\mathbf{2.2 \times 10^{-4}}$ |
| KAN D=3, W=10 | $\mathbf{7.1 \times 10^{-6}}$ | $8.0 \times 10^{-4}$ | $4.2 \times 10^{-6}$ | $4.3 \times 10^{-4}$ | $2.5 \times 10^{-6}$ | $3.6 \times 10^{-4}$ |
| KAN D=3, W=100 | $2.0 \times 10^{-5}$ | $1.5 \times 10^{-3}$ | $5.6 \times 10^{-6}$ | $7.7 \times 10^{-4}$ | $8.8 \times 10^{-6}$ | $9.1 \times 10^{-4}$ |
| KAN D=4, W=10 | $2.2 \times 10^{-5}$ | $5.4 \times 10^{-6}$ | $\mathbf{3.9 \times 10^{-6}}$ | $2.4 \times 10^{-3}$ | $3.9 \times 10^{-6}$ | $2.4 \times 10^{-3}$ |
| MLP D=4, W=100 | $8.2 \times 10^{-6}$ | $1.6 \times 10^{-3}$ | $8.4 \times 10^{-6}$ | $1.6 \times 10^{-3}$ | $2.7 \times 10^{-6}$ | $8.5 \times 10^{-4}$ |

Table 6: Results for the example: $f(x, y) = \exp(\sin(\pi x + y^2))$ and the LBFGS optimizer. Grid search width, depth, and learning rate.

| Method | lr = 1e-4 | | lr = 1e-3 | | lr = 1e-2 | |
|---|---|---|---|---|---|---|
| | train loss↓ | test loss↓ | train loss↓ | test loss↓ | train loss↓ | test loss↓ |
| MLP D=2, W=10 | $1.6 \times 10^{-1}$ | $1.7 \times 10^{-1}$ | $4.6 \times 10^{-2}$ | $4.9 \times 10^{-2}$ | $4.7 \times 10^{-2}$ | $5.2 \times 10^{-2}$ |
| MLP D=2, W=100 | $2.3 \times 10^{-2}$ | $2.4 \times 10^{-2}$ | $2.4 \times 10^{-2}$ | $2.6 \times 10^{-2}$ | $1.6 \times 10^{-2}$ | $1.7 \times 10^{-2}$ |
| MLP D=3, W=10 | $4.1 \times 10^{-3}$ | $4.2 \times 10^{-3}$ | $3.4 \times 10^{-3}$ | $3.4 \times 10^{-3}$ | $4.1 \times 10^{-3}$ | $4.3 \times 10^{-3}$ |
| MLP D=3, W=100 | $3.5 \times 10^{-3}$ | $4.1 \times 10^{-3}$ | $3.0 \times 10^{-3}$ | $3.9 \times 10^{-3}$ | $2.7 \times 10^{-3}$ | $4.0 \times 10^{-3}$ |
| MLP D=4, W=10 | $3.4 \times 10^{-3}$ | $3.7 \times 10^{-3}$ | $2.1 \times 10^{-3}$ | $2.1 \times 10^{-3}$ | $1.5 \times 10^{-3}$ | $1.5 \times 10^{-3}$ |
| MLP D=4, W=100 | $7.3 \times 10^{-4}$ | $8.3 \times 10^{-4}$ | $7.6 \times 10^{-4}$ | $9.2 \times 10^{-4}$ | $7.9 \times 10^{-4}$ | $9.4 \times 10^{-4}$ |
| KAN D=2, W=10 | $2.4 \times 10^{-7}$ | $3.4 \times 10^{-4}$ | $2.4 \times 10^{-7}$ | $\mathbf{1.0 \times 10^{-4}}$ | $2.2 \times 10^{-7}$ | $\mathbf{1.3 \times 10^{-4}}$ |
| KAN D=2, W=100 | $1.4 \times 10^{-7}$ | $\mathbf{2.0 \times 10^{-4}}$ | $1.6 \times 10^{-7}$ | $3.1 \times 10^{-4}$ | $\mathbf{1.6 \times 10^{-7}}$ | $4.8 \times 10^{-4}$ |
| KAN D=3, W=10 | $1.5 \times 10^{-6}$ | $7.1 \times 10^{-3}$ | $2.1 \times 10^{-7}$ | $4.4 \times 10^{-3}$ | $3.2 \times 10^{-7}$ | $2.5 \times 10^{-3}$ |
| KAN D=3, W=100 | $\mathbf{1.2 \times 10^{-7}}$ | $8.2 \times 10^{-4}$ | $\mathbf{1.8 \times 10^{-7}}$ | $1.2 \times 10^{-3}$ | $1.8 \times 10^{-7}$ | $1.7 \times 10^{-3}$ |
| KAN D=4, W=10 | $2.2 \times 10^{-5}$ | $2.5 \times 10^{-2}$ | $2.8 \times 10^{-6}$ | $1.0 \times 10^{-2}$ | $5.6 \times 10^{-7}$ | $1.0 \times 10^{-2}$ |
| KAN D=4, W=100 | $1.9 \times 10^{-7}$ | $3.2 \times 10^{-3}$ | $1.9 \times 10^{-7}$ | $2.5 \times 10^{-3}$ | $3.4 \times 10^{-7}$ | $2.7 \times 10^{-3}$ |

Table 7: Results for the example: $f(x, y) = \exp(\sin(\pi x + y^2))$ and the Adam optimizer. Grid search width, depth, and learning rate.

## F    EXISTENCE OF REDUNDANT NEURONS/EDGES WITH SMALL OR NO SPARSITY PENALTY

In Figure 15, we show that sparsity penalty strength $\lambda$ controls the number of redundant neurons. When $\lambda = 0$, all five neurons appear to be active, with a few neurons/edges appearing to be highly similar. When $\lambda = 0.001$, only two neurons are active and they appear almost identical (except that they differ by a minus sign). When $\lambda = 0.1$, there is only one active neuron in the hidden layer hence there is no redundant neuron.

## G    THE NECESSITY OF SKIP CONNECTIONS

In Figure 16, we show the necessity of using skip connections, i.e., the learnable function $f(x) = b(x) + \text{spline}(x)$ with non-zero $b(x)$. By default we choose $b(x) = \text{SiLU}(x)$. To test the necessity of such a $b(x)$, we use a simple 2D function regression task $f(x, y) = \exp(\sin(\pi x) + y^2)$. [2,1,1] KANs ($G = 10$) are trained with the LBFGS optimizer with samples drawn from $U[-1, 1]^2$ (1000 training and 1000 test samples). We visualize KANs at step 10: KANs using SiLU and linear residual connections have already learned the correct representation, while KANs without skip connections still struggle to learn the correct representations.

The intuition is: since B-splines are piecewise polynomials, they behave like order-$k$ polynomials locally. When KANs become deeper (layer $L$) without skip connections, the function would become

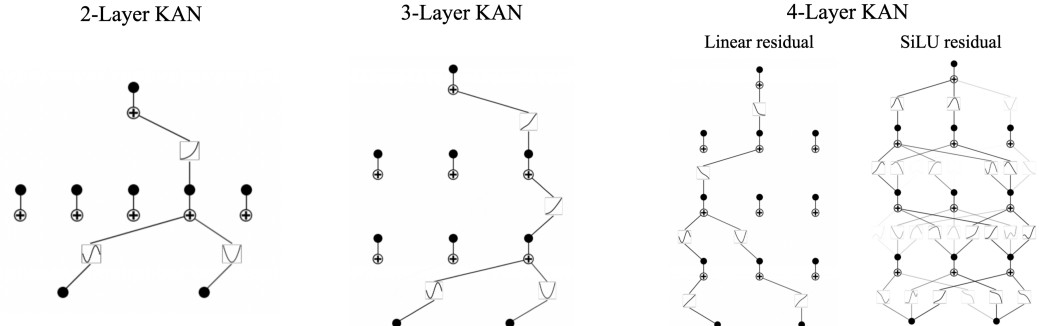

Figure 14: Fitting a 2D function $f(x, y) = \exp(\sin(\pi x) + y^2)$. We know that a 2L KAN can smoothly represent the function (left). When we attempt to fit the function with more layers (with sparsity penalty), some edges would become (nearly) identities, shown for 3L (middle) and 4L (right). The identity shortcuts are easy to form with linear residuals, while SiLU residuals can lead to a complex network structure even under sparsity penalty.

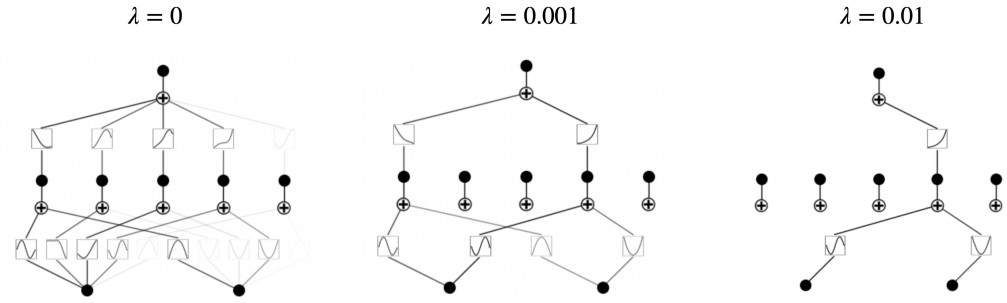

Figure 15: Existence of redundant neurons/edges with small or no sparsity penalty. $\lambda$ is the sparsity penalty strength.

order-$kL$ polynomials which is quite pathological (it is known that high-order polynomials have bad numerical properties). By including the skip connections, the function can have low-order polynomial components by leveraging the skip connections.

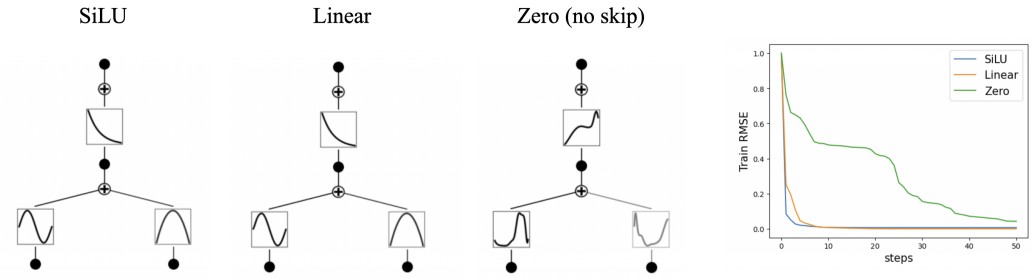

Figure 16: Skip connections (either SiLU or linear) make training landscapes smoother, leading to faster training. The visualizations are for KANs at step 10.

## H  THE NECESSITY OF GRID EXTENSIONS

One may ask: why don't we just use the large grid size from scratch, instead of using a small grid size first and then do grid extension? In Figure 17, we show that a KAN with a large grid size can easily get stuck at local minima (probably due to a bad loss landscape). By contrast, a well-initialized KAN with a large grid size (obtained by grid extension from smaller grid sizes) does not have such a problem. With grid extension: we train the model starting from $G = 3$ for 50 steps with LBFGS, and then we do grid extension to increase $G$ to be $5, 10, 20$ (each grid is trained for another 50 steps). Without grid extension: The KAN is initialized to have $G = 20$ and is trained for 200 steps.

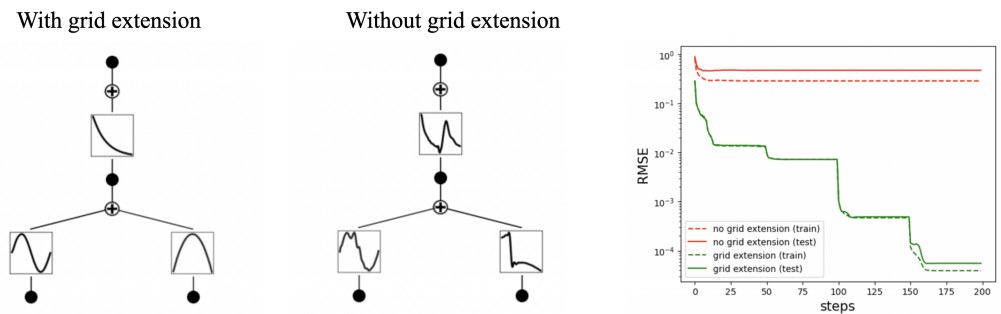

Figure 17: Grid extension is needed to avoid bad loss landscapes when initial grid sizes are large.

## I  IMPLEMENTATION DETAILS OF KAN

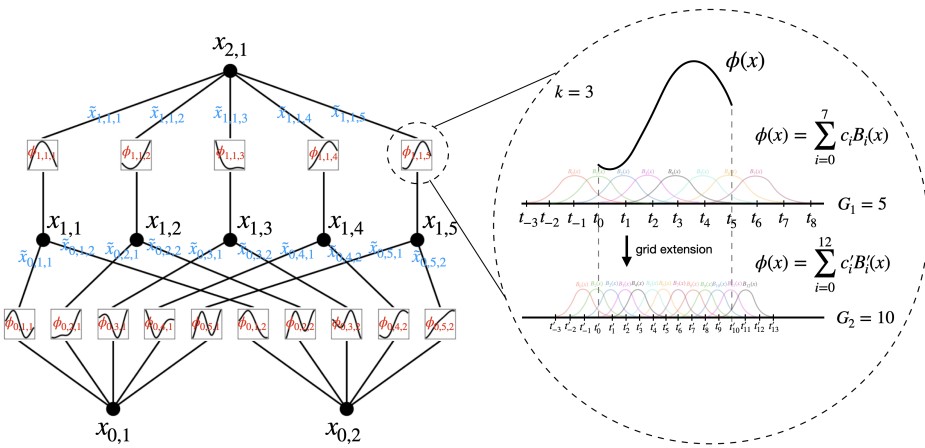

Figure 18: Left: Notations of activations that flow through the network. Right: an activation function is parameterized as a B-spline, which allows switching between coarse-grained and fine-grained grids.

**Implementation details.** Although a KAN layer Eq. (5) looks extremely simple, it is non-trivial to make it well optimizable. The key tricks are:

(1) Residual activation functions. We include a basis function $b(x)$ (similar to residual connections) such that the activation function $\phi(x)$ is the sum of the basis function $b(x)$ and the spline function:

$$\phi(x) = w_b b(x) + w_s \text{spline}(x). \tag{15}$$

We set

$$b(x) = \text{silu}(x) = x/(1 + e^{-x}) \tag{16}$$

in most cases. $\mathrm{spline}(x)$ is parametrized as a linear combination of B-splines such that

$$\mathrm{spline}(x) = \sum_i c_i B_i(x) \qquad (17)$$

where $c_i$s are trainable (see Figure 18 for an illustration). In principle $w_b$ and $w_s$ are redundant since it can be absorbed into $b(x)$ and $\mathrm{spline}(x)$. However, we still include these factors (which are by default trainable) to better control the overall magnitude of the activation function.

(2) Initialization scales. Each activation function is initialized to have $w_s = 1$ and $\mathrm{spline}(x) \approx 0$ [2]. $w_b$ is initialized according to the Xavier initialization, which has been used to initialize linear layers in MLPs.

(3) Update of spline grids. We update each grid on the fly according to its input activations, to address the issue that splines are defined on bounded regions but activation values can evolve out of the fixed region during training [3] Grid updates (grid size $G_1 \to G_1$) use the same least square method as grid extensions (grid size $G_1 \to G_2 > G_1$), as detailed in L.

**Parameter count.** For simplicity, let us assume a network

(1) of depth $L$,

(2) with layers of equal width $n_0 = n_1 = \cdots = n_L = N$,

(3) with each spline of order $k$ (usually $k = 3$) on $G$ intervals (for $G + 1$ grid points).

Then there are in total $O(N^2 L(G + k)) \sim O(N^2 LG)$ parameters. In contrast, an MLP with depth $L$ and width $N$ only needs $O(N^2 L)$ parameters, which appears to be more efficient than KAN. Fortunately, KANs usually require much smaller $N$ than MLPs, which not only saves parameters, but also achieves better generalization (see e.g., Figure 5 and 8) and facilitates interpretability. We remark that for 1D problems, we can take $N = L = 1$ and the KAN network in our implementation is nothing but a spline approximation. For higher dimensions, we characterize the generalization behavior of KANs with a theorem below.

## J    PROOF OF THEOREM 2.1

*Proof.* By the classical 1D B-spline theory De Boor (1978) and the fact that $\Phi_{l,i,j}$ as continuous functions can be uniformly bounded on a bounded domain, we know that there exist finite-grid B-spline functions $\Phi_{l,i,j}^G$ such that for any $0 \le m \le k$,

$$\|(\Phi_{l,i,j} \circ \boldsymbol{\Phi}_{l-1} \circ \boldsymbol{\Phi}_{l-2} \circ \cdots \circ \boldsymbol{\Phi}_1 \circ \boldsymbol{\Phi}_0)\mathbf{x} - (\Phi_{l,i,j}^G \circ \boldsymbol{\Phi}_{l-1} \circ \boldsymbol{\Phi}_{l-2} \circ \cdots \circ \boldsymbol{\Phi}_1 \circ \boldsymbol{\Phi}_0)\mathbf{x}\|_{C^m} \le C_0 G^{-k-1+m},$$

with a constant $C_0$ independent of $G$. We fix those B-spline approximations. Therefore we have that the residue $R_l$ defined via

$$R_l := (\boldsymbol{\Phi}_{L-1}^G \circ \cdots \circ \boldsymbol{\Phi}_{l+1}^G \circ \boldsymbol{\Phi}_l \circ \boldsymbol{\Phi}_{l-1} \circ \cdots \circ \boldsymbol{\Phi}_0)\mathbf{x} - (\boldsymbol{\Phi}_{L-1}^G \circ \cdots \circ \boldsymbol{\Phi}_{l+1}^G \circ \boldsymbol{\Phi}_l^G \circ \boldsymbol{\Phi}_{l-1} \circ \cdots \circ \boldsymbol{\Phi}_0)\mathbf{x}$$

satisfies

$$\|R_l\|_{C^m} \le C_1 G^{-k-1+m},$$

with another constant independent of $G$. Finally notice that

$$f - (\boldsymbol{\Phi}_{L-1}^G \circ \boldsymbol{\Phi}_{L-2}^G \circ \cdots \circ \boldsymbol{\Phi}_1^G \circ \boldsymbol{\Phi}_0^G)\mathbf{x} = R_{L-1} + R_{L-2} + \cdots + R_1 + R_0,$$

we know that (11) holds for another constant $C$ independent of $G$. $\qquad \square$

**Remark**: We can be more precise about the dependence of the constant $C$ in the theorem. Define the compositionally smooth function class $\mathcal{C}^{n,W,L,k}$ as the class of functions in the form of (10) such that the input dimension equals $n$, the width or $\max_{0 \le i \le L} n_i$ in the definition (3) equals $W \ge n$, depth equals $L$, smoothness equals $k$. Then $C$ only depends on $W, L, k$ and $\max \|\phi_{l,i,j}\|_{C^m}$.

| Paper | Idea | Scaling exponent $\alpha$ |
|---|---|---|
| Sharma & Kaplan Sharma & Kaplan (2020) | Intrinsic dimensionality | $(k+1)/d$ |
| Michaud et al. Michaud et al. (2023b) | maximum arity | $(k+1)/2$ |
| Poggio et al. Poggio et al. (2020) | compositional sparsity | $m/2$ |
| Ours | K-A representation | $k+1$ |

Table 8: Scaling exponents from different theories $\ell \propto N^{-\alpha}$. $\ell$: test RMSE loss, $N$: number of model parameters, $d$: input intrinsic dimension, $k$: order of piecewise polynomial, $m$: derivative order as in function class $W_m$.

## K  NEURAL SCALING LAWS

We remark that although the Kolmogorov-Arnold theorem Eq. (1) corresponds to a KAN representation with shape $[d, 2d+1, 1]$, its functions are not necessarily smooth. On the other hand, if we are able to identify a smooth representation (maybe at the cost of extra layers or making the KAN wider than the theory prescribes), then Theorem 2.1 indicates that we can beat the curse of dimensionality (COD). This should not come as a surprise since we can inherently learn the structure of the function and make our finite-sample KAN approximation interpretable.

**Neural scaling laws: comparison to other theories.** Neural scaling laws are the phenomenon where test loss decreases with more model parameters, i.e., $\ell \propto N^{-\alpha}$ where $\ell$ is test RMSE, $N$ is the number of parameters, and $\alpha$ is the scaling exponent. A larger $\alpha$ promises more improvement by simply scaling up the model. Different theories have been proposed to predict $\alpha$. Sharma & Kaplan Sharma & Kaplan (2020) suggest that $\alpha$ comes from data fitting on an input manifold of intrinsic dimensionality $d$. If the model function class is piecewise polynomials of order $k$ ($k = 1$ for ReLU), then the standard approximation theory implies $\alpha = (k+1)/d$ from the approximation theory. This bound suffers from the curse of dimensionality, so people have sought other bounds independent of $d$ by leveraging compositional structures. In particular, Michaud et al. Michaud et al. (2023b) considered computational graphs that only involve unary (e.g., squared, sine, exp) and binary (+ and ×) operations, finding $\alpha = (k+1)/d^* = (k+1)/2$, where $d^* = 2$ is the maximum arity. Poggio et al. Poggio et al. (2020) leveraged the idea of compositional sparsity and proved that given function class $W_m$ (function whose derivatives are continuous up to $m$-th order), one needs $N = O(\epsilon^{-\frac{2}{m}})$ number of parameters to achieve error $\epsilon$, which is equivalent to $\alpha = \frac{m}{2}$. Our approach, which assumes the existence of smooth Kolmogorov-Arnold representations, decomposes the high-dimensional function into several 1D functions, giving $\alpha = k+1$ (where $k$ is the piecewise polynomial order of the splines). We choose $k = 3$ cubic splines so $\alpha = 4$ which is the largest and best scaling exponent compared to other works. We will show in Section 4 toy datasets that this bound $\alpha = 4$ can in fact be achieved empirically with KANs, while previous work Michaud et al. (2023b) reported that MLPs have problems even saturating slower bounds (e.g., $\alpha = 1$) and plateau quickly. Of course, we can increase $k$ to match the smoothness of functions, but too high $k$ might be too oscillatory, leading to optimization issues.

**Comparison between KAT and UAT.** The power of fully-connected neural networks is justified by the universal approximation theorem (UAT), which states that given a function and error tolerance $\epsilon > 0$, a two-layer network with $k > N(\epsilon)$ neurons can approximate the function within error $\epsilon$. However, the UAT guarantees no bound for how $N(\epsilon)$ scales with $\epsilon$. Indeed, it suffers from the COD, and $N$ has been shown to grow exponentially with $d$ in some cases Lin et al. (2017). The difference between KAT and UAT is a consequence that KANs take advantage of the intrinsically low-dimensional representation of the function while MLPs do not. In KAT, we highlight quantifying the approximation error in the compositional space. In the literature, generalization error bounds, taking into account finite samples of training data, for a similar space have been studied for regression problems; see Horowitz & Mammen (2007); Kohler & Langer (2021), and also specifically for MLPs with ReLU activations Schmidt-Hieber (2020). On the other hand, for general function spaces like Sobolev or Besov spaces, the nonlinear $n$-widths theory DeVore et al. (1989; 1993);

---

[2]This is done by drawing B-spline coefficients $c_i \sim \mathcal{N}(0, \sigma^2)$ with a small $\sigma$, typically we set $\sigma = 0.1$.

[3]Other possibilities are: (a) the grid is learnable with gradient descent, e.g., Xu et al. (2015); (b) use normalization such that the input range is fixed. We tried (b) at first but its performance is inferior to our current approach.

Siegel (2024) indicates that we can never beat the curse of dimensionality, while MLPs with ReLU activations can achieve the tight rate Yarotsky (2017); Bartlett et al. (2019); Siegel (2023). This fact again motivates us to consider functions of compositional structure, the much "nicer" functions that we encounter in practice and in science, to overcome the COD. Compared with MLPs, we may use a smaller architecture in practice, since we learn general nonlinear activation functions; see also Schmidt-Hieber (2020) where the depth of the ReLU MLPs needs to reach at least $\log n$ to have the desired rate, where $n$ is the number of samples. Indeed, we will show that KANs are nicely aligned with symbolic functions while MLPs are not.

## L  DETAILS OF GRID EXTENSION

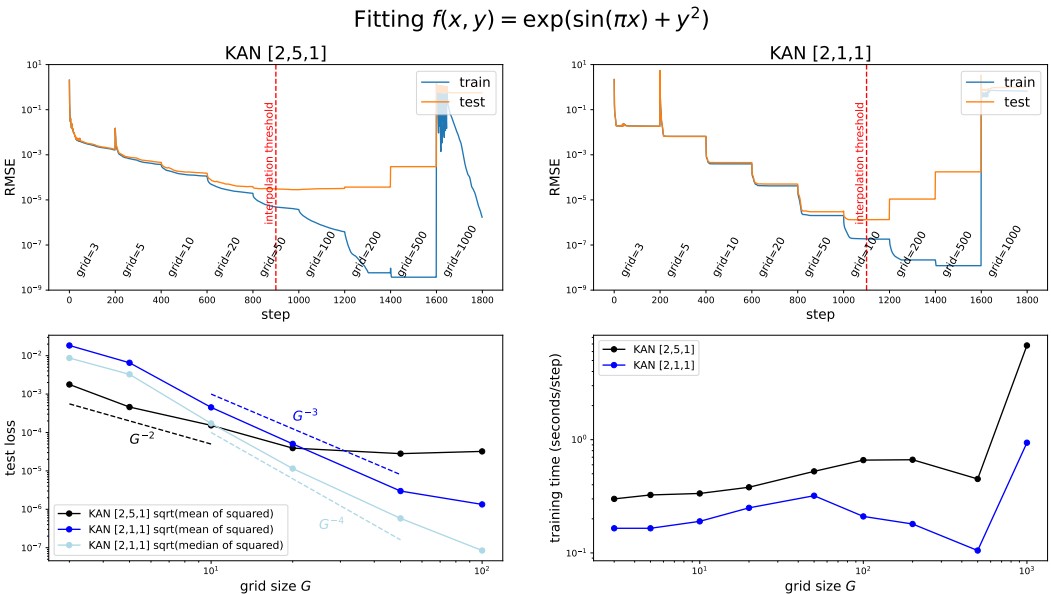

Figure 19: We can make KANs more accurate by grid extension (fine-graining spline grids). Top left (right): training dynamics of a $[2, 5, 1]$ ($[2, 1, 1]$) KAN. Both models display staircases in their loss curves, i.e., loss suddenly drops then plateaus after grid extension. Bottom left: test RMSE follows scaling laws against grid size $G$. Bottom right: training time scales favorably with grid size $G$.

We next describe how to perform grid extension (illustrated in Figure 18 right), which is basically fitting a new fine-grained spline to an old coarse-grained spline. Suppose we want to approximate a 1D function $f$ in a bounded region $[a, b]$ with B-splines of order $k$. A coarse-grained grid with $G_1$ intervals has grid points at $\{t_0 = a, t_1, t_2, \cdots, t_{G_1} = b\}$, which is augmented to $\{t_{-k}, \cdots, t_{-1}, t_0, \cdots, t_{G_1}, t_{G_1+1}, \cdots, t_{G_1+k}\}$. There are $G_1 + k$ B-spline basis functions, with the $i^{\text{th}}$ B-spline $B_i(x)$ being non-zero only on $[t_{-k+i}, t_{i+1}]$ ($i = 0, \cdots, G_1 + k - 1$). Then $f$ on the coarse grid is expressed in terms of linear combination of these B-splines basis functions $f_{\text{coarse}}(x) = \sum_{i=0}^{G_1+k-1} c_i B_i(x)$. Given a finer grid with $G_2$ intervals, $f$ on the fine grid is correspondingly $f_{\text{fine}}(x) = \sum_{j=0}^{G_2+k-1} c'_j B'_j(x)$. The parameters $c'_j$s can be initialized from the parameters $c_i$ by minimizing the distance between $f_{\text{fine}}(x)$ to $f_{\text{coarse}}(x)$ (over some distribution of $x$):

$$\{c'_j\} = \underset{\{c'_j\}}{\operatorname{argmin}} \ \underset{x \sim p(x)}{\mathbb{E}} \left( \sum_{j=0}^{G_2+k-1} c'_j B'_j(x) - \sum_{i=0}^{G_1+k-1} c_i B_i(x) \right)^2, \tag{18}$$

which can be implemented by the least squares algorithm. We perform grid extension for all splines in a KAN independently.

**Complexity of grid extension** Suppose we have batch size $B$, the number of evaluations to create the "supervised" dataset is $O(G_1 B)$. The least-square problem requires the number of operations

$O(G_2^2 B)$ since it is a singular-value decomposition and typically $B > G_2$. This analysis also applies to grid refinements, only by setting $G_2 = G_1$.

**Toy example: staircase-like loss curves.** We use a toy example $f(x, y) = \exp(\sin(\pi x) + y^2)$ to demonstrate the effect of grid extension. In Figure 19 (top left), we show the train and test RMSE for a $[2, 5, 1]$ KAN. The number of grid points starts as 3, increases to a higher value every 200 LBFGS steps, ending up with 1000 grid points. It is clear that every time fine graining happens, the training loss drops faster than before (except for the finest grid with 1000 points, where optimization ceases to work probably due to bad loss landscapes). However, the test losses first go down then go up, displaying a U-shape, due to the bias-variance tradeoff (underfitting vs. overfitting). We conjecture that the optimal test loss is achieved at the interpolation threshold when the number of parameters match the number of data points. Since our training samples are 1000 and the total parameters of a $[2, 5, 1]$ KAN is $15G$ ($G$ is the number of grid intervals), we expect the interpolation threshold to be $G = 1000/15 \approx 67$, which roughly agrees with our experimentally observed value $G \sim 50$.

**Small KANs generalize better.** Is this the best test performance we can achieve? Notice that the synthetic task can be represented exactly by a $[2, 1, 1]$ KAN, so we train a $[2, 1, 1]$ KAN and present the training dynamics in Figure 19 top right. Interestingly, it can achieve even lower test losses than the $[2, 5, 1]$ KAN, with clearer staircase structures and the interpolation threshold is delayed to a larger grid size as a result of fewer parameters. This highlights a subtlety of choosing KAN architectures. If we do not know the problem structure, how can we determine the minimal KAN shape? In Section 2.4, we will propose a method to auto-discover such minimal KAN architecture via regularization and pruning.

**Scaling laws: comparison with theory.** We are also interested in how the test loss decreases as the number of grid parameters increases. In Figure 19 (bottom left), a [2,1,1] KAN scales roughly as test RMSE $\propto G^{-3}$. However, according to the Theorem 2.1, we would expect test RMSE $\propto G^{-4}$. We found that the errors across samples are not uniform. This is probably attributed to boundary effects Michaud et al. (2023b). In fact, there are a few samples that have significantly larger errors than others, making the overall scaling slow down. If we plot the square root of the *median* (not *mean*) of the squared losses, we get a scaling closer to $G^{-4}$. Despite this suboptimality (probably due to optimization), KANs still have much better scaling laws than MLPs, for data fitting (Figure 5) and PDE solving (Figure 8). In addition, the training time scales favorably with the number of grid points $G$, shown in Figure 19 bottom right [4].

**External vs Internal degrees of freedom.** A new concept that KANs highlights is a distinction between external versus internal degrees of freedom (parameters). The computational graph of how nodes are connected represents external degrees of freedom ("dofs"), while the grid points inside an activation function are internal degrees of freedom. KANs benefit from the fact that they have both external dofs and internal dofs. External dofs (that MLPs also have but splines do not) are responsible for learning compositional structures of multiple variables. Internal dofs (that splines also have but MLPs do not) are responsible for learning univariate functions.

## M  Techniques for increasing interpretability

### M.1  Simplification techniques

**1. Sparsification.** For MLPs, L1 regularization of linear weights is used to favor sparsity. KANs can adapt this high-level idea, but need two modifications:

(1) There is no linear "weight" in KANs. Linear weights are replaced by learnable activation functions, so we should define the L1 norm of these activation functions.

(2) We find L1 to be insufficient for sparsification of KANs; instead an additional entropy regularization is necessary (see Appendix W for more details).

---

[4]When $G = 1000$, training becomes significantly slower, which is specific to the use of the LBFGS optimizer with line search. We conjecture that the loss landscape becomes bad for $G = 1000$, so line search with trying to find an optimal step size within maximal iterations without early stopping.

We define the L1 norm of an activation function $\phi$ to be its average magnitude over its $N_p$ inputs, i.e.,

$$|\phi|_1 \equiv \frac{1}{N_p} \sum_{s=1}^{N_p} \left| \phi(x^{(s)}) \right|. \tag{19}$$

Then for a KAN layer $\mathbf{\Phi}$ with $n_{\text{in}}$ inputs and $n_{\text{out}}$ outputs, we define the L1 norm of $\mathbf{\Phi}$ to be the sum of L1 norms of all activation functions, i.e.,

$$|\mathbf{\Phi}|_1 \equiv \sum_{i=1}^{n_{\text{in}}} \sum_{j=1}^{n_{\text{out}}} |\phi_{i,j}|_1. \tag{20}$$

In addition, we define the entropy of $\mathbf{\Phi}$ to be

$$S(\mathbf{\Phi}) \equiv - \sum_{i=1}^{n_{\text{in}}} \sum_{j=1}^{n_{\text{out}}} \frac{|\phi_{i,j}|_1}{|\mathbf{\Phi}|_1} \log \left( \frac{|\phi_{i,j}|_1}{|\mathbf{\Phi}|_1} \right). \tag{21}$$

The total training objective $\ell_{\text{total}}$ is the prediction loss $\ell_{\text{pred}}$ plus L1 and entropy regularization of all KAN layers:

$$\ell_{\text{total}} = \ell_{\text{pred}} + \lambda \left( \mu_1 \sum_{l=0}^{L-1} |\mathbf{\Phi}_l|_1 + \mu_2 \sum_{l=0}^{L-1} S(\mathbf{\Phi}_l) \right), \tag{22}$$

where $\mu_1, \mu_2$ are relative magnitudes usually set to $\mu_1 = \mu_2 = 1$, and $\lambda$ controls overall regularization magnitude.

**2. Visualization.** When we visualize a KAN, to get a sense of magnitudes, we set the transparency of an activation function $\phi_{l,i,j}$ proportional to $\tanh(\beta A_{l,i,j})$ where $\beta = 3$. Hence, functions with small magnitude appear faded out to allow us to focus on important ones.

**3. Pruning.** After training with sparsification penalty, we may also want to prune the network to a smaller subnetwork. We sparsify KANs on the node level (rather than on the edge level). For each node (say the $i^{\text{th}}$ neuron in the $l^{\text{th}}$ layer), we define its incoming and outgoing score as

$$I_{l,i} = \max_k(|\phi_{l-1,k,i}|_1), \qquad O_{l,i} = \max_j(|\phi_{l+1,j,i}|_1), \tag{23}$$

and consider a node to be important if both incoming and outgoing scores are greater than a threshold hyperparameter $\theta = 10^{-2}$ by default. All unimportant neurons are pruned.

**4. Symbolification.** In cases where we suspect that some activation functions are in fact symbolic (e.g., $\cos$ or $\log$), we provide an interface to set them to be a specified symbolic form, `fix_symbolic(l,i,j,f)` can set the $(l, i, j)$ activation to be $f$. However, we cannot simply set the activation function to be the exact symbolic formula, since its inputs and outputs may have shifts and scalings. So, we obtain preactivations $x$ and postactivations $y$ from samples, and fit affine parameters $(a, b, c, d)$ such that $y \approx cf(ax + b) + d$. The fitting is done by iterative grid search of $a, b$ and linear regression.

Besides these techniques, we provide additional tools that allow users to apply more fine-grained control to KANs, listed in Appendix V.

M.2   A TOY EXAMPLE: HOW HUMANS CAN INTERACT WITH KANS

Above we have proposed a number of simplification techniques for KANs. We can view these simplification choices as buttons one can click on. A user interacting with these buttons can decide which button is most promising to click next to make KANs more interpretable. We use an example below to showcase how a user could interact with a KAN to obtain maximally interpretable results.

Let us again consider the regression task

$$f(x, y) = \exp\left(\sin(\pi x) + y^2\right). \tag{24}$$

Given data points $(x_i, y_i, f_i)$, $i = 1, 2, \cdots, N_p$, a hypothetical user Alice is interested in figuring out the symbolic formula. The steps of Alice's interaction with the KANs are described below (illustrated in Figure 20):

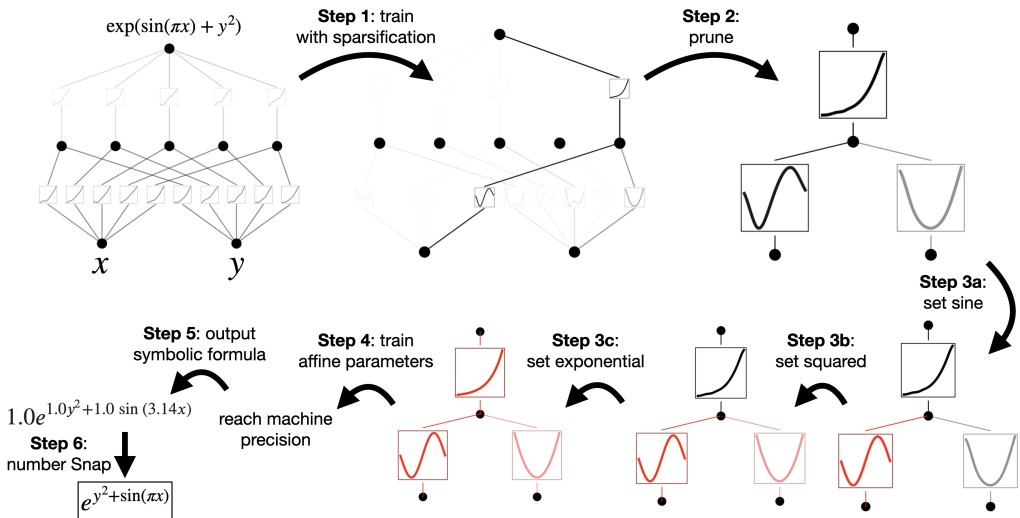

Figure 20: An example of how to do symbolic regression with KAN.

**Step 1: Training with sparsification.** Starting from a fully-connected $[2, 5, 1]$ KAN, training with sparsification regularization can make it quite sparse. 4 out of 5 neurons in the hidden layer appear useless, hence we want to prune them away.

**Step 2: Pruning.** Automatic pruning is seen to discard all hidden neurons except the last one, leaving a $[2, 1, 1]$ KAN. The activation functions appear to be known symbolic functions.

**Step 3: Setting symbolic functions.** Assuming that the user can correctly guess these symbolic formulas from staring at the KAN plot, they can set

$$
\begin{aligned}
&\texttt{fix\_symbolic(0,0,0,`sin')} \\
&\texttt{fix\_symbolic(0,1,0,`x\^{}2')} \\
&\texttt{fix\_symbolic(1,0,0,`exp').}
\end{aligned}
\tag{25}
$$

In case the user has no domain knowledge or no idea which symbolic functions these activation functions might be, we provide a function `suggest_symbolic` to suggest symbolic candidates.

**Step 4: Further training.** After symbolifying all the activation functions in the network, the only remaining parameters are the affine parameters. We continue training these affine parameters, and when we see the loss dropping to machine precision, we know that we have found the correct symbolic expression.

**Step 5: Output the symbolic formula.** `Sympy` is used to compute the symbolic formula of the output node. The user obtains $1.0e^{1.0y^2 + 1.0\sin(3.14x)}$, which is the true answer (we only displayed two decimals for $\pi$).

## N  ACCURACY: TOY SYMBOLIC DATASETS

We train these KANs by increasing grid points every 200 steps, in total covering $G = \{3, 5, 10, 20, 50, 100, 200, 500, 1000\}$. We train MLPs with different depths and widths as baselines. Both MLPs and KANs are trained with LBFGS for 1800 steps in total. We plot test RMSE as a function of the number of parameters for KANs and MLPs in Figure 5, showing that KANs have better scaling curves than MLPs, especially for the high-dimensional example. For comparison, we plot the lines predicted from our KAN theory as red dashed ($\alpha = k+1 = 4$), and the lines predicted from Sharma & Kaplan Sharma & Kaplan (2020) as black-dashed ($\alpha = (k+1)/d = 4/d$). KANs can almost saturate the steeper red lines, while MLPs struggle to converge even as fast as the slower black lines and plateau quickly. We also note that for the last example, the 2-Layer KAN $[4, 9, 1]$ behaves much worse than the 3-Layer KAN (shape $[4, 2, 2, 1]$). This highlights the greater expressive power of deeper KANs, which is the same for MLPs: deeper MLPs have more expressive power than shallower ones. Note that we have adopted the vanilla setup where both KANs and MLPs are

| Name | scipy.special API | Minimal KAN shape test RMSE $< 10^{-2}$ | Minimal KAN test RMSE | Best KAN shape | Best KAN test RMSE | MLP test RMSE |
|---|---|---|---|---|---|---|
| Jacobian elliptic functions | ellipj$(x,y)$ | [2,2,1] | $7.29 \times 10^{-3}$ | [2,3,2,1,1,1] | $\mathbf{1.33 \times 10^{-4}}$ | $6.48 \times 10^{-4}$ |
| Incomplete elliptic integral of the first kind | ellipkinc$(x,y)$ | [2,2,1,1] | $1.00 \times 10^{-3}$ | [2,2,1,1,1] | $\mathbf{1.24 \times 10^{-4}}$ | $5.52 \times 10^{-4}$ |
| Incomplete elliptic integral of the second kind | ellipeinc$(x,y)$ | [2,2,1,1] | $8.36 \times 10^{-5}$ | [2,2,1,1] | $\mathbf{8.26 \times 10^{-5}}$ | $3.04 \times 10^{-4}$ |
| Bessel function of the first kind | jv$(x,y)$ | [2,2,1] | $4.93 \times 10^{-3}$ | [2,3,1,1,1] | $\mathbf{1.64 \times 10^{-3}}$ | $5.52 \times 10^{-3}$ |
| Bessel function of the second kind | yv$(x,y)$ | [2,3,1] | $1.89 \times 10^{-3}$ | [2,2,2,1] | $\mathbf{1.49 \times 10^{-5}}$ | $3.45 \times 10^{-4}$ |
| Modified Bessel function of the second kind | kv$(x,y)$ | [2,1,1] | $4.89 \times 10^{-3}$ | [2,2,1] | $\mathbf{2.52 \times 10^{-5}}$ | $1.67 \times 10^{-4}$ |
| Modified Bessel function of the first kind | iv$(x,y)$ | [2,4,3,2,1,1] | $9.28 \times 10^{-3}$ | [2,4,3,2,1,1] | $\mathbf{9.28 \times 10^{-3}}$ | $1.07 \times 10^{-2}$ |
| Associated Legendre function ($m=0$) | lpmv$(0,x,y)$ | [2,2,1] | $5.25 \times 10^{-5}$ | [2,2,1] | $\mathbf{5.25 \times 10^{-5}}$ | $1.74 \times 10^{-2}$ |
| Associated Legendre function ($m=1$) | lpmv$(1,x,y)$ | [2,4,1] | $6.90 \times 10^{-4}$ | [2,4,1] | $\mathbf{6.90 \times 10^{-4}}$ | $1.50 \times 10^{-3}$ |
| Associated Legendre function ($m=2$) | lpmv$(2,x,y)$ | [2,2,1] | $4.88 \times 10^{-3}$ | [2,3,2,1] | $\mathbf{2.26 \times 10^{-4}}$ | $9.43 \times 10^{-4}$ |
| spherical harmonics ($m=0,n=1$) | sph_harm$(0,1,x,y)$ | [2,1,1] | $2.21 \times 10^{-7}$ | [2,1,1] | $\mathbf{2.21 \times 10^{-7}}$ | $1.25 \times 10^{-6}$ |
| spherical harmonics ($m=1,n=1$) | sph_harm$(1,1,x,y)$ | [2,2,1] | $7.86 \times 10^{-4}$ | [2,3,2,1] | $\mathbf{1.22 \times 10^{-4}}$ | $6.70 \times 10^{-4}$ |
| spherical harmonics ($m=0,n=2$) | sph_harm$(0,2,x,y)$ | [2,1,1] | $1.95 \times 10^{-7}$ | [2,1,1] | $\mathbf{1.95 \times 10^{-7}}$ | $2.85 \times 10^{-6}$ |
| spherical harmonics ($m=1,n=2$) | sph_harm$(1,2,x,y)$ | [2,2,1] | $4.70 \times 10^{-4}$ | [2,2,2,1] | $\mathbf{1.50 \times 10^{-5}}$ | $1.84 \times 10^{-3}$ |
| spherical harmonics ($m=2,n=2$) | sph_harm$(2,2,x,y)$ | [2,2,1] | $1.12 \times 10^{-3}$ | [2,2,3,2,1] | $\mathbf{9.45 \times 10^{-5}}$ | $6.21 \times 10^{-4}$ |

Table 9: Special functions

trained with LBFGS without advanced techniques, e.g., switching between Adam and LBFGS, or boosting Wang & Lai (2024). We leave the comparison of KANs and MLPs in advanced setups for future work.

## O    ACCURACY: SPECIAL FUNCTIONS

We choose MLPs with fixed width 5 or 100 and depths swept in $\{2, 3, 4, 5, 6\}$. We run KANs both with and without pruning. *KANs without pruning*: We fix the shape of KAN, whose width are set to 5 and depths are swept in $\{2,3,4,5,6\}$. *KAN with pruning*. We use the sparsification ($\lambda = 10^{-2}$ or $10^{-3}$) and pruning technique in Section M.1 to obtain a smaller KAN pruned from a fixed-shape KAN. Each KAN is initialized to have $G = 3$, trained with LBFGS, with increasing number of grid points every 200 steps to cover $G = \{3, 5, 10, 20, 50, 100, 200\}$. For each hyperparameter combination, we run 3 random seeds.

For each dataset and each model family (KANs or MLPs), we plot the Pareto frontier [5], in the (number of parameters, RMSE) plane, shown in Figure 6. KANs' performance is shown to be consistently better than MLPs, i.e., KANs can achieve lower training/test losses than MLPs, given the same number of parameters. Moreover, we report the (surprisingly compact) shapes of our auto-discovered KANs for special functions in Table 9. On one hand, it is interesting to interpret what these compact representations mean mathematically. On the other hand, these compact representations imply the possibility of breaking down a high-dimensional lookup table into several 1D lookup tables, which can potentially save a lot of memory, with the (almost negligible) overhead to perform a few additions at inference time.

## P    ACCURACY: FEYNMAN DATASETS

**Feynman datasets** Given the structure of the dataset, we may construct KANs by hand, but can KANs find more compact representations? The Feynman dataset Udrescu & Tegmark (2020); Udrescu et al. (2020), consisting of symbolic equations in physics, is a good testbed. We find that KAN shapes discovered by pruning are usually smaller than human-constructed KAN shapes, with comparable accuracy. Here we focus on a sample equation called the relativistic velocity addition formula $f(u, v) = (u + v)/(1 + uv)$. We can manually construct a 5-Layer KAN to compute this function, considering the resources required by two multiplications, one inversion and two additions. However, the auto-discovered KANs are only 2 layers deep! In hindsight, this is actually expected if we recall the rapidity trick in relativity: define the two "rapidities" $a \equiv \operatorname{arctanh} u$ and $b \equiv \operatorname{arctanh} v$. The relativistic composition of velocities are simple additions in rapidity space, i.e., $\frac{u+v}{1+uv} = \tanh(\operatorname{arctanh} u + \operatorname{arctanh} v)$, which can be realized by a two-layer KAN.

We compare four kinds of neural networks:

---

[5]Pareto frontier is defined as fits that are optimal in the sense of no other fit being both simpler and more accurate.

| Feynman Eq. | Original Formula | Dimensionless formula | Variables | Human-constructed KAN shape | Pruned KAN shape (smallest shape that achieves RMSE < $10^{-2}$) | Pruned KAN shape (lowest loss) | Human-constructed KAN loss (lowest test RMSE) | Pruned KAN loss (lowest test RMSE) | Unpruned KAN loss (lowest test RMSE) | MLP loss (lowest test RMSE) |
|---|---|---|---|---|---|---|---|---|---|---|
| I.6.2 | $\exp(-\frac{\theta^2}{2\sigma^2})/\sqrt{2\pi\sigma^2}$ | $\exp(-\frac{\theta^2}{2\sigma^2})/\sqrt{2\pi\sigma^2}$ | $\theta,\sigma$ | [2,2,1,1] | [2,2,1] | [2,2,1,1] | $7.66\times10^{-5}$ | $\mathbf{2.86\times10^{-5}}$ | $4.60\times10^{-5}$ | $1.45\times10^{-4}$ |
| I.6.2b | $\exp(-\frac{(\theta-\theta_1)^2}{2\sigma^2})/\sqrt{2\pi\sigma^2}$ | $\exp(-\frac{(\theta-\theta_1)^2}{2\sigma^2})/\sqrt{2\pi\sigma^2}$ | $\theta,\theta_1,\sigma$ | [3,2,2,1,1] | [3,4,1] | [3,2,2,1,1] | $1.22\times10^{-3}$ | $\mathbf{4.45\times10^{-4}}$ | $1.25\times10^{-3}$ | $7.40\times10^{-4}$ |
| I.9.18 | $\frac{Gm_1m_2}{(x_2-x_1)^2+(y_2-y_1)^2+(z_2-z_1)^2}$ | $\frac{a}{(b-1)^2+(c-d)^2+(e-f)^2}$ | $a,b,c,d,e,f$ | [6,4,2,1,1] | [6,4,1,1] | [6,4,1,1] | $\mathbf{1.48\times10^{-3}}$ | $8.62\times10^{-3}$ | $6.56\times10^{-3}$ | $1.59\times10^{-3}$ |
| I.12.11 | $q(E_f+Bv\sin\theta)$ | $1+a\sin\theta$ | $a,\theta$ | [2,2,2,1] | [2,2,1] | [2,2,1] | $2.07\times10^{-3}$ | $1.39\times10^{-3}$ | $9.13\times10^{-4}$ | $\mathbf{6.71\times10^{-4}}$ |
| I.13.12 | $Gm_1m_2(\frac{1}{r_2}-\frac{1}{r_1})$ | $a(\frac{1}{b}-1)$ | $a,b$ | [2,2,1] | [2,2,1] | [2,2,1] | $7.22\times10^{-3}$ | $4.81\times10^{-3}$ | $2.72\times10^{-3}$ | $\mathbf{1.42\times10^{-3}}$ |
| I.15.3x | $\frac{x-ut}{\sqrt{1-(\frac{u}{c})^2}}$ | $\frac{1-a}{\sqrt{1-b^2}}$ | $a,b$ | [2,2,1,1] | [2,1,1] | [2,2,1,1,1] | $7.35\times10^{-3}$ | $1.58\times10^{-3}$ | $1.14\times10^{-3}$ | $\mathbf{8.54\times10^{-4}}$ |
| I.16.6 | $\frac{u+v}{1+\frac{uv}{c^2}}$ | $\frac{a+b}{1+ab}$ | $a,b$ | [2,2,2,2,1] | [2,2,1] | [2,2,1] | $1.06\times10^{-3}$ | $1.19\times10^{-3}$ | $1.53\times10^{-3}$ | $\mathbf{6.20\times10^{-4}}$ |
| I.18.4 | $\frac{m_1r_1+m_2r_2}{m_1+m_2}$ | $\frac{1+ab}{1+a}$ | $a,b$ | [2,2,2,1,1] | [2,2,1] | [2,2,1] | $3.92\times10^{-4}$ | $\mathbf{1.50\times10^{-4}}$ | $1.32\times10^{-3}$ | $3.68\times10^{-4}$ |
| I.26.2 | $\arcsin(n\sin\theta_2)$ | $\arcsin(n\sin\theta_2)$ | $n,\theta_2$ | [2,2,2,1,1] | [2,2,1] | [2,2,2,1,1] | $1.22\times10^{-1}$ | $\mathbf{7.90\times10^{-1}}$ | $8.63\times10^{-4}$ | $1.24\times10^{-3}$ |
| I.27.6 | $\frac{1}{\frac{1}{d_1}+\frac{n}{d_2}}$ | $\frac{1}{1+ab}$ | $a,b$ | [2,2,1,1] | [2,1,1] | [2,1,1] | $2.22\times10^{-4}$ | $\mathbf{1.94\times10^{-4}}$ | $2.14\times10^{-4}$ | $2.46\times10^{-4}$ |
| I.29.16 | $\sqrt{x_1^2+x_2^2-2x_1x_2\cos(\theta_1-\theta_2)}$ | $\sqrt{1+a^2-2a\cos(\theta_1-\theta_2)}$ | $a,\theta_1,\theta_2$ | [3,2,2,3,2,1,1] | [3,2,2,1] | [3,2,3,1] | $2.36\times10^{-1}$ | $3.99\times10^{-3}$ | $\mathbf{3.20\times10^{-3}}$ | $4.64\times10^{-3}$ |
| I.30.3 | $I_{*,0}\frac{\sin^2(\frac{n\theta}{2})}{\sin^2(\frac{\theta}{2})}$ | $\frac{\sin^2(\frac{n\theta}{2})}{\sin^2(\frac{\theta}{2})}$ | $n,\theta$ | [2,3,2,2,1,1] | [2,4,3,1] | [2,3,2,3,1,1] | $3.85\times10^{-1}$ | $\mathbf{1.03\times10^{-3}}$ | $1.11\times10^{-2}$ | $1.50\times10^{-2}$ |
| I.30.5 | $\arcsin(\frac{\lambda}{nd})$ | $\arcsin(\frac{a}{n})$ | $a,n$ | [2,1,1] | [2,1,1] | [2,1,1,1,1,1] | $2.23\times10^{-4}$ | $\mathbf{3.49\times10^{-5}}$ | $6.92\times10^{-5}$ | $9.45\times10^{-5}$ |
| I.37.4 | $I_*=I_1+I_2+2\sqrt{I_1I_2}\cos\delta$ | $1+a+2\sqrt{a}\cos\delta$ | $a,\delta$ | [2,3,2,1] | [2,2,1] | [2,2,1] | $7.57\times10^{-5}$ | $\mathbf{4.91\times10^{-6}}$ | $3.41\times10^{-4}$ | $5.67\times10^{-4}$ |
| I.40.1 | $n_0\exp(-\frac{mgx}{k_BT})$ | $n_0e^{-a}$ | $n_0,a$ | [2,1,1] | [2,2,1] | [2,2,1,1,1,2,1] | $3.45\times10^{-3}$ | $5.01\times10^{-4}$ | $\mathbf{3.12\times10^{-4}}$ | $3.99\times10^{-4}$ |
| I.44.4 | $nk_BT\ln(\frac{V_2}{V_1})$ | $n\ln a$ | $n,a$ | [2,2,1] | [2,2,1] | [2,2,1] | $\mathbf{2.30\times10^{-5}}$ | $2.43\times10^{-5}$ | $1.10\times10^{-4}$ | $3.99\times10^{-4}$ |
| I.50.26 | $x_1(\cos(\omega t)+\alpha\cos^2(\omega t))$ | $\cos a+\alpha\cos^2 a$ | $a,\alpha$ | [2,2,3,1] | [2,3,1] | [2,3,2,1] | $\mathbf{1.52\times10^{-4}}$ | $5.82\times10^{-4}$ | $4.90\times10^{-4}$ | $1.53\times10^{-3}$ |
| II.2.42 | $\frac{k(T_2-T_1)A}{d}$ | $(a-1)b$ | $a,b$ | [2,2,1] | [2,2,1] | [2,2,2,1] | $8.54\times10^{-4}$ | $7.22\times10^{-4}$ | $1.22\times10^{-3}$ | $\mathbf{1.81\times10^{-4}}$ |
| II.6.15a | $\frac{3}{4\pi}\frac{p_d z}{\epsilon r^5}\sqrt{x^2+y^2}$ | $\frac{1}{4\pi}c\sqrt{a^2+b^2}$ | $a,b,c$ | [3,2,2,2,1] | [3,2,1,1] | [3,2,1,1] | $2.61\times10^{-3}$ | $3.28\times10^{-3}$ | $1.35\times10^{-3}$ | $\mathbf{5.92\times10^{-4}}$ |
| II.11.7 | $n_0(1+\frac{p_dE_f\cos\theta}{k_BT})$ | $n_0(1+a\cos\theta)$ | $n_0,a,\theta$ | [3,3,3,2,2,1] | [3,3,1,1] | [3,3,1,1] | $7.10\times10^{-3}$ | $8.52\times10^{-3}$ | $5.03\times10^{-3}$ | $\mathbf{5.92\times10^{-4}}$ |
| II.11.27 | $\frac{n\alpha}{1-\frac{n\alpha}{3}}\epsilon E_f$ | $\frac{n\alpha}{1-\frac{n\alpha}{3}}$ | $n,\alpha$ | [2,2,1,2,1] | [2,1,1] | [2,2,1] | $2.67\times10^{-5}$ | $4.40\times10^{-5}$ | $\mathbf{1.43\times10^{-5}}$ | $7.18\times10^{-5}$ |
| II.35.18 | $\frac{n_0}{\exp(\frac{\mu_mB}{k_BT})+\exp(-\frac{\mu_mB}{k_BT})}$ | $\frac{n_0}{\exp(a)+\exp(-a)}$ | $n_0,a$ | [2,1,1] | [2,1,1] | [2,1,1,1] | $4.13\times10^{-4}$ | $1.58\times10^{-4}$ | $\mathbf{7.71\times10^{-5}}$ | $7.92\times10^{-5}$ |
| II.36.38 | $\frac{\mu_mB}{k_BT}+\frac{\mu_m\alpha M}{\epsilon c^2k_BT}$ | $a+\alpha b$ | $a,\alpha,b$ | [3,3,1] | [3,2,1] | [3,2,1] | $2.85\times10^{-3}$ | $\mathbf{1.15\times10^{-3}}$ | $3.03\times10^{-3}$ | $2.15\times10^{-3}$ |
| II.38.3 | $\frac{YAx}{d}$ | $\frac{a}{b}$ | $a,b$ | [2,1,1] | [2,1,1] | [2,2,1,1,1,1] | $1.47\times10^{-4}$ | $\mathbf{8.78\times10^{-5}}$ | $6.43\times10^{-4}$ | $5.26\times10^{-4}$ |
| III.9.52 | $\frac{p_dE_f}{\hbar}\frac{\sin^2((\omega-\omega_0)t/2)}{((\omega-\omega_0)t/2)^2}$ | $a\frac{\sin^2(\frac{b-c}{2})}{(\frac{b-c}{2})^2}$ | $a,b,c$ | [3,2,3,1,1] | [3,3,2,1] | [3,3,2,1,1,1] | $4.43\times10^{-2}$ | $3.90\times10^{-3}$ | $2.11\times10^{-2}$ | $\mathbf{9.07\times10^{-4}}$ |
| III.10.19 | $\mu_m\sqrt{B_x^2+B_y^2+B_z^2}$ | $\sqrt{1+a^2+b^2}$ | $a,b$ | [2,1,1] | [2,1,1] | [2,1,2,1] | $2.54\times10^{-3}$ | $1.18\times10^{-3}$ | $8.16\times10^{-4}$ | $\mathbf{1.67\times10^{-4}}$ |
| III.17.37 | $\beta(1+\alpha\cos\theta)$ | $\beta(1+\alpha\cos\theta)$ | $\alpha,\beta,\theta$ | [3,3,3,2,2,1] | [3,3,1] | [3,3,1] | $1.10\times10^{-3}$ | $5.03\times10^{-4}$ | $\mathbf{4.12\times10^{-4}}$ | $6.80\times10^{-4}$ |

Table 10: Feynman dataset

(1) Human-constructued KAN. Given a symbolic formula, we rewrite it in Kolmogorov-Arnold representations. For example, to multiply two numbers $x$ and $y$, we can use the identity $xy = \frac{(x+y)^2}{4} - \frac{(x-y)^2}{4}$, which corresponds to a $[2, 2, 1]$ KAN. The constructued shapes are listed in the "Human-constructed KAN shape" in Table 10.

(2) KANs without pruning. We fix the KAN shape to width 5 and depths are swept over $\{2,3,4,5,6\}$.

(3) KAN with pruning. We use the sparsification ($\lambda = 10^{-2}$ or $10^{-3}$) and the pruning technique from Section M.1 to obtain a smaller KAN from a fixed-shape KAN from (2).

(4) MLPs with fixed width 20, depths swept in $\{2, 3, 4, 5, 6\}$, and activations chosen from $\{\text{Tanh}, \text{ReLU}, \text{SiLU}\}$.

Each KAN is initialized to have $G = 3$, trained with LBFGS, with an increasing number of grid points every 200 steps to cover $G = \{3, 5, 10, 20, 50, 100, 200\}$. For each hyperparameter combination, we try 3 random seeds. For each dataset (equation) and each method, we report the results of the best model (minimal KAN shape, or lowest test loss) over random seeds and depths in Table 10. We find that MLPs and KANs behave comparably on average. We conjecture that the Feynman datasets are too simple to let KANs make further improvements, in the sense that variable dependence is usually smooth or monotonic, which is in contrast to the complexity of special functions which often demonstrate oscillatory behavior.

We report the pruned KAN shape in two columns of Table 10; one column is for the minimal pruned KAN shape that can achieve reasonable loss (i.e., test RMSE smaller than $10^{-2}$); the other column is for the pruned KAN that achieves lowest test loss. It is interesting to observe that auto-discovered KAN shapes (for both minimal and best) are usually smaller than our human constructions. This means that KA representations can be more efficient than we imagine. At the same time, this may make interpretability subtle because information is being squashed into a smaller space than what we are comfortable with.

## Q  "CONTINUAL LEARNING" OF A 1D TOY FUNCTION?

We show that KANs have local plasticity and can avoid catastrophic forgetting by leveraging the locality of splines, for 1D functions. The idea is simple: since spline bases are local, a sample will only affect a few nearby spline coefficients, leaving far-away coefficients intact (which is desirable

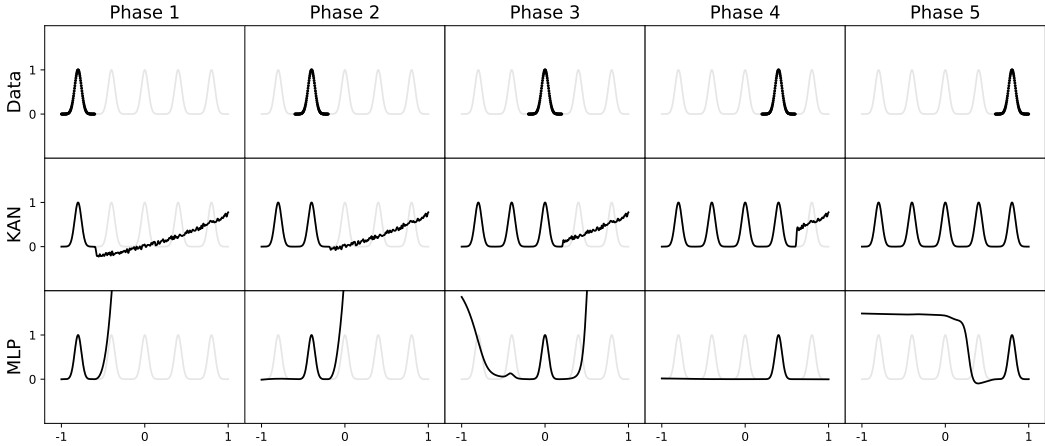

Figure 21: A toy continual learning problem. The dataset is a 1D regression task with 5 Gaussian peaks (top row). Data around each peak is presented sequentially (instead of all at once) to KANs and MLPs. KANs (middle row) can perfectly avoid catastrophic forgetting, while MLPs (bottom row) display severe catastrophic forgetting.

since far-away regions may have already stored information that we want to preserve). By contrast, since MLPs usually use global activations, e.g., ReLU/Tanh/SiLU etc., any local change may propagate uncontrollably to regions far away, destroying the information being stored there.

We use a toy example to validate this intuition. The 1D regression task is composed of 5 Gaussian peaks. Data around each peak is presented sequentially (instead of all at once) to KANs and MLPs, as shown in Figure 21 top row. KAN and MLP predictions after each training phase are shown in the middle and bottom rows. As expected, KAN only remodels regions where data is present on in the current phase, leaving previous regions unchanged. By contrast, MLPs remodels the whole region after seeing new data samples, leading to catastrophic forgetting. We want to mention that this toy example is somewhat trivial and is attributed to local activation functions rather than the KAN architecture. We simply feel this is a cute example to share in case anyone is inspired by it. However, this should not be interpreted as solving the continual learning problem. Indeed, when we try a deeper KAN [1,5,5,1], the continual learning feature is partially lost, depending on grid sizes (see Figure 22).

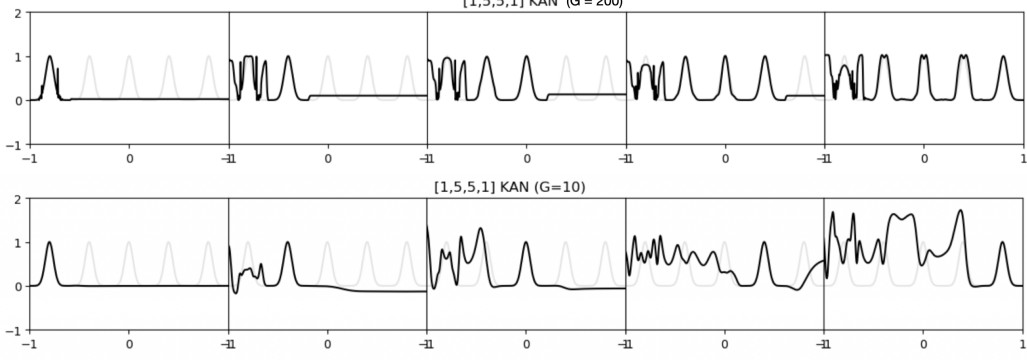

Figure 22: When KANs become deep, the continual learning ability is partially lost. Top: grid size 200; bottom: grid size 10.

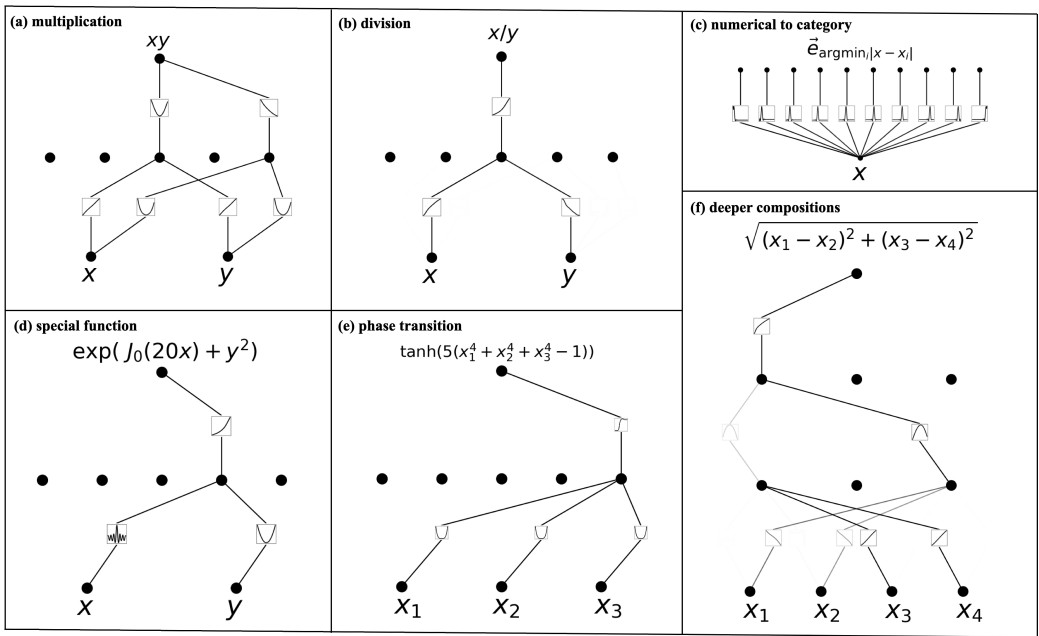

Figure 23: KANs are interepretable for simple symbolic tasks.

## R  INTERPRETABILITY: SUPERVISED TOY DATASETS

We tried KANs for 6 symbolic tasks, shown in Figure 23. (1) Multiplication $f(x, y) = xy$. A $[2, 5, 1]$ KAN is pruned to a $[2, 2, 1]$ KAN. The learned activation functions are linear and quadratic. From the computation graph, we see that the way it computes $xy$ is leveraging $2xy = (x+y)^2 - (x^2+y^2)$. (2) Division of positive numbers $f(x, y) = x/y$. A $[2, 5, 1]$ KAN is pruned to a $[2, 1, 1]$ KAN. The learned activation functions are logarithmic and exponential functions, and the KAN is computing $x/y$ by leveraging the identity $x/y = \exp(\log x - \log y)$. (3) Numerical to categorical. The task is to convert a real number in $[0, 1]$ to its first decimal digit (as one hots), e.g., $0.0618 \rightarrow [1, 0, 0, 0, 0, \cdots]$, $0.314 \rightarrow [0, 0, 0, 1, 0, \cdots]$. Notice that activation functions are learned to be spikes located around the corresponding decimal digits. (4) Special function $f(x, y) = \exp(J_0(20x) + y^2)$. One limitation of symbolic regression is that it will never find the correct formula of a special function if the special function is not provided as prior knowledge. KANs can learn special functions – the highly wiggly Bessel function $J_0(20x)$ is learned (numerically) by KAN. (5) Phase transition $f(x_1, x_2, x_3) = \tanh(5(x_1^4 + x_2^4 + x_3^4 - 1))$. Phase transitions are of great interest in physics, so we want KANs to be able to detect phase transitions and to identify the correct order parameters. We use the tanh function to simulate the phase transition behavior, and the order parameter is the combination of the quartic terms of $x_1, x_2, x_3$. Both the quartic dependence and tanh dependence emerge after KAN training. This is a simplified case of a localization phase transition discussed in Section U. (6) Deeper compositions $f(x_1, x_2, x_3, x_4) = \sqrt{(x_1 - x_2)^2 + (x_3 - x_4)^2}$. To compute this, we would need the identity function, squared function, and square root, which requires at least a three-layer KAN. Indeed, we find that a $[4, 3, 3, 1]$ KAN can be auto-pruned to a $[4, 2, 1, 1]$ KAN, which exactly corresponds to the computation graph we would expect.

## S  INTERPRETABILITY: UNSUPERVISED TOY DATASETS

Given a set of variables $(x_1, x_2, \cdots, x_d)$, we want to discover a structural relationship between the variables. Specifically, we want to find a non-zero $f$ such that $f(x_1, x_2, \cdots, x_d) \approx 0$.

**Unsupervised toy dataset** Given a set of variables $(x_1, x_2, \cdots, x_d)$, we want to discover a structural relationship between the variables. Specifically, we want to find a non-zero $f$ such that $f(x_1, x_2, \cdots, x_d) \approx 0$. Via contrastive learning formulation, we are able to turn this unsupervised learning problem into supervised learning (details in Appendix S).

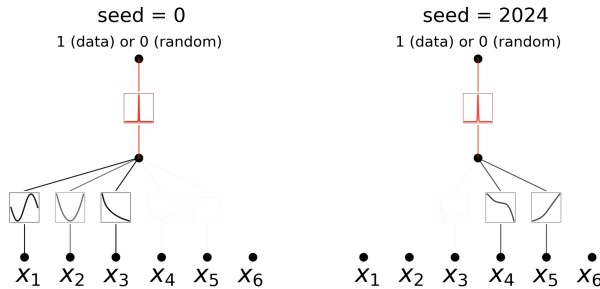

Figure 24: Unsupervised learning of a toy task. KANs can identify groups of dependent variables, i.e., $(x_1, x_2, x_3)$ and $(x_4, x_5)$ in this case.

We demonstrate that the unsupervised paradigm works for a synthetic 6D dataset, where $(x_1, x_2, x_3)$ are dependent variables such that $x_3 = \exp(\sin(x_1) + x_2^2)$; $(x_4, x_5)$ are dependent variables with $x_5 = x_4^3$; $x_6$ is independent of the other variables. In Figure 24, we show that for seed = 0, KAN reveals the functional dependence among $x_1, x_2$, and $x_3$; for another seed = 2024, KAN reveals the functional dependence between $x_4$ and $x_5$. Our preliminary results rely on randomness (different seeds) to discover different relations; in the future we would like to investigate a more systematic and more controlled way to discover a complete set of relations. Even so, our tool in its current status can provide insights for scientific tasks. We present our results with the knot dataset in Appendix T.

We tackle the unsupervised learning problem by turning it into a supervised learning problem on all of the $d$ features, without requiring the choice of a splitting. The essential idea is to learn a function $f(x_1, \ldots, x_d) = 0$ such that $f$ is not the 0-function. To do this, similar to contrastive learning, we define positive samples and negative samples: positive samples are feature vectors of real data. Negative samples are constructed by feature corruption. To ensure that the overall feature distribution for each topological invariant stays the same, we perform feature corruption by random permutation of each feature across the entire training set. Now we want to train a network $g$ such that $g(\mathbf{x}_{\text{real}}) = 1$ and $g(\mathbf{x}_{\text{fake}}) = 0$ which turns the problem into a supervised problem. However, remember that we originally want $f(\mathbf{x}_{\text{real}}) = 0$ and $f(\mathbf{x}_{\text{fake}}) \neq 0$. We can achieve this by having $g = \sigma \circ f$ where $\sigma(x) = \exp(-\frac{x^2}{2w^2})$ is a Gaussian function with a small width $w$, which can be conveniently realized by a KAN with shape $[..., 1, 1]$ whose last activation is set to be the Gaussian function $\sigma$ and all previous layers form $f$. Except for the modifications mentioned above, everything else is the same for supervised training.

## T    INTERPRETABILITY: KNOT THEORY

Knot theory is a subject in low-dimensional topology that sheds light on topological aspects of three-manifolds and four-manifolds and has a variety of applications, including in biology and topological quantum computing. Mathematically, a knot $K$ is an embedding of $S^1$ into $S^3$. Two knots $K$ and $K'$ are topologically equivalent if one can be deformed into the other via deformation of the ambient space $S^3$, in which case we write $[K] = [K']$. Some knots are topologically trivial, meaning that they can be smoothly deformed to a standard circle. Knots have a variety of deformation-invariant features $f$ called topological invariants, which may be used to show that two knots are topologically inequivalent, $[K] \neq [K']$ if $f(K) \neq f(K')$. In some cases the topological invariants are geometric in nature. For instance, a hyperbolic knot $K$ has a knot complement $S^3 \setminus K$ that admits a canonical hyperbolic metric $g$ such that $\text{vol}_g(K)$ is a topological invariant known as the hyperbolic volume. Other topological invariants are algebraic in nature, such as the Jones polynomial.

Given the fundamental nature of knots in mathematics and the importance of its applications, it is interesting to study whether ML can lead to new results. For instance, in Gukov et al. (2023) reinforcement learning was utilized to establish ribbonness of certain knots, which ruled out many potential counterexamples to the smooth 4d Poincaré conjecture.

Our results have one subtle difference from results in Davies et al. (2021): they find that signature is mostly dependent on $\mu_i$, while we find that signature is mostly dependent on $\mu_r$. This difference could be due to subtle algorithmic choices, but has led us to carry out the following experiments: (a)

| Id | Formula | Discovered by | test acc | $r^2$ with Signature | $r^2$ with DM formula |
|---|---|---|---|---|---|
| A | $\frac{\lambda\mu_r}{(\mu_r^2+\mu_i^2)}$ | Human (DM) | 83.1% | 0.946 | 1 |
| B | $-0.02\sin(4.98\mu_i+0.85)+0.08\|4.02\mu_r+6.28\|-0.52-0.04e^{-0.88(1-0.45\lambda)^2}$ | [3, 1] KAN | 62.6% | 0.837 | 0.897 |
| C | $0.17\tan(-1.51+0.1e^{-1.43(1-0.4\mu_i)^2}+0.09e^{-0.06(1-0.21\lambda)^2}+1.32e^{-3.18(1-0.43\mu_r)^2})$ | [3, 1, 1] KAN | 71.9% | 0.871 | 0.934 |
| D | $-0.09 + 1.04\exp(-9.59(-0.62\sin(0.61\mu_r + 7.26)) - 0.32\tan(0.03\lambda - 6.59) + 1 - 0.11e^{-1.77(0.31-\mu_i)^2)^2} - 1.09e^{-7.6(0.65(1-0.01\lambda)^3} + 0.27\text{atan}(0.53\mu_i - 0.6) + 0.09 + \exp(-2.58(1 - 0.36\mu_r)^2))$ | [3, 2, 1] KAN | 84.0% | 0.947 | 0.997 |
| E | $\frac{4.76\lambda\mu_r}{3.09\mu_i+6.05\mu_r^2+3.54\mu_i^2}$ | [3,2,1] KAN + Pade approx | 82.8% | 0.946 | 0.997 |
| F | $\frac{2.94-2.92(1-0.10\mu_r)^2}{0.32(0.18-\mu_r)^2+5.36(1-0.04\lambda)^2+0.50}$ | [3, 1] KAN/[3, 1] KAN | 77.8% | 0.925 | 0.977 |

Table 11: Symbolic formulas of signature as a function of meridinal translation $\mu$ (real $\mu_r$, imag $\mu_i$) and longitudinal translation $\lambda$. In Davies et al. (2021), formula A was discovered by human scientists inspired by neural network attribution results. Formulas B-F are auto-discovered by KANs. KANs can trade-off between simplicity and accuracy (B, C, D). By adding more inductive biases, KAN is able to discover formula E which is not too dissimilar from formula A. KANs also discovered a formula F which only involves two variables ($\mu_r$ and $\lambda$) instead of all three variables, with little sacrifice in accuracy.

ablation studies. We show that $\mu_r$ contributes more to accuracy than $\mu_i$ (see Figure 4): for example, $\mu_r$ alone can achieve 65.0% accuracy, while $\mu_i$ alone can only achieve 43.8% accuracy. (b) We find a symbolic formula (in Table 11) which only involves $\mu_r$ and $\lambda$, but can achieve 77.8% test accuracy.

To investigate (2), i.e., obtain the symbolic form of $\sigma$, we formulate the problem as a regression task. Using auto-symbolic regression introduced in Section M.1, we can convert a trained KAN into symbolic formulas. We train KANs with shapes [3, 1], [3, 1, 1], [3, 2, 1], whose corresponding symbolic formulas are displayed in Table 11 B-D. It is clear that by having a larger KAN, both accuracy and complexity increase. So KANs provide not just a single symbolic formula, but a whole Pareto frontier of formulas, trading off simplicity and accuracy. However, KANs need additional inductive biases to further simplify these equations to rediscover the formula from Davies et al. (2021) (Table 11 A). We have tested two scenarios: (1) in the first scenario, we assume the ground truth formula has a multi-variate Pade representation (division of two multi-variate Taylor series). We first train [3, 2, 1] and then fit it to a Pade representation. We can obtain Formula E in Table 11, which bears similarity with DeepMind's formula. (2) We hypothesize that the division is not very interpretable for KANs, so we train two KANs (one for the numerator and the other for the denominator) and divide them manually. Surprisingly, we end up with the formula F (in Table 11) which only involves $\mu_r$ and $\lambda$, although $\mu_i$ is also provided but ignored by KANs.

**unsupervised learning** Knot data are positive samples, and we randomly shuffle features to obtain negative samples. An [18, 1, 1] KAN is trained to classify whether a given feature vector belongs to a positive sample (1) or a negative sample (0). We manually set the second layer activation to be the Gaussian function with a peak one centered at zero, so positive samples will have activations at (around) zero, implicitly giving a relation among knot invariants $\sum_{i=1}^{18} g_i(x_i) = 0$ where $x_i$ stands for a feature (invariant), and $g_i$ is the corresponding activation function which can be readily read off from KAN diagrams. We train the KANs with $\lambda = \{10^{-2}, 10^{-3}\}$ to favor sparse combination of inputs, and seed $= \{0, 1, \cdots, 99\}$.

## U  INTERPRETABILITY: ANDERSON LOCALIZATION

**Application to Physics: Anderson localization** Anderson localization is the fundamental phenomenon in which disorder in a quantum system leads to the localization of electronic wave functions, causing all transport to be ceased Anderson (1958). More background information is available in Appendix U. Here, we apply KANs to numerical data generated from quasiperiodic tight-binding models to extract their mobility edges (phase transition boundaries), including generalized Aubry-

| System | Origin | Mobility Edge Formula | Accuracy |
|---|---|---|---|
| GAAM | Theory | $\alpha E + 2\lambda - 2 = 0$ | 99.2% |
| | KAN auto | $\cancel{1.52E^2} + 21.06\alpha E + \cancel{0.66E} + 3.55\cancel{\alpha^2} + 0.91\cancel{\alpha} + 45.13\lambda - 54.45 = 0$ | 99.0% |
| MAAM | Theory | $E + \exp(p) - \lambda\cosh p = 0$ | 98.6% |
| | KAN auto | $13.99\sin(0.28\sin(0.87\lambda + 2.22) - 0.84\arctan(0.58E - 0.26) + 0.85\arctan(0.94p + 0.13) - 8.14) - 16.74 + 43.08\exp(-0.93(0.06(0.13-p)^2 - 0.27\tanh(0.65E + 0.25) + 0.63\arctan(0.54\lambda - 0.62) + 1)^2) = 0$ | 97.1% |
| | KAN man (step 2) + auto | $4.19(0.28\sin(0.97\lambda + 2.17) - 0.77\arctan(0.83E - 0.19) + \arctan(0.97p + 0.15) - 0.35)^2 - 28.93 + 39.27\exp(-0.6(0.28\cosh^2(0.49p - 0.16) - 0.34\arctan(0.65E + 0.51) + 0.83\arctan(0.54\lambda - 0.62) + 1)^2) = 0$ | 97.7% |
| | KAN man (step 3) + auto | $-4.63E - 10.25(-0.94\sin(0.97\lambda - 6.81) + \tanh(0.8p - 0.45) + 0.09)^2 + 11.78\sin(0.76p - 1.41) + 22.49\arctan(1.08\lambda - 1.32) + 31.72 = 0$ | 97.7% |
| | KAN man (step 4A) | $6.92E - 6.23(-0.92\lambda - 1)^2 + 2572.45(-0.05\lambda + 0.95\cosh(0.11p + 0.4) - 1)^2 - 12.96\cosh^2(0.53p + 0.16) + 19.89 = 0$ | 96.6% |
| | KAN man (step 4B) | $7.25E - 8.81(-0.83\lambda - 1)^2 - 4.08(-p - 0.04)^2 + 12.71(-0.71\lambda + (0.3p + 1)^2 - 0.86)^2 + 10.29 = 0$ | 95.4% |

Table 12: Symbolic formulas for two systems GAAM and MAAM, ground truth ones and KAN-discovered ones.

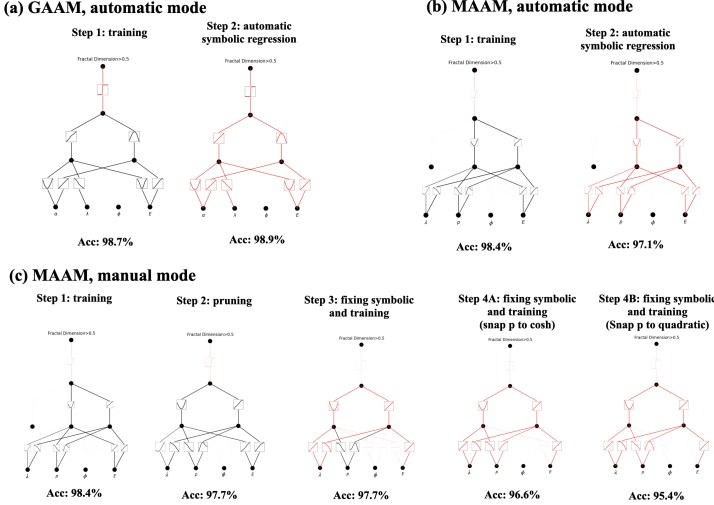

Figure 25: Human-KAN collaboration to discover mobility edges of GAAM and MAAM. The human user can choose to be lazy (using the auto mode) or more involved (using the manual mode).

André model (GAAM) Ganeshan et al. (2015) and the modified Aubry-André model (MAAM) Biddle & Sarma (2010), leaving results on a simpler tutorial case, the Mosaic model (MM) Wang et al. (2020), to Appendix U.

We highlight how users (scientists) can interact with KANs to get more interpretable results (in Figure 25). For the simpler GAAM case where the mobility edge is a quadratic function, the user can choose to be lazy and let KANs automatically do everything all the way through. KANs will be able to output the correct formula with some negligible error terms (shown in Table 12). However, for the more complex MAAM case, the fully automated mode find a too complicated formula. A user can choose to interact with KANs by fixing some activation to be known symbolic formulas and do further training. In the end, the user can obtain a family of symbolic formulas (instead of just one) that trade off between accuracy and simplicity.

Anderson localization is the fundamental phenomenon in which disorder in a quantum system leads to the localization of electronic wave functions, causing all transport to be ceased Anderson (1958). In one and two dimensions, scaling arguments show that all electronic eigenstates are exponen-

tially localized for an infinitesimal amount of random disorder Thouless (1972); Abrahams et al. (1979). In contrast, in three dimensions, a critical energy forms a phase boundary that separates the extended states from the localized states, known as a mobility edge. The understanding of these mobility edges is crucial for explaining various fundamental phenomena such as the metal-insulator transition in solids Lagendijk et al. (2009), as well as localization effects of light in photonic devices Segev et al. (2013); Vardeny et al. (2013); John (1987); Lahini et al. (2009); Vaidya et al. (2023). It is therefore necessary to develop microscopic models that exhibit mobility edges to enable detailed investigations. Developing such models is often more practical in lower dimensions, where introducing quasiperiodicity instead of random disorder can also result in mobility edges that separate localized and extended phases. Furthermore, experimental realizations of analytical mobility edges can help resolve the debate on localization in interacting systems De Roeck et al. (2016); Li et al. (2015). Indeed, several recent studies have focused on identifying such models and deriving exact analytic expressions for their mobility edges An et al. (2021); Biddle & Sarma (2010); Duthie et al. (2021); Ganeshan et al. (2015); Wang et al. (2020; 2021); Zhou et al. (2023).

Here, we apply KANs to numerical data generated from quasiperiodic tight-binding models to extract their mobility edges. In particular, we examine three classes of models: the Mosaic model (MM) Wang et al. (2020), the generalized Aubry-André model (GAAM) Ganeshan et al. (2015) and the modified Aubry-André model (MAAM) Biddle & Sarma (2010). For the MM, we testify KAN's ability to accurately extract mobility edge as a 1D function of energy. For the GAAM, we find that the formula obtained from a KAN closely matches the ground truth. For the more complicated MAAM, we demonstrate yet another example of the symbolic interpretability of this framework. A user can simplify the complex expression obtained from KANs (and corresponding symbolic formulas) by means of a "collaboration" where the human generates hypotheses to obtain a better match (e.g., making an assumption of the form of certain activation function), after which KANs can carry out quick hypotheses testing.

To quantify the localization of states in these models, the inverse participation ratio (IPR) is commonly used. The IPR for the $k^{th}$ eigenstate, $\psi^{(k)}$, is given by

$$\text{IPR}_k = \frac{\sum_n |\psi_n^{(k)}|^4}{\left(\sum_n |\psi_n^{(k)}|^2\right)^2} \tag{26}$$

where the sum runs over the site index. Here, we use the related measure of localization – the fractal dimension of the states, given by

$$D_k = -\frac{\log(\text{IPR}_k)}{\log(N)} \tag{27}$$

where $N$ is the system size. $D_k = 0(1)$ indicates localized (extended) states.

**Mosaic Model (MM)** We first consider a class of tight-binding models defined by the Hamiltonian Wang et al. (2020)

$$H = t \sum_n \left(c_{n+1}^\dagger c_n + \text{H.c.}\right) + \sum_n V_n(\lambda, \phi) c_n^\dagger c_n, \tag{28}$$

where $t$ is the nearest-neighbor coupling, $c_n(c_n^\dagger)$ is the annihilation (creation) operator at site $n$ and the potential energy $V_n$ is given by

$$V_n(\lambda, \phi) = \begin{cases} \lambda \cos(2\pi n b + \phi) & j = m\kappa \\ 0, & \text{otherwise,} \end{cases} \tag{29}$$

To introduce quasiperiodicity, we set $b$ to be irrational (in particular, we choose $b$ to be the golden ratio $\frac{1+\sqrt{5}}{2}$). $\kappa$ is an integer and the quasiperiodic potential occurs with interval $\kappa$. The energy ($E$) spectrum for this model generically contains extended and localized regimes separated by a mobility edge. Interestingly, a unique feature found here is that the mobility edges are present for an arbitrarily strong quasiperiodic potential (i.e. there are always extended states present in the system that co-exist with localized ones).

The mobility edge can be described by $g(\lambda, E) \equiv \lambda - |f_\kappa(E)| = 0$. $g(\lambda, E) > 0$ and $g(\lambda, E) < 0$ correspond to localized and extended phases, respectively. Learning the mobility edge therefore

hinges on learning the "order parameter" $g(\lambda, E)$. Admittedly, this problem can be tackled by many other theoretical methods for this class of models Wang et al. (2020), but we will demonstrate below that our KAN framework is ready and convenient to take in assumptions and inductive biases from human users.

Let us assume a hypothetical user Alice, who is a new PhD student in condensed matter physics, and she is provided with a $[2, 1]$ KAN as an assistant for the task. Firstly, she understands that this is a classification task, so it is wise to set the activation function in the second layer to be sigmoid by using the `fix_symbolic` functionality. Secondly, she realizes that learning the whole 2D function $g(\lambda, E)$ is unnecessary because in the end she only cares about $\lambda = \lambda(E)$ determined by $g(\lambda, E) = 0$. In so doing, it is reasonable to assume $g(\lambda, E) = \lambda - h(E) = 0$. Alice simply sets the activation function of $\lambda$ to be linear by again using the `fix_symbolic` functionality. Now Alice trains the KAN network and conveniently obtains the mobility edge, as shown in Figure 26. Alice can get both intuitive qualitative understanding (bottom) and quantitative results (middle), which well match the ground truth (top).

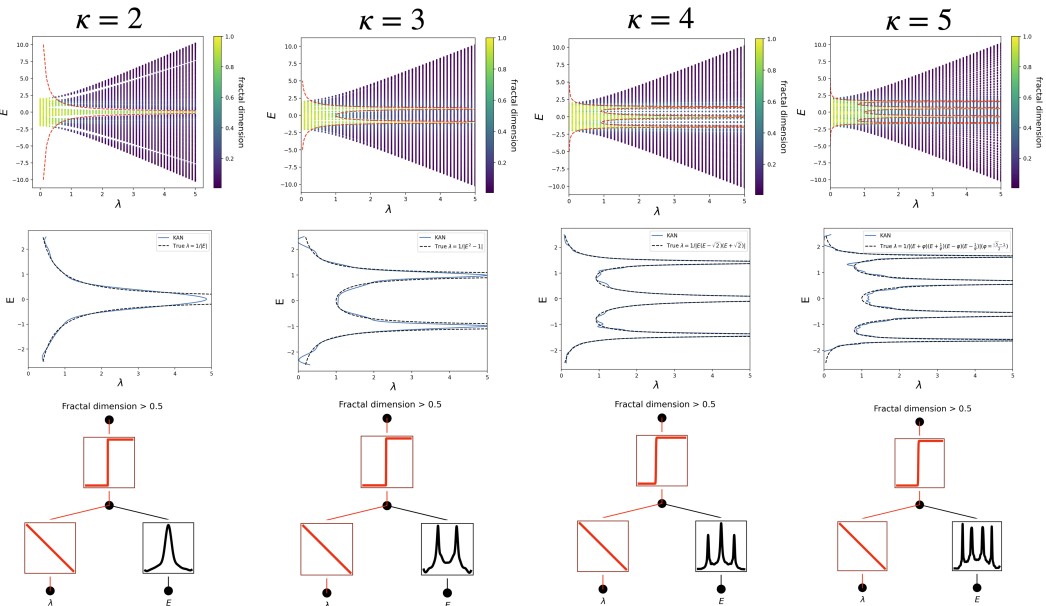

Figure 26: Results for the Mosaic Model. Top: phase diagram. Middle and Bottom: KANs can obtain both qualitative intuition (bottom) and extract quantitative results (middle). $\varphi = \frac{1+\sqrt{5}}{2}$ is the golden ratio.

**Generalized Andre-Aubry Model (GAAM)** We next consider a class of tight-binding models defined by the Hamiltonian Ganeshan et al. (2015)

$$H = t \sum_n \left( c_{n+1}^\dagger c_n + \text{H.c.} \right) + \sum_n V_n(\alpha, \lambda, \phi) c_n^\dagger c_n, \tag{30}$$

where $t$ is the nearest-neighbor coupling, $c_n(c_n^\dagger)$ is the annihilation (creation) operator at site $n$ and the potential energy $V_n$ is given by

$$V_n(\alpha, \lambda, \phi) = 2\lambda \frac{\cos(2\pi n b + \phi)}{1 - \alpha \cos(2\pi n b + \phi)}, \tag{31}$$

which is smooth for $\alpha \in (-1, 1)$. To introduce quasiperiodicity, we again set $b$ to be irrational (in particular, we choose $b$ to be the golden ratio). As before, we would like to obtain an expression for the mobility edge. For these models, the mobility edge is given by the closed form expression Ganeshan et al. (2015); Wang et al. (2021),

$$\alpha E = 2(t - \lambda). \tag{32}$$

We randomly sample the model parameters: $\phi$, $\alpha$ and $\lambda$ (setting the energy scale $t = 1$) and calculate the energy eigenvalues as well as the fractal dimension of the corresponding eigenstates, which forms our training dataset.

Here the "order parameter" to be learned is $g(\alpha, E, \lambda, \phi) = \alpha E + 2(\lambda - 1)$ and mobility edge corresponds to $g = 0$. Let us again assume that Alice wants to figure out the mobility edge but only has access to IPR or fractal dimension data, so she decides to use KAN to help her with the task. Alice wants the model to be as small as possible, so she could either start from a large model and use auto-pruning to get a small model, or she could guess a reasonable small model based on her understanding of the complexity of the given problem. Either way, let us assume she arrives at a $[4, 2, 1, 1]$ KAN. First, she sets the last activation to be sigmoid because this is a classification problem. She trains her KAN with some sparsity regularization to accuracy 98.7% and visualizes the trained KAN in Figure 25 (a) step 1. She observes that $\phi$ is not picked up on at all, which makes her realize that the mobility edge is independent of $\phi$ (agreeing with Eq. (32)). In addition, she observes that almost all other activation functions are linear or quadratic, so she turns on automatic symbolic snapping, constraining the library to be only linear or quadratic. After that, she immediately gets a network which is already symbolic (shown in Figure 25 (a) step 2), with comparable (even slightly better) accuracy 98.9%. By using `symbolic_formula` functionality, Alice conveniently gets the symbolic form of $g$, shown in Table 12 GAAM-KAN auto (row three). Perhaps she wants to cross out some small terms and snap coefficient to small integers, which takes her close to the true answer.

This hypothetical story for Alice would be completely different if she is using a symbolic regression method. If she is lucky, SR can return the exact correct formula. However, the vast majority of the time SR does not return useful results and it is impossible for Alice to "debug" or interact with the underlying process of symbolic regression. Furthermore, Alice may feel uncomfortable/inexperienced to provide a library of symbolic terms as prior knowledge to SR before SR is run. By constrast in KANs, Alice does not need to put any prior information to KANs. She can first get some clues by staring at a trained KAN and only then it is her job to decide which hypothesis she wants to make (e.g., "all activations are linear or quadratic") and implement her hypothesis in KANs. Although it is not likely for KANs to return the correct answer immediately, KANs will always return something useful, and Alice can collaborate with it to refine the results.

**Modified Andre-Aubry Model (MAAM)** The last class of models we consider is defined by the Hamiltonian Biddle & Sarma (2010)

$$H = \sum_{n \neq n'} te^{-p|n-n'|} \left( c_n^\dagger c_{n'} + \text{H.c.} \right) + \sum_n V_n(\lambda, \phi) c_n^\dagger c_n, \tag{33}$$

where $t$ is the strength of the exponentially decaying coupling in space, $c_n (c_n^\dagger)$ is the annihilation (creation) operator at site $n$ and the potential energy $V_n$ is given by

$$V_n(\lambda, \phi) = \lambda \cos(2\pi nb + \phi), \tag{34}$$

As before, to introduce quasiperiodicity, we set $b$ to be irrational (the golden ratio). For these models, the mobility edge is given by the closed form expression Biddle & Sarma (2010),

$$\lambda \cosh(p) = E + t = E + t_1 \exp(p) \tag{35}$$

where we define $t_1 \equiv t\exp(-p)$ as the nearest neighbor hopping strength, and we set $t_1 = 1$ below.

Let us assume Alice wants to figure out the mobility edge for MAAM. This task is more complicated and requires more human wisdom. As in the last example, Alice starts from a $[4, 2, 1, 1]$ KAN and trains it but gets an accuracy around 75% which is less than acceptable. She then chooses a larger $[4, 3, 1, 1]$ KAN and successfully gets 98.4% which is acceptable (Figure 25 (b) step 1). Alice notices that $\phi$ is not picked up on by KANs, which means that the mobility edge is independent of the phase factor $\phi$ (agreeing with Eq. (35)). If Alice turns on the automatic symbolic regression (using a large library consisting of exp, tanh etc.), she would get a complicated formula in Tabel 12-MAAM-KAN auto, which has 97.1% accuracy. However, if Alice wants to find a simpler symbolic formula, she will want to use the manual mode where she does the symbolic snapping by herself. Before that she finds that the $[4, 3, 1, 1]$ KAN after training can then be pruned to be $[4, 2, 1, 1]$, while maintaining 97.7% accuracy (Figure 25 (b)). Alice may think that all activation functions except those dependent on $p$ are linear or quadratic and snap them to be either linear or quadratic manually by using `fix_symbolic`. After snapping and retraining, the updated KAN is shown

| Functionality | Descriptions |
|---|---|
| `model.fit(dataset)` | training model on dataset |
| `model.plot()` | plotting |
| `model.prune()` | pruning |
| `model.fix_symbolic(l,i,j,fun)` | fix the activation function $\phi_{l,i,j}$ to be the symbolic function `fun` |
| `model.suggest_symbolic(l,i,j)` | suggest symbolic functions that match the numerical value of $\phi_{l,i,j}$ |
| `model.auto_symbolic()` | use top 1 symbolic suggestions from `suggest_symbolic` to replace all activation functions |
| `model.symbolic_formula()` | return the symbolic formula |

Table 13: KAN functionalities

in Figure 25 (c) step 3, maintaining 97.7% accuracy. From now on, Alice may make two different choices based on her prior knowledge. In one case, Alice may have guessed that the dependence on $p$ is cosh, so she sets the activations of $p$ to be cosh function. She retrains KAN and gets 96.9% accuracy (Figure 25 (c) Step 4A). In another case, Alice does not know the cosh $p$ dependence, so she pursues simplicity and again assumes the functions of $p$ to be quadratic. She retrains KAN and gets 95.4% accuracy (Figure 25 (c) Step 4B). If she tried both, she would realize that cosh is better in terms of accuracy, while quadratic is better in terms of simplicity. The formulas corresponding to these steps are listed in Table 12. It is clear that the more manual operations are done by Alice, the simpler the symbolic formula is (which slight sacrifice in accuracy). KANs have a "knob" that a user can tune to trade-off between simplicity and accuracy (sometimes simplicity can even lead to better accuracy, as in the GAAM case).

## V  KAN FUNCTIONALITIES

Table 13 includes common functionalities that users may find useful.

## W  DEPENDENCE ON HYPERPARAMETERS

We show the effects of hyperparamters on the $f(x,y) = \exp(\sin(\pi x) + y^2)$ case in Figure 27. To get an interpretable graph, we want the number of active activation functions to be as small (ideally 3) as possible.

(1) We need entropy penalty to reduce the number of active activation functions. Without entropy penalty, there are many duplicate functions.

(2) Results can depend on random seeds. With some unlucky seed, the pruned network could be larger than needed.

(3) The overall penalty strength $\lambda$ effectively controls the sparsity.

(4) The grid number $G$ also has a subtle effect on interpretability. When $G$ is too small, because each one of activation function is not very expressive, the network tends to use the ensembling strategy, making interpretation harder.

(5) The piecewise polynomial order $k$ only has a subtle effect on interpretability. However, it behaves a bit like the random seeds which do not display any visible pattern in this toy example.

## X  REMARK ON GRID SIZE

For both PDE and regression tasks, when we choose the training data on uniform grids, we witness a sudden increase in training loss (i.e., sudden drop in performance) when the grid size is updated to a large level, comparable to the different training points in one spatial direction. This could be due to implementation of B-spline in higher dimensions and needs further investigation.

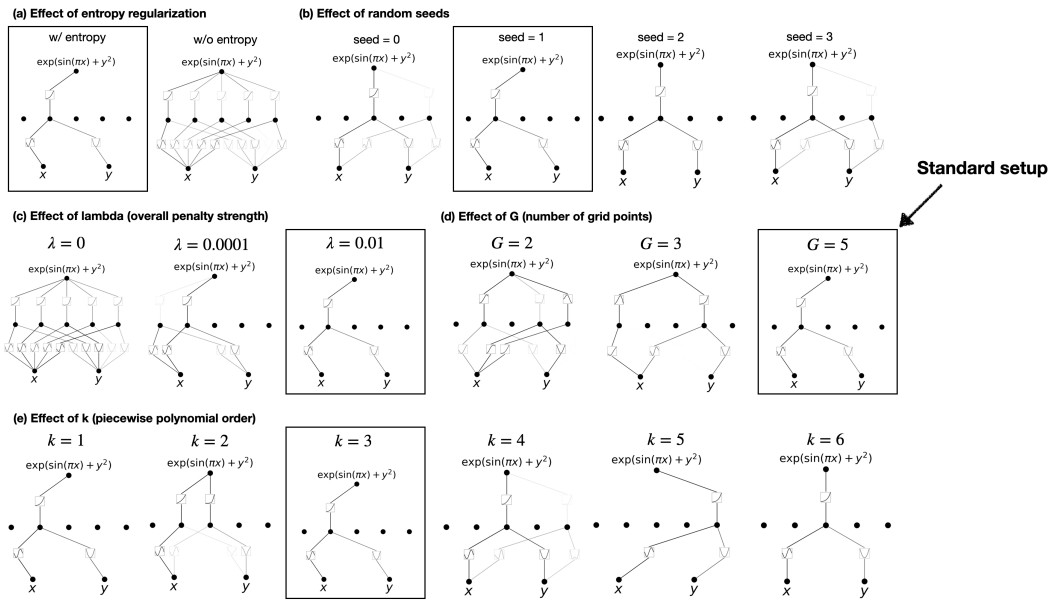

Figure 27: Effects of hyperparameters on interpretability results.

## Y  RELATED WORKS

**Kolmogorov-Arnold theorem and neural networks.** The connection between the Kolmogorov-Arnold theorem (KAT) and neural networks is not new in the literature Poggio (2022); Schmidt-Hieber (2021); Sprecher & Draghici (2002); Köppen (2002); Lin & Unbehauen (1993); Lai & Shen (2021); Leni et al. (2013); Fakhoury et al. (2022); Ismayilova & Ismailov (2024); Poluektov & Polar (2023), but the pathological behavior of inner functions makes KAT appear unpromising in practice Poggio (2022). Most of these prior works stick to the original 2-layer width-$(2n+1)$ networks, which were limited in expressive power and many of them are even predating back-propagation. Therefore, most studies were built on theories with rather limited or artificial toy experiments. More broadly speaking, KANs are also somewhat related to generalized additive models (GAMs) Agarwal et al. (2021), graph neural networks Zaheer et al. (2017) and kernel machines Song et al. (2018). The connections are intriguing and fundamental but might be out of the scope of the current paper. Our contribution lies in generalizing the Kolmogorov network to arbitrary widths and depths, revitalizing and contexualizing them in today's deep learning stream, as well as highlighting its potential role as a foundation model for AI + Science.

**Neural Scaling Laws (NSLs).** NSLs are the phenomena where test losses behave as power laws against model size, data, compute etc Kaplan et al. (2020); Henighan et al. (2020); Gordon et al. (2021); Hestness et al. (2017); Sharma & Kaplan (2020); Bahri et al. (2021); Michaud et al. (2023a); Song et al. (2024). The origin of NSLs still remains mysterious, but competitive theories include intrinsic dimensionality Kaplan et al. (2020), quantization of tasks Michaud et al. (2023a), resource theory Song et al. (2024), random features Bahri et al. (2021), compositional sparsity Poggio (2022), and maximum arity Michaud et al. (2023b). This paper contributes to this space by showing that a high-dimensional function can surprisingly scale as a 1D function (which is the best possible bound one can hope for) if it has a smooth Kolmogorov-Arnold representation. Our paper brings fresh optimism to neural scaling laws. We have shown in our experiments that this fast neural scaling law can be achieved on synthetic datasets, but future research is required to address the question whether this fast scaling is achievable for more complicated tasks (e.g., language modeling): Do KA representations exist for general tasks? If so, does our training find these representations in practice?

**Mechanistic Interpretability (MI).** MI is an emerging field that aims to mechanistically understand the inner workings of neural networks Olsson et al. (2022); Meng et al. (2022); Wang et al. (2023); Elhage et al. (2022b); Nanda et al. (2023); Zhong et al. (2023); Liu et al. (2023); Elhage

et al. (2022a); Cunningham et al. (2023). MI research can be roughly divided into passive and active MI research. Most MI research is passive in focusing on understanding existing neural networks trained with standard methods. Active MI research attempts to achieve interpretability by designing intrinsically interpretable architectures or developing training methods to explicitly encourage interpretability Liu et al. (2023); Elhage et al. (2022a). Our work lies in the second category, where the model and training method are by design interpretable.

**Learnable activations.** The idea of learnable activations in neural networks is not new in machine learning. Trainable activations functions are learned in a differentiable way Goyal et al. (2019); Fakhoury et al. (2022); Ramachandran et al. (2017); Zhang et al. (2022) or searched in a discrete way Bingham & Miikkulainen (2022). Activation function are parametrized as polynomials Goyal et al. (2019), splines Fakhoury et al. (2022); Bohra et al. (2020); Aziznejad & Unser (2019), sigmoid linear unit Ramachandran et al. (2017), or neural networks Zhang et al. (2022). KANs use B-splines to parametrize their activation functions.

**Symbolic Regression.** There are many off-the-shelf symbolic regression methods based on genetic algorithms (Eureka Dubcáková (2011), GPLearn gpl, PySR Cranmer (2023)), neural-network based methods (EQL Martius & Lampert (2016), OccamNet Dugan et al. (2020)), physics-inspired method (AI Feynman Udrescu & Tegmark (2020); Udrescu et al. (2020)), and reinforcement learning-based methods Mundhenk et al. (2021). KANs are most similar to neural network-based methods, but differ from previous works in that our activation functions are continuously learned before symbolic snapping rather than manually fixed Dubcáková (2011); Dugan et al. (2020).

**Physics-Informed Neural Networks (PINNs) and Physics-Informed Neural Operators (PINOs).** In Section 4 PDE, we demonstrate that KANs can replace the paradigm of using MLPs for imposing PDE loss when solving PDEs. We refer to Deep Ritz Method Yu et al. (2018), PINNs Raissi et al. (2019); Karniadakis et al. (2021) for PDE solving, and Fourier Neural operator Li et al. (2020), PINOs Li et al. (2021); Kovachki et al. (2023); Maust et al. (2022), DeepONet Lu et al. (2021) for operator learning methods learning the solution map. There is potential to replace MLPs with KANs in all the aforementioned networks.

**AI for Mathematics.** AI has recently been applied to several problems in Knot theory, including detecting whether a knot is the unknot Gukov et al. (2021); Kauffman et al. (2020) or a ribbon knot Gukov et al. (2023), and predicting knot invariants and uncovering relations among them Hughes (2020); Craven et al. (2021; 2022); Davies et al. (2021). For a summary of data science applications to datasets in mathematics and theoretical physics see e.g. Ruehle (2020); He (2023), and for ideas how to obtain rigorous results from ML techniques in these fields, see Gukov et al. (2024).

## Z  DISCUSSION

In this section, we discuss KANs' limitations and future directions from the perspective of mathematical foundation, algorithms and applications.

**Mathematical aspects:** Although we have presented preliminary mathematical analysis of KANs (Theorem 2.1), our mathematical understanding of them is still very limited. The Kolmogorov-Arnold representation theorem has been studied thoroughly in mathematics, but the theorem corresponds to KANs with shape $[n, 2n + 1, 1]$, which is a very restricted subclass of KANs. Does our empirical success with deeper KANs imply something fundamental in mathematics? An appealing generalized Kolmogorov-Arnold theorem could define "deeper" Kolmogorov-Arnold representations beyond depth-2 compositions, and potentially relate smoothness of activation functions to depth. Hypothetically, there exist functions which cannot be represented smoothly in the original (depth-2) Kolmogorov-Arnold representations, but might be smoothly represented with depth-3 or beyond. Can we use this notion of "Kolmogorov-Arnold depth" to characterize function classes?

**Algorithmic aspects:** We discuss the following:

(1) Accuracy. Multiple choices in architecture design and training are not fully investigated so alternatives can potentially further improve accuracy. For example, spline activation functions might be replaced by radial basis functions or other local kernels. Adaptive grid strategies can be used.

(2) Efficiency. One major reason why KANs run slowly is because different activation functions cannot leverage batch computation (large data through the same function). Actually, one can interpolate between activation functions being all the same (MLPs) and all different (KANs), by grouping activation functions into multiple groups ("multi-head"), where members within a group share the same activation function.

(3) Hybrid of KANs and MLPs. KANs have two major differences compared to MLPs:

    (i) activation functions are on edges instead of on nodes,

    (ii) activation functions are learnable instead of fixed.

(4) Adaptivity. Thanks to the intrinsic locality of spline basis functions, we can introduce adaptivity in the design and training of KANs to enhance both accuracy and efficiency: see the idea of multi-level training like multigrid methods as in Zhang et al. (2021); Xu & Zikatanov (2017), or domain-dependent basis functions like multiscale methods as in Chen et al. (2023).

**Application aspects:** We have presented some preliminary evidences that KANs are more effective than MLPs in science-related tasks, e.g., fitting physical equations and PDE solving. We expect that KANs may also be promising for solving Navier-Stokes equations, density functional theory, or any other tasks that can be formulated as regression or PDE solving. We would also like to apply KANs to machine-learning-related tasks, which would require integrating KANs into current architectures, e.g., transformers – one may propose "kansformers" which replace MLPs by KANs in transformers.

**KAN as a "language model" for AI + Science** The reason why large language models are so transformative is because they are useful to anyone who can speak natural language. The language of science is functions. KANs are composed of interpretable functions, so when a human user stares at a KAN, it is like communicating with it using the language of functions. This paragraph aims to promote the AI-Scientist-Collaboration paradigm rather than our specific tool KANs. Just like people use different languages to communicate, we expect that in the future KANs will be just one of the languages for AI + Science, although KANs will be one of the very first languages that would enable AI and human to communicate. However, enabled by KANs, the AI-Scientist-Collaboration paradigm has never been this easy and convenient, which leads us to rethink the paradigm of how we want to approach AI + Science: Do we want AI scientists, or do we want AI that helps scientists? The intrinsic difficulty of (fully automated) AI scientists is that it is hard to make human preferences quantitative, which would codify human preferences into AI objectives. In fact, scientists in different fields may feel differently about which functions are simple or interpretable. As a result, it is more desirable for scientists to have an AI that can speak the scientific language (functions) and can conveniently interact with inductive biases of individual scientist(s) to adapt to a specific scientific domain.

