# OpenReview forum: "KAN: Kolmogorov–Arnold Networks"
_ICLR.cc/2025/Conference — ICLR 2025 Oral_

### Official Review · Reviewer_zHkY · 2024-10-21

**Soundness:** 3
**Presentation:** 3
**Contribution:** 3
**Rating:** 8
**Confidence:** 3

**Summary:**

This paper proposes Kolmogorov-Arnold networks (KANs) as a more interpretable alternative to multi-layer perceptrons (MLPs), particularly for scientific applications. KANs replace the fixed activation functions on nodes in MLPs with learnable activation functions on edges, represented by B-splines. The paper explores the theoretical approximation capabilities of KANs, proposes training techniques for interpretability and accuracy (pruning, symbolifying, grid extension), and demonstrates their application in mathematics (knot theory), physics (Anderson localization), and other (toy) examples.

**Strengths:**

+ The paper presents an interesting and original neural network architecture inspired by the Kolmogorov-Arnold representation theorem, which has very appealing properties for some applications (e.g. interpetability).

+ The paper is generally well-written and easy to follow. The architecture of KANs is clearly explained, and the figures effectively illustrate the differences between KANs and MLPs.

+ The emphasis on interpretability (and the ability to extract symbolic formulas) is relevant for scientific applications, where understanding the underlying mechanisms learned by the model is crucial.

**Weaknesses:**

- Technically, the paper could be considered to be over the page limit. While the main text does keep to 9 pages, this is only the case because extremely important sections, e.g., related works and the discussion, are moved to the appendix. I feel important sections like these have to be included in the main text, even if this means other parts of the text need to be shortened to make space for them. I am sure many authors of other submissions have faced similar "space issues" and consequently had to cut parts from their main text. This paper is the only submission I am reviewing that "misuses" the appendix to circumvent the page limit restrictions (I think the appendix should be reserved for things like detailed proofs or additional, but non-essential, information). Sorry for bringing up a technicality like this, but I feel it is necessary in the interest of fairness.

- The paper contains very strong claims, e.g. that KANs with finite grid size beat the curse of dimensionality. At the same time, the authors admit that the constant $C$ in Eq.11 (Theorem 2.1) implicitly depends on the dimension (but details are left to future work). As the saying goes, extraordinary claims require extraordinary evidence, which unfortunately is missing here. The authors should either weaken their claims or provide strong empirical evidence.

- While reading the paper, I couldn't help but feel that the name "Kolmogorov-Arnold network", at least to some degree, overstates the difference to MLPs: KANs and MLPs have more in common than a superficial reading of the paper would suggest. Comparing the Kolmogorov-Arnold Theorem (KAT) with the Universal Approximation Theorem (UAT), a very important aspect is that KAT requires a width of only $2n+1$, whereas the UAT only holds in the limit of infinite width. I understand that relaxing the width-restriction from KAT may be practically useful (and the authors motivate this choice well), but it also blurs the lines between KAT and UAT. To me, KANs seem more like a KAT-inspired re-interpretation of MLPs (rather than something entirely new), which is useful for tasks where a symbolic interpretation of the learned function is desirable. For example, one distinguishing feature of KANs pointed out in the abstract are the learnable activation functions, wheareas MLPs have a fixed activation function. But if I re-write an MLP $W_2\sigma(W_1\mathbf{x})$ as $W_2\phi(\mathbf{x})$ with $phi(\mathbf{x}) = W_1\mathbf{x}$, I can also consider $phi$ to be a "learnable activation function". Similarly, the basis functions of the B-splines in KANs could be equivalently considered simply as additional input features, similar to Fourier features (https://arxiv.org/abs/2006.10739), or other tricks that were sometimes used in older works (e.g. feeding the square of an input to a layer, see e.g., "Neural Networks: Tricks of the Trade"). I think the authors should try to "de-mistify" KANs by pointing out and discussing these similarities between MLPs and KANs more transparently.

- While the examples in mathematics and physics are interesting, the scale of these problems is relatively small. The paper lacks evaluation on larger, more complex datasets. This makes it difficult to assess the true scalability and practical utility of KANs for many real-world applications. For example, a very simple test could be to compare KANs vs. MLPs on something like MNIST. Even if KANs perform worse than MLPs here, this does not mean at all that they are not useful. However, it would certainly help readers to assess whether KANs (in their current form) are likely to be directly relevant for their work or not. I encourage the authors to include at least one example like this.

- The example in Appendix J on continual learning is practically not relevant and too trivial. Of course, the B-splines in KANs are "local" and thus the corresponding weights/coefficients are not affected by catastrophic forgetting in this setting, but the toy example is also clearly constructed to take advantage of this fact. The authors should either demonstrate that KANs are less affected by catastrophic forgetting than MLPs in a more realistic/"less constructed" setting, or remove this claim from the paper.

- The paper mentions that KANs are slower to train than MLPs by roughly a factor of 10, and that KANs cannot leverage batch computation in the same way as MLPs. However, these discussion are hidden in the appendix. I think this is a highly important and practically relevant aspect, which should be discussed more prominently and in greater detail in the main text. An analysis of the computational cost, including training time and memory usage, is needed to fully understand the trade-off between interpretability and efficiency.

- The appendix is very long with >25 additional pages, not all of which feel strictly necessary to me. I think the authors should consider cutting some parts of the appendix in the interest of brevity. For example, the secion on LANs, while interesting, feels better suited as a separate submission (after doing more experiments/refining the idea).

**Questions:**

* How do KANs compare to other interpretable models, like Neural Additive Models or symbolic regression methods, in terms of accuracy, interpretability, and computational cost, especially on larger datasets?

* Can you provide a more detailed analysis of the computational complexity of KANs, both for training and inference? Have you explored any optimizations beyond the "multi-head" approach?

* Can you elaborate on the dependence of the constant C in Theorem 2.1 on the dimension? How does this impact the scalability of KANs to higher-dimensional problems?

* Can you provide empirical evidence supporting the claim that KANs can naturally work in continual learning without catastrophic forgetting, beyond the toy example in Appendix J?

* How sensitive are KANs to the choice of hyperparameters, such as the grid size $G$ and the spline order $k$? Have you explored techniques for automatically selecting these hyperparameters?

**Additional Feedback** (this does not affect my score)

* In my opinion, some formulations, e.g. "the Kolmogorov-Arnold representation theorem was basically sentenced to death" (l.119), or "the fastest scaling exponent ever" (l.1888) appear overly "dramatic" for a scientific paper. I suggest the authors scan their paper for cases like these and try to use more neutral/grounded language.

* There are several typos, e.g. "altnertives" (abstract), "On" (l.084), "staricase-like" (l.965), "theLANs" (l.1705), "maximu" (l.1885). I suggest the authors use an automated spell-checker to find problems like these and correct them (I likely didn't find all typos).

* Some terms are spelled inconsistently in different places of the text, for example "Deepmind" vs. "deepmind" (I actually would recommend to use "DeepMind" instead), or "ReLU" vs. "ReLu". The authors should decide on one spelling and keep it throughout the paper.

**In summary**
I believe this is a very interesting paper that should be published, but I feel major revisions are necessary (see above). If the authors implement these revisions (or at least most of them), I will gladly raise my score.

---

> ### Author Response · Authors · 2024-11-20
> **Response to Reviewer zHkY Part 1**
>
> We would like to thank the reviewer for their careful reading and constructive suggestions. We have conducted additional experiments to showcase KANs’ ability to perform more complicated tasks and compare them with stronger baselines (Appendix A, B, C), and study hyperparameter dependence of KANs & MLPs (Appendix D). We respond to each point below.
>
> > Q1:The paper contains very strong claims, but the evidence isn’t enough to support the strong claims. As the saying goes, extraordinary claims require extraordinary evidence, which unfortunately is missing here. The authors should either weaken their claims or provide strong empirical evidence.
>
> A1: We totally agree! We attempt to work on both directions here: (1) We added more empirical evidence to demonstrate KAN’s ability to tackle more complex problems (including image fitting, complex PDE solving, MNIST classification in appendices A, B, C); (2) we have toned down our contributions and highlighted the limitations. Specifically regarding the neural scaling law, we have added a caveat “We also remark that: since the assumption in the theorem is a  strong one, the neural scaling law should not be expected to be universally applicable to all machine learning applications.”. We also acknowledge the slow training problem in the abstract and the PDE section.
>
> > Q2: The name “Kolmogorov-Arnold network” overstates the difference to MLPs.
>
> A2: Thanks for bringing up the concern. We have downplayed our novelty claim in conclusions by acknowledging the possibility that MLPs and KANs might be equivalent in some aspects but different in others, which will need more formal study in the future. We have added this paragraph in the conclusion part: “We also acknowledge that the similarities and differences between MLPs and KANs require more study, both theoretically and empirically. For example, a reasonable criticism of KANs is that they can be rewritten as MLPs or the other way around since the notion of  "edge" vs "node" is somewhat dual. Future work should aim to better clarify similarities and differences from the perspective of optimization, generalization, etc. For example, a recent preprint Wang et al. (2024b) shows that although KANs and MLPs are both universal approximators, KANs have fewer spectral biases than MLPs.”
>
> > Q3: The scale of the problems is relatively small. The paper lacks evaluation on larger, more complex datasets. The reviewer suggests adding the MNIST experiment.
>
> A3: Thanks for the great suggestion! We have now added MNIST experiments in Appendix C. KANs and MLPs have comparable performance in terms of loss and accuracy. This is probably because the MNIST dataset is too simple. KANs consume much more training time than MLPs. We expect these slowdown factors to have straightforward solutions, and combining KANs with Convolutional neural networks is a promising direction to incorporate symmetry inductive biases into architectures. Another interesting observation: the shape [784,100,10] (1 hidden layer of size 100) is optimal both for MLP and for KAN, which seems to imply there is something universal across different architectures. This might be a good starting point to formally study Q2 in the future.
>
> Besides the MNIST dataset, we also added image-fitting experiments (Appendix A) and complex PDE experiments (Appendix B), demonstrating KANs’ applicability to handle more complex datasets.

---

> > ### Author Response · Authors · 2024-11-20
> > **Response to Reviewer zHkY Part 2**
> >
> > > Q4: The continual learning example is somewhat trivial.
> >
> > A4: We completely agree. We find that by employing deeper KANs, the continual learning feature can partially or completely go away (new experiments in Appendix Q). we have carefully reframed our language to avoid overclaims that may cause readers’ misunderstanding.
> >
> > > Q5: KANs are slower to train than MLPs. This fact should be stated more explicitly in the paper. Also, there should be an analysis of computational cost.
> >
> > A5: Thanks for the great suggestions. (1) We have added texts clarifying that “KANs are much slower than MLPs to train” in several places throughout the paper, including the abstract and the results section. (2) Theoretically, we added complexity analysis in Section 2.2. (3) Empirically, we report wall time in Appendices A-C.
> >
> > > Q6: The appendices are too long and some parts feel unnecessary.
> >
> > A6: We have now removed the LAN (learnable activation network) part, as well as two sections containing a lot of KAN diagrams for the Feynman dataset and special functions. We’re happy to remove more sections if necessary.
> >
> > > Q7: How do KANs compare to other interpretable models, like Neural Additive Models or symbolic regression methods, in terms of accuracy, interpretability, and computational cost, especially on larger datasets?
> >
> > A7: Thanks for the great question!
> >
> > Regarding comparison to symbolic regression methods: KAN, as a network-based method, has strong capability (in fitting even non-symbolic functions) that makes it unfavorable for standard symbolic regression benchmarks. For example, KAN ranks second-to-last in GEOBENCH (https://openreview.net/forum?id=TqzNI4v9DT), whereas the last-ranked one EQL is also a network-based model, which turned out to be useful at least for certain problems despite its inability to do well on benchmarks. On the one hand, we would like to explore ways to restrict KANs' hypothesis space so that KANs can achieve good performance on symbolic regression benchmarks. On the other hand, we want to point out that KANs have good features that are hard to evaluate with existing benchmarks: (1) interactivity. It is very hard to "debug" evolutionary-based symbolic regression methods since the evolution process is long and random. However, it is relatively easier to visualize the training dynamics of KANs, which gives human users intuition on what could go wrong. (2) The ability to ``discover'' new functions. Since most symbolic regression methods require the input of the symbolic library, they cannot discover things they are not given. For example, if the ground truth formula contains a special function but is not given in the symbolic library, all SR methods will fail definitely. However, KANs can discover the need for a new function whose numerical behavior suggests maybe it is a Bessel function; see Figure 23 (d) for an example.
> >
> > Regarding relation to additive models: [N, 1] KAN is equivalent to an additive model, and [N, 1, 1] KAN is equivalent to a generalized additive model, so in some sense, KANs can be viewed as a generalized version of neural additive models.

---

> > > ### Author Response · Authors · 2024-11-20
> > > **Response to Reviewer zHkY Part 3**
> > >
> > > > Q8: Can you elaborate on the dependence of the constant C in Theorem 2.1 on the dimension? How does this impact the scalability of KANs to higher-dimensional problems?
> > >
> > > A8: The constant C has the order of the number of edges in the KAN network which is O(N^2) when N is network width. In the original KA theorem, a width N = 2d+1 is enough (d is the input dimension). If this conclusion applies to deeper KANs with smooth representations (this is a big IF though!), we would expect the constant to be O(d^2 L) where d is the input dimension and L is the network depth. Since C is only polynomially dependent on d, this is a reasonably good scaling. However, to turn these speculations into formal theorems, more theoretical work needs to be done.
> > >
> > > > Q9: How sensitive are KANs to the choice of hyperparameters?
> > >
> > > A9: Empirically we found k < 3 is not very expressive, while k > 3 can lead to stability issues, so we always choose k = 3 throughout the paper. If one cares about interpretability but not accuracy, typical grid sizes like 3, 5, 10 should work fine. If one cares about interpretability, one may start from a small grid size and then do grid refinement as training goes on.
> > >
> > > > Q10: In my opinion, some formulations, e.g. "the Kolmogorov-Arnold representation theorem was basically sentenced to death" (l.119), or "the fastest scaling exponent ever" (l.1888) appear overly "dramatic" for a scientific paper. I suggest the authors scan their paper for cases like these and try to use more neutral/grounded language.
> > >
> > > A10: We apologize for not being professional! We have rephrased them so that they sound more scientific.
> > >
> > > > Q11: There are several typos and some terms are used inconsistently.
> > >
> > > A11: Thanks for reading so carefully! We have corrected them.

---

> > > > ### Comment · Reviewer_zHkY · 2024-11-20
> > > >
> > > > The authors have answered my questions and tried their best in taking the reviewers' criticism (including my own) to improve their paper. The strong claims made in the original version have been toned down and additional experiments have been added to support them. Overall, I feel that the paper is greatly improved as a result. Even if KANs and MLPs turn out to be equivalent in some sense, or "dual" to each other (as the authors put it), this additional perspective can be very useful for some applications. Also, in the new version of the manuscript, the authors are upfront about the limitations of KANs, so readers can decide whether the tradeoffs (e.g. interpretability vs. efficiency) are worth it for their particular application.
> > > >
> > > > I have raised my score accordingly.

---

> > > > > ### Author Response · Authors · 2024-11-24
> > > > >
> > > > > We want to thank the reviewer for their valuable time and kind reconsideration of the updated manuscript. We learned a lot from the suggestions which helped us greatly improve our manuscript.

---

### Official Review · Reviewer_whLk · 2024-11-01

**Soundness:** 3
**Presentation:** 2
**Contribution:** 3
**Rating:** 6
**Confidence:** 4

**Summary:**

The authors propose a novel neural network architecture based on the Kolmogorov-Arnold Representation Theorem (KAT), called Kolmogorov-Arnold Networks (KANs). Although the KAT has been studied in the context of neural networks, KANs differ from most other works by allowing for wider/deeper representations, in addition to implementing functions implied by the theorem via B-splines. The authors formally describe the main component of their new architecture, the KAN-layer, which draws inspiration from the representation formula implied by the KAT. In essence, a KAN-layer works by approximating the functions implied by KAT via B-splines, as well as adding a 'residual' connection using a silu activation. These KAN-layers can then be composed together to create a deeper KAN architecture. The authors also state a theorem regarding the approximation capabilities of KANs, assuming a specific smooth representation of the target function.

The authors then go on to detail experiments using their new architecture on two main goals: a) providing interpretable results from scientific data; and b) creating accurate predictions for regression tasks and learning the solution of a PDE. On their first goal, they evaluate KANs on tasks from mathematics (knot theory) and physics (Anderson localization), as well as some toy problems on symbolic regression. For their second goal, they compare KANs against certain MLPs on regression for several smooth functions with closed form solutions, as well as a physics-informed problem where they learn the solution of a 2D Poisson equation.

Finally, the authors summarize their results and argue that KANs offer significant advantages over traditional MLPs for scientific tasks that are "small-scale", leaving more thorough experimentation on larger-scale problems for future work.

**Strengths:**

### Originality/Significance:
The paper presents several interesting original thoughts, and helps bring attention to new forms of computation to scientific deep learning. Although the connections between KAT and neural networks are quite old, dating back to the 1960s and 1970s, their work helps renew attention to this field, and applies modern knowledge about neural networks to improve on previous failed attempts of using KAT for applied problems. The authors also present interesting ideas about interpretability for deep neural networks, by using KANs to guide the discovery of closed form expressions for unknown functions.

### Quality:
The main idea (exploiting deeper/wider forms of KAT) is well-motivated and appears theoretically promising. The authors provide valuable arguments on section 2 about why such an idea could work. The paper also comes with many experiments being conducted in the main text, as well as in the appendix. Overall, I believe this paper contains very interesting ideas, and helps spark discussion about an old topic using modern lenses.


### Clarity:
The ideas are presented in a mostly clear manner, with helpful figures to illustrate the main arguments. Their main theorem (Theorem 2.1) is clearly stated in the main text, including all of its assumptions.

**Weaknesses:**

## Summary of weaknesses:
In summary, although the main idea of the paper seems very interesting, my main criticism of this submission is regarding their experiments, which are not conclusive to me as a reviewer. Despite presenting a very large number of different experiments, they are all very simple, and comparisons against MLPs appear to be using very weak baselines. The authors do recognize that they mainly consider small-scale experiments [line 485], but even for scientific tasks there are more realistic and interesting problems to be considered, as will be detailed later. In addition to this, their main theorem regarding approximation power of KANs (theorem 2.1), which implies very strong scaling laws, use an assumption which is known to be wrong for KAT functions (ie: inner functions being $(k+1)$-times continuously differentiable). I further detail these criticisms below, and how the authors might address them.

## Experimental Setup:
I will mainly focus on the results reported in section 4 here, which is an area I am more familiar with. My main criticisms regarding experiments is that a) all functions considered are very well-behaved and not something for which deep learning would be used in practice, with other regression tools generally being better suited; b) I am not confident that their comparisons against MLPs are done in a consistent and fair way; c) there is not enough information to fully reproduce experiments. I expand on each of these points below.

a) For regression tasks (Toy datasets + Special Functions + Feynman Datasets), all the target functions are very simple, beyond what is typical for scientific machine learning. The functions from the "Special Functions" dataset are slightly more complicated than from the other two, but still a very simple task given enough training data. A better testbed for examining approximation properties and accuracy would be for example: 1) regression against complicated PDE solutions from publicly available dataset such as PDEBench and/or PDEArena, which are more relevant to scientific computing; 2) regression against images, which was in fact considered in the appendix for their "LAN" architecture, but not KANs; 3) weather modeling, using public datasets such as ERA5 (very challenging) or NOAA's SST-V2, for example. Regarding the experiment on PINNs, it would also be more comprehensive to study more than just the Poisson equation, which is again very simple for tasks of this nature. Suggestions for other PDEs to consider, which are not unrealistic for the first iteration of a new architecture include the Allen-Cahn equation and the Darcy problem. In short, although it is true that some problems in scientific computing are simpler or lower dimensional than ones in computer vision and NLP, which are big topics in deep learning, there is still no shortage of more realistic problems in science, which would provide a better testbed for their architecture.

b) Many aspects of the experiments provided are not likely to yield a fair comparison against MLP architectures. In some cases, only MLPs of very small width (such as 5 or 20) are considered, which is not realistic of what someone would use in practice, even if the number of parameters is similar to that of KANs used. For some of the regression tasks, the grid definition of KAN's B-splines can go as high as 1000, which means that a KAN of comparable width/depth still presents far more parameters (and likely flops) than an MLP. To address this issue, I would recommend keeping a fixed number of trainable parameters (or flops/training time), comparing different hyperparameter combinations, both for KANs and MLPs. Ideally, MLPs should have widths of at least 32 or 64 for any real scenario. In addition to this, MLPs are more efficiently trained using the Adam optimizer, whereas all experiments used the L-BFGS algorithm, which is not as common and can often lead to MLPs being stuck on local minima.

c) There are many missing details for reproducing the experiments in the paper, and no code was provided as supplementary material. For example, on many of the regression tasks, the target functions are reported, but not the total number of points in the training/testing set, nor what the domain for these functions is. Other missing information for reproducibility include: 1) the batch size used during training for different benchmarks; 2) the learning rate of the optimizer; 3) the architectures used in the continual learning example. The best way of solving all this missing information issue would be to **provide the code used for each experiment as a supplementary material**, organized neatly for each benchmark.


## Theoretical Assumptions:
A main assumption of theorem 2.1 of the paper is that the functions $\phi_{l,i,j}$ are $(k+1)$-times continuously differentiable. However, the KAT only guarantees that these functions are continuous, as and the authors themselves mention, it has been shown in the literature that these functions can provably be very chaotic, with infinite set of non-differentiability. The authors argue that by using wider/deeper representations, these irregularities may disappear, but this would require further theoretical evidence. I believe there are more reasonable ways of proving universal approximation guarantees for KANs, albeit without the strong scaling laws implied by the $k$-order spline functions. This main theorem therefore can likely be stated differently, in order to obtain a weaker but more theoretically reasonable statement.




## Other minor considerations that did not affect my score:
- [line 15 + many other mentions] The analogy of referring to computations on 'edges' vs 'nodes' is not very clear or compelling to me. Depending on how you chose to represent any given architecture as a graph, it could be argued either way whether an operation happens at an 'edge' or at a 'node'.
- [line 54] Although it is true that "In science, symbolic functions are prevalent," that is not generally the case in scientific computing, where solutions are often defined on complicated domains without reasonable closed-form expressions. In fact, for tasks where numerical solutions are needed (such as solutions to ODEs, PDEs, etc), it is almost always the case that the solutions are not symbolic.
- [line 260] There are only three examples in Figure 2, not six.
- [Figure 7] It is more common for the y-axis of the plots to show the error itself, not the error squared, as is done in the figure.
- [line 875] It would also be nice to see a comparison on the computational complexity of KANs vs MLPs, in addition to parameter count.
- [line 1819] In the case of PINNs, it is almost always better to sample points randomly and independently at each iteration, as opposed to maintaining a fixed grid. I would be interested to see how KANs perform in this setting.

**Questions:**

I detail below some questions/suggestions to the authors, in order of how important they would be toward increasing the review score of their submission.

- **[More Challenging Problems For Testing Accuracy]** As mentioned above, I think the paper would greatly benefit from more challenging benchmarks, as the ones reported are too simple and largely not relevant to scientific computing. Seeing KANs outperform realistic MLPs (using higher widths and Fourier features, for example) on more challenging tasks could highly increase my score for the submission, depending on the results. Some potential tasks are detailed above in the previous section.

- **[Comparison Against Symbolic Regression Baselines]** The authors highlight discovery of close-form expressions from data as a key feature of KANs, but experiments lack comparison against other symbolic regression baselines, as the ones detailed in lines 2115-2121. Seeing KANs outperform these baselines could make for a much stronger paper.

- **[Spline Grid Boundary]** Using B-splines require all inputs to fall within a specified interval, which may not be the case for intermediate layers of a deep KAN. On line 852, the authors mention they "update each grid on the fly according to its input activations," but no further information is given as to how this is done, to the best of my knowledge. I feel like this is an important aspect to detail in the paper, maybe even in the main text. Related to this, how would the authors deal with problems where the inputs are defined in an unbounded (or very large) domain, such as when inputs follow a distribution with a 'fat' tail?

- **[Skip Connections]** What is the effect of removing the silu 'basis function' used in the KAN layer? Are there other suitable functions to choose instead of silu? This would be a good ablation to consider reporting.

- **[Grid Extension]** The authors propose progressively extending the grid used for computing B-splines, which improves the predictive accuracy of a given model. Is there an advantage to progressively increasing this grid as opposed to starting with a large one? That would be an interesting ablation to include in the paper.

- **[Continual Learning]** For the continual learning experiment, what happens if a deeper KAN is used? It makes sense that the local nature of B-spline computations yields this nice feature, but it seems to me that once more compositions are at play this property would be lost.

---

> ### Author Response · Authors · 2024-11-20
> **Response to Reviewer whLk Part 1**
>
> We would like to thank the reviewer for their careful reading and constructive suggestions. We have conducted additional experiments to showcase KANs’ ability to perform more complicated tasks and compare them with stronger baselines (Appendix A, B, C), and study hyperparameter dependence of KANs & MLPs (Appendix D). The new experiments use standard training protocols to avoid the possibility of favoring KANs. We have also downplayed our claims and acknowledged the limitations of our method. We respond to each point below.
>
> > Q1: There are three major problems regarding experiments: (a) Regression tasks are very simple. Reviewers suggest regressing against complex PDE solutions, images and do weather modeling. (b) Training protocols are not standard and may disfavor MLPs. (c) Missing details for reproducibility.
>
> A1: Thanks for the very constructive suggestions!
>
> **To address (a), we conducted three families of experiments:**
>
> **(i) Image Fitting (Appendix A)**, given that images usually contain sharp local features or high-frequency features. We show that KANs can outperform all baselines (including more advanced baselines, i.e., SIREN, and MLP with random Fourier features) in terms of PSNR. However, we also want to mention that KANs have 10x - 20x higher training wall time. Our chosen images include “Camera Man” (a natural image), “Turbulence” (from PDEBench), and “The Starry Night” (Van Gohn’s painting), to show KANs’ consistently high performance across different styles.
>
> **(ii) PDE solving (Appendix B)**. In B.1, we use the Poisson equation with high-frequency solutions to test KANs’ ability to learn high-frequency modes. In B.2, we use the Allen-Cahn equation to test KAN’s ability to learn temporal PDEs and sharp boundaries. In B.3, we use Darcy flow to test KAN’s ability to learn unstructured solutions. We find that KANs can achieve comparable accuracy with MLPs and MLPs with random Fourier features (under “fair” comparison, i.e., using the Adam optimizer instead of LBFGS and without grid extension). However, we again note that KANs are significantly slow to train, as reflected in the report of wall time.
>
> **(iii) MNIST classification (Appendix C)**. We test KANs’ applicability to high-dimensional data.  We find that KANs and MLPs have similar performance (arguably KANs are slightly better but not significantly), again with significantly more training time. We find an interesting phenomenon that the optimal MLP shape (i.e., depth & width) is the same as the optimal KAN shape, which might suggest something universal about MLPs and KANs.
>
> **To address (b), we do a thorough scan of hyperparameters (Appendix D)**, including depths, widths, optimization methods, and learning rates. We find that KANs consistently outperform MLPs across these hyperparameter choices in terms of training losses, however, KANs overfit more significantly than MLPs when KANs are too large (either too deep, or fine-grained) and training data size is small. We also report wall time to show that training wall time is a big limitation of the current KAN implementation.
>
> **To address (c), We have uploaded our codes as a zip file to facilitate reproducibility.**
>
> > Q2: Theoretical assumptions. A main assumption of theorem 2.1 of the paper is that the functions ϕl,i,j are (k+1)-times continuously differentiable. However, the KAT only guarantees that these functions are continuous, as and the authors themselves mention, it has been shown in the literature that these functions can provably be very chaotic, with infinite set of non-differentiability. The authors argue that by using wider/deeper representations, these irregularities may disappear, but this would require further theoretical evidence. I believe there are more reasonable ways of proving universal approximation guarantees for KANs, albeit without the strong scaling laws implied by the k-order spline functions. This main theorem therefore can likely be stated differently, in order to obtain a weaker but more theoretically reasonable statement.
>
> A2: Thanks a lot for the comments. We completely agree. We demonstrated using examples and showing empirically that by using wider/deeper representations, these irregularities may disappear, in figure 2. See also the comment that a deeper KAN is at least as good as shallower ones by setting the subsequent layers to identity. See a version of universal approximation established in a recent preprint by Wang et al. (2024b).
>
> Although proving the advantage of deeper KANs in terms of smoothness can be challenging, we present a concrete example in Figure 2 and Section 2.6. For this function, a three-layer KAN can smoothly represent it, while a two-layer KAN learns highly oscillatory activation functions. And the RMSE loss of the 2-layer KAN is 200x worse than the three-layer KAN.

---

> > ### Author Response · Authors · 2024-11-20
> > **Response to Reviewer whLk Part 2**
> >
> > > Q3: Comparison against symbolic regression benchmarks.
> >
> > A3: Thanks for the question!  As a network-based method, KANs’ strong capability (in fitting even non-symbolic functions) makes it unfavorable for standard symbolic regression benchmarks. For example, KAN ranks second-to-last in GEOBENCH (https://openreview.net/forum?id=TqzNI4v9DT), whereas the last-ranked one EQL is also a network-based model although EQL has been shown to be useful in many scientific applications despite its inability to recover exact symbolic formulas. On the one hand, we would like to explore ways to restrict KANs' hypothesis space so that KANs can achieve good performance on standard symbolic regression benchmarks. On the other hand, we want to point out that KANs have good features that are not reflected by existing benchmarks: (1) interactivity. It is very hard to ``debug'' evolutionary-based symbolic regression methods since the evolution process is long and random. However, it is relatively easier to visualize the training dynamics of KANs, which gives human users intuition on what could go wrong. (2) The ability to “discover” new functions. Since most symbolic regression methods require the input of the symbolic library, they cannot discover things they are not given. For example, if the ground truth formula contains a special function but is not given in the symbolic library, all SR methods will fail definitely. However, KANs are able to discover the need for a new function whose numerical behavior suggests maybe it is a Bessel function; see Figure 23 (d) for an example. We have added this discussion in the main paper, at the end of section 3.
> >
> > > Q4: Spline grid boundary and grid extension. How would you deal with a ‘fat’ tail distribution?
> >
> > A4: We apologize for missing these important details in the main paper! We now add the analysis in Section 2.5 and Appendix L. To handle fat tail distribution, we can determine the grid based on sample quantiles rather than using a uniform grid. In practice, we have a parameter (called grid_eps) that interpolates between the uniform grid and the adaptive grid based on sample quantiles. The user can set the parameter based on how fat tail the data distribution is. However, we acknowledge that this is only a workaround. To deal with extremely flat distributions, splines are probably not fit for them. Instead, Hermite polynomials or similar families are better suited for heavy-tailed distributions.
> >
> > > Q5: Effect of skip connections.
> >
> > A5: Thanks for the nice question! We have included an ablation study in Appendix G. We find that skip connections can make loss landscape better (avoid being stuck at local minima), while the form of skip connections (linear or silu) does not appear to have big differences (for the simple example presented).
> >
> > > Q6: Will continual learning hold true for deeper KANs?
> >
> > A6: Thanks for the great suggestion! We implemented the suggested experiment, using a [1,5,5,1] KAN for the 1D toy function. We find that the continual learning feature is partially or almost completely lost depending on the grid size. New results are included in Appendix Q.

---

> > > ### Comment · Reviewer_whLk · 2024-11-21
> > >
> > > Thank you for providing all the answers and new experiments! I have some follow-up questions/comments:
> > >
> > > - Could you give some more details about this grid update (not grid extension, but rather how it is updated when activations lie outside the [-1,1] interval)? Does this procedure automatically imply increasing the hyperparameter $G$? It is not clear to me how this is carried out in practice. I apologize if this is explained in the text already (feel free to point me to the relevant sections/lines if so).
> > > - Are relative L2 errors for the different methods reported for the Allen-Cahn example? I can't seem to find them in the appendix.
> > > - For the tables reporting results for different benchmarks in the appendix (tables 2, 3 & 4 for example) it would be good to bold the best performing method for easier comparison. Also, on table 3 I feel like its hard to read the values for error and time simultaneously. It might be better to split this data into two separate tables.
> > > - Figure 13 is pretty hard to read with all the information plotted. Instead of plotting the full loss curves, it might be more beneficial to graph the final test error obtained. Additionally, I feel like the function chosen $f(x,y)=\exp(\sin(\pi x)+y^2)$ is probably too smooth/easy to approximate to provide very useful insights on hyperparameters for realistic cases.
> > > - Overall, I think the new experiments provide a much better testbed for evaluating KANs against MLP-based architectures. In fact, in my opinion the experiments presented in the appendix overshadow the ones from the main text. I understand re-writing the main text is a demanding ask, and don't expect the authors to do it before the end of the discussion period, but I encourage them to consider shifting sections to highlight their results on more challenging tasks in the main text.
> > >
> > > Generally speaking, I believe this new version of the paper presents a good improvement over the original, and intend to increase my score of their submission. Having said this, I still have some reservations regarding the organization and clarity of the paper. I understand these are time-consuming efforts, but instead of presenting a very large array of simple benchmarks in the main text, KAN might be better highlighted by focusing on a few of the more challenging tasks, which are currently only reported in the appendix. I also appreciate the authors for highlighting the limitations of KANs and toning down some of the more bold claims for their method. There are still some places in the text where I believe the language can be vague and not well-suited for a scientific paper (eg. "can totally mess up training" - line 1004), but I agree that this has also been improved from the previous version.

---

> > > > ### Author Response · Authors · 2024-11-24
> > > > **Response to Reviewer whLk**
> > > >
> > > > We want to thank the reviewer for their valuable time and kind reconsideration of the updated manuscript. We learned a lot from the suggestions which helped us greatly improve our manuscript. We also appreciate the reviewer’s second round of comments, which we respond below:
> > > >
> > > > > Q1: Provide more details about the grid update
> > > >
> > > > A1: Sure! Please refer to line 241-246 in the updated manuscript.
> > > >
> > > > > A2: Are results reported for the Allen-Cahn example?
> > > >
> > > > Q2: We now report them in Table 3. We did not include them because we thought that Figure 11 (predicted solutions from various methods) already shows qualitative differences. The new Table 3 is more informative about quantitative differences.
> > > >
> > > > > Q3: Boldface the best-performing methods in Tables. And split error and time in Table 3.
> > > >
> > > > A3: Thanks for the nice suggestion! We have implemented them.
> > > >
> > > > > A4: Figure 13 is pretty hard to read. The chosen example is probably too smooth/easy.
> > > >
> > > > Q4: We apologize for the poor readability! We now report the final errors obtained in Tables 6 and 7 and boldface the best numbers. We totally agree this chosen example could be too simple. The point is to set up a pipeline for hyperparameter scans, which can apply to more complicated examples as well. We hope to perform hyperparameter scans for image fitting and PDEs in the future. In particular, the effect of grid size G is particularly interesting to study in the image fitting task, since our preliminary results (in Table 1, comparing KAN 1 and KAN 2) suggest that scaling grid sizes seems more effective than scaling widths.
> > > >
> > > > > A5: The experiments presented in the appendix overshadow the ones from the main text. May consider moving examples in appendices to the main text.
> > > >
> > > > Q5: Thanks for the suggestion! The image-fitting task is an example we want to highlight in the main paper. Due to the page limit, we moved the Anderson localization example (which might be less relevant for the machine learning community) to the appendix. By doing this, we hope that current contents strike a better balance between machine learning applications and scientific applications.

---

> > > > > ### Comment · Reviewer_whLk · 2024-11-24
> > > > >
> > > > > First of all, I want to thank and congratulate the authors for their work during the rebuttal process, which I believe improved their paper a good degree. Based on these changes, I am changing my score from a 3 ("reject") to a 6 ("marginally above the acceptance threshold").
> > > > >
> > > > > Specifically, the factors that influenced the most the change in my score were:
> > > > >  - Additional experiments using more challenging/realistic benchmarks better showcase the potential of KANs. Particularly, the image regression problem highlights their architecture well, and I commend the authors for including this new benchmark in the main text.
> > > > > - The toning down of some of the claims make their paper more grounded and realistic.
> > > > > - Additional explanations and reporting were provided for for the sake of reproducibility, especially the inclusion of code to reproduce some of the main experiments as supplementary material.
> > > > >
> > > > > Having said this, there are still some places where I believe the paper lacks some clarity and/or theoretical and experimental validity, which prevent me from increasing the score to an 8 ("accept"), which is the next available score. In particular:
> > > > > - Depending on the benchmarks, there is not a clear advantage of KANs over MLPs, with both architectures achieving reasonable results, but KANs performing significantly slower (even after the authors improved the performance of KANs for PDE problems). The authors are open about this in the paper, but it still presents a significant limitation. I understand that ICLR reviewing guidelines mention that a paper "does not necessarily require state-of-the-art results", but MLPs are a very old and fundamental building block, and surpassing this baseline is not necessarily surprising. For example,
> > > > >   - In the Allen-Cahn problem, an MLP outperforms KAN by a whole order of magnitude (table 3).
> > > > >   - In the 2D Poisson problem MLPs often outperform KANs, particularly for higher frequency solutions (table 3). Additionally, for this benchmark, L2 errors seem to be clipped at 1e-3, which limits the comparison for high-accuracy solutions (other papers find it possible to reach L2 errors in the order of 1e-5 or 1e-6 for this problem).
> > > > >   - Similarly in the Feynman dataset of simple smooth functions (table 10) and the Darcy flow problem (table 4) KANs and MLPs effectively yield neck-to-neck performance.
> > > > > - The paper currently encompasses a large quantity of experiments, but I believe only a handful of them are meaningful to their claims, as many of the benchmarks/comparisons are too simple to yield useful information for machine learning tasks in science. These weak baselines make the paper more cluttered and harder to parse.
> > > > > - Theorem 2.1, which states the approximation properties of KANs bases itself on the assumption that the inner function from the KAT theorem are $(k+1)$-times continuously differentiable, but the KAT only provides continuity, and subsequent theoretical work from the original theorem shows that these inner functions generally have infinite sets of non-differentiability. While I understand the author's argument that making the representation deeper/wider may mitigate this issue to a degree, I still feel assuming $(k+1)$-differentiability would require further theoretical work. This makes me skeptical of the theoretical $(k+1)$ order of their neural scaling law.
> > > > >
> > > > > Once again, I thank the authors for their time and congratulate them on the improvements to the paper. I hope my review has been helpful, and I am available to continue discussing these points if the authors find it to be helpful to their paper.

---

> > > > > > ### Author Response · Authors · 2024-11-24
> > > > > >
> > > > > > We want to thank the reviewer for their valuable time and kind reconsideration of the updated manuscript. We appreciate that the reviewer valued our efforts in the rebuttal.
> > > > > >
> > > > > > For the specific points further raised, we have the following quick comments in the interest of time.
> > > > > >
> > > > > > 1.  there is not a clear advantage of KANs over MLPs:
> > > > > >
> > > > > > For Allen-Cahn and 2D Poisson, we did not use any tricks of KANs, like grid extension, which greatly enhanced the performance of KANs. The goal was just to show the KANs are capable. (Of course one needs to tune the grid extension parameters to get a good performance) In fact we were also able to show that KANs can out perform MLPs in high-frequency problems with the tricks encoded but it was beyond the scope of this paper.
> > > > > >
> > > > > > 2. These weak baselines make the paper more cluttered and harder to parse.
> > > > > >
> > > > > > Thanks for the comment. We agree that the draft encompasses huge amount of materials, targeted at maybe different audiences in different domains.
> > > > > >
> > > > > > 3. While I understand the author's argument that making the representation deeper/wider may mitigate this issue to a degree, I still feel assuming k+1 differentiability would require further theoretical work.
> > > > > >
> > > > > > Thanks. We agree that this law is not universal for all kinds of problems; as mentioned one can also derive the UAT version theorem for KANs. For the symbolic problems we considered and likewise for a lot of problems in science, this should be valid however.
> > > > > >
> > > > > > Thanks again

---

### Official Review · Reviewer_56QY · 2024-11-02

**Soundness:** 2
**Presentation:** 3
**Contribution:** 3
**Rating:** 6
**Confidence:** 4

**Summary:**

This paper introduces Kolmogorov-Arnold Networks (KANs) as an alternative to Multi-Layer Perceptrons (MLPs), particularly when interpretability is desired, such as for extracting symbolic formulas from datasets in scientific applications. The authors contextualize the Kolmogorov-Arnold representation theorem within modern machine learning, relating it to MLPs and generalizing the representation from two layers to multiple layers via introduced KAN layers, thereby enhancing expressive power while addressing some pathological behavior of Kolmogorov-Arnold representations.

The paper aims to demonstrate the interpretability of KANs and their potential to serve as a useful tool for scientific discoveries, presenting examples from mathematics (knot theory) and physics (Anderson localization) where KANs can assist scientists in (re)discovering mathematical and physical laws. Furthermore, the authors show through theory and experiments that KANs are accurate and have favorable scaling laws, while MLPs scale more slowly and plateau quickly.

The focus of the paper is on relatively small-scale numerical examples from various scientific domains, with the scalability and extensibility of KANs to large-scale machine learning tasks left as future work. Overall, the paper introduces KANs as a promising alternative to MLPs, especially for interpretable and accurate learning in scientific applications, and lays the groundwork for further exploration of their potential.

**Strengths:**

The paper presents a novel neural network architecture inspired by the Kolmogorov-Arnold representation theorem (KST). The authors creatively combine the insights from the KST with modern deep learning techniques, generalizing the original two-layer representation to multiple layers through the introduction of KAN layers. This novel approach to network design demonstrates originality and has the potential to open new avenues for research at the intersection of deep learning and approximation theory.


The authors provide extensive numerical evaluations across a diverse collection of experiments. There they aim to showcase the effectiveness of KANs in various domains, including synthetic toy datasets, special functions, physics equations (Feynman datasets), and partial differential equations. These experiments aim to provide a comprehensive assessment of KANs' performance, interpretability, and scaling laws compared to traditional MLPs.


The paper is well-structured and clearly written, making it accessible to a broad audience. The authors provide sufficient background information on the KST and its relation to neural networks, ensuring that readers can understand the motivation behind KANs. The mathematical formulation of KANs and the approximation theory are presented in a rigorous and understandable manner.


The significance of this work lies in its potential to bridge the gap between deep learning and approximation theory, as well as its implications for interpretable and accurate learning in scientific applications. The authors demonstrate that KANs can be effective in discovering mathematical and physical laws, showcasing some early promise as a tool for scientific discovery. Moreover, the scaling laws exhibited by KANs suggest that they may be a promising alternative to MLPs in certain domains.


I would also like to acknowledge that this research has already generated considerable interest in the recent literature, with several studies exploring the possibilities brought by combining deep learning with the KST. The paper has opened new avenues for research, as the community seeks to assess and enhance the capabilities of KANs and related architectures.

**Weaknesses:**

While I believe that this paper puts forth a creative idea that is worthwhile publishing, the current presentation makes it very difficult to assess whether KANs are indeed a good and competitive method to employ for all the different tasks presented in this work: symbolic regression, function regression, PDE solving, etc. The main reason is that the vast majority of numerical evaluations are poorly designed, either by disadvantaging competing methods, or by comparing against very weak baselines, or by providing no comparison to competing methods (for example, in the symbolic regression benchmarks).

1. **Comparison to state-of-the-art methods:** While the paper presents promising results for KANs in various application settings, such as symbolic regression and solving PDEs, the effectiveness of KANs compared to state-of-the-art methods in each domain remains unclear. The authors primarily compare KANs against basic MLP architectures, which may not represent the best-performing approaches in these specific areas. To provide a more comprehensive assessment of KANs' capabilities, it would be beneficial to compare them against well-established and state-of-the-art methods that are commonly used in practice for each application. This would help to better understand the relative strengths and weaknesses of KANs and their potential to advance the state of the art in these domains.

2. **Overstated claims and conclusions:** Due to the lack of comparisons against state-of-the-art methods in each application area, the current claims and conclusions in the paper may be somewhat overstated. While the results demonstrate the potential of KANs, it is difficult to assess their true effectiveness and superiority without a more comprehensive evaluation against well-established benchmarks and leading approaches. The authors should consider tempering their claims and conclusions to reflect the limitations of their current experimental setup and the need for further validation against state-of-the-art methods.

3. **Interpretability claims and symbolic regression protocol:** The authors emphasize the interpretability of KANs as one of their key strengths. However, the symbolic regression protocol presented in Section E.2 and Figure 10 appears to be largely heuristic and sensitive to various hyperparameter choices. The effectiveness of this protocol in recovering interpretable models from real-world data remains questionable, as the current results focus on rediscovering known scientific formulas. Moreover, the authors do not provide a comparison against state-of-the-art symbolic regression alternatives, making it difficult to assess the relative interpretability of KANs. The arguments against symbolic regression methods in the appendix are not entirely convincing, given the reported success of such methods in challenging settings [1]. To strengthen the interpretability claims, the authors should consider evaluating KANs against leading symbolic regression approaches and demonstrating their effectiveness in recovering interpretable models from real-world data.

4. **Unconventional protocols for training MLPs:** The authors employ unconventional protocols for training MLPs, such as using networks with small widths and full-batch L-BFGS optimization for a limited number of training steps. While these choices may be made to match the parameter complexity and training cost of KANs, they could potentially disadvantage the effectiveness of MLPs and render the comparisons and conclusions questionable. Recent results [3,4] suggest that a fair comparison between KANs and MLPs may lead to different conclusions than those reported in this paper. To address this concern, the authors should consider using more standard and well-established training protocols for MLPs, ensuring a fair and unbiased comparison between the two architectures.

5. **Comparison against plain MLPs in PINNs:** In the context of Physics-Informed Neural Networks (PINNs), the authors compare KANs against plain MLP networks initialized using the Glorot scheme. However, this setup is known to yield a poor baseline in the current PINNs literature, as the derivatives of such networks have limited expressive power to adequately minimize PDE residuals [citation needed]. To provide a more meaningful comparison, the authors should consider evaluating KANs against state-of-the-art PINN frameworks such as PirateNets [5] that have been specifically designed to address the challenges of PDE solving. This would help to better understand the potential advantages of KANs in the context of PINNs and their ability to advance the state of the art in this domain.

[1] Cranmer, M. (2023). Interpretable machine learning for science with PySR and SymbolicRegression. jl. arXiv preprint arXiv:2305.01582.

[2] Cranmer, M., Sanchez Gonzalez, A., Battaglia, P., Xu, R., Cranmer, K., Spergel, D., & Ho, S. (2020). Discovering symbolic models from deep learning with inductive biases. Advances in neural information processing systems, 33, 17429-17442.

[3] Yu, R., Yu, W., & Wang, X. (2024). Kan or mlp: A fairer comparison. arXiv preprint arXiv:2407.16674.

[4] Shukla, K., Toscano, J. D., Wang, Z., Zou, Z., & Karniadakis, G. E. (2024). A comprehensive and FAIR comparison between MLP and KAN representations for differential equations and operator networks. arXiv preprint arXiv:2406.02917.

[5] Wang, S., Li, B., Chen, Y., & Perdikaris, P. (2024). PirateNets: Physics-informed Deep Learning with Residual Adaptive Networks. arXiv preprint arXiv:2402.00326.

**Questions:**

Here are some questions and suggestions for the authors to consider during the rebuttal and discussion phase:

1. **Baselines for symbolic regression:** In the symbolic regression experiments, the authors do not include comparisons to state-of-the-art approaches like PySR (Cranmer, 2023). Such approaches have demonstrated reliable performance on a wide range of symbolic regression tasks and have been successfully applied to real-world scientific discovery problems. To better assess the effectiveness of KANs for symbolic regression, it is important to compare them against the best-performing methods in this domain. Can the authors include experiments comparing KANs to PySR and other leading symbolic regression techniques, and discuss how KANs perform relative to these state-of-the-art baselines in terms of accuracy, efficiency, and interpretability?

2. **Baselines for function regression:** In the function regression experiments, to ensure a fair and meaningful comparison, the authors should consider employing best practices for MLP training, such as using appropriate network sizes, optimization algorithms (e.g., Adam or SGD with learning rate scheduling), and regularization techniques (e.g., dropout or weight decay). Can the authors update their function regression experiments to include MLP baselines trained using best practices and discuss how KANs compare to these more robustly trained MLP models?

3. **Baselines for PDE solving:** In the PDE solving experiments, the authors compare KANs against vanilla MLPs with Glorot initialization, which is known to be a weak baseline in the current literature on Physics-Informed Neural Networks (PINNs). To better demonstrate the effectiveness of KANs in this context, the authors should consider comparing against more advanced and well-established PINN architectures and initialization schemes. Can the authors include experiments comparing KANs to state-of-the-art PINN methods like PirateNets and discuss how KANs perform relative to these stronger baselines?

4. **Continual learning comparison:** The continual learning results presented in the paper are not entirely surprising, given that the spline activations used in KANs are compactly supported, and their local behavior can be modulated by neighboring knots without altering their global structure. For a more fair and informative comparison, the authors should consider evaluating KANs against MLPs with compactly supported activations, such as Radial Basis Functions (RBF), as networks with activations like ReLU or Tanh are inherently prone to catastrophic forgetting. Can the authors provide additional experiments comparing KANs to MLPs with compactly supported activations in the continual learning setting and discuss the implications of these results?

5. **Appendix content and organization:** The Appendix contains a substantial amount of supplementary results that may not significantly contribute to the main messages of the paper and could easily be overlooked by readers. For example, the LAN approach and results presented in Appendix P are interesting but could potentially be the topic of a separate paper (although LANs appear to only introduce a small tweak to the SIREN architecture). This also raises the question of how KANs would perform for an image regression task and why such an evaluation was not provided. In general, to enhance the overall clarity and focus of the manuscript, the authors should carefully evaluate which key results from the Appendix are essential to strengthen the main messages of the paper and consider removing or reorganizing the remaining content. Can the authors provide a rationale for the current organization of the Appendix and discuss how they plan to refine it to better support the core ideas of the paper?

6. **Training details and data normalization:** Some important training details, such as data normalization protocols, are omitted from the text. It appears that KANs require inputs to be in the range [0, 1]. Are the inputs to the MLP networks normalized in the same way? Inconsistencies in data normalization can lead to unfair comparisons and affect the interpretation of the results. Can the authors clarify their data normalization procedures for both KANs and MLPs and discuss any potential impact on the reported findings?

7. **Computational complexity of grid extension and refinement:** The paper introduces the concepts of grid extension and grid refinement for KANs but does not provide a detailed discussion on their computational complexity. Understanding the computational overhead associated with these operations is crucial for assessing the practical applicability of KANs in various settings. Can the authors provide a more in-depth analysis of the computational complexity of grid extension and refinement, including theoretical bounds and empirical measurements, and discuss the implications for the scalability and efficiency of KANs? Moreover, the update of spline grids mentioned in p. 16 seems to be a very important step of the algorithm but it is not adequately explained.

8. **Code availability:**  The authors did not provide any code with this submission and I was unable to assess their implementation. This may have helped with resolving some of the above questions.

**Details Of Ethics Concerns:**

No ethics concerns to report.

---

> ### Author Response · Authors · 2024-11-20
> **Response to Reviewer 56QY Part 1**
>
> We would like to thank the reviewer for their careful reading and constructive suggestions. We have conducted additional experiments to showcase KANs’ ability to perform more complicated tasks and compare them with stronger baselines (Appendix A, B, C), and study hyperparameter dependence of KANs & MLPs (Appendix D). The new experiments use standard training protocols to avoid the possibility of favoring KANs. We have also downplayed our claims and acknowledge the limitations of our method. We respond to each point below.
>
> > Q1: comparison to state-of-the-art methods.
>
> A1: In our added image fitting task (Appendix A), we have used stronger baselines: including SIREN (MLP with sine activation functions), and MLP with random Feature features. We show that KANs can outperform all baselines in terms of PSNR. However, we also want to mention that KANs have 10x - 20x higher training wall time. Our chosen images include “Camera Man” (a natural image), “Turbulence” (from PDEBench), and “The Starry Night” (Van Gohn’s painting), to show KANs’ consistently high performance across different styles. In our new PDE-solving tasks (Appendix B), we also use the MLP with random Fourier features as a stronger baseline, and KANs appear to obtain comparable accuracy with baselines.
>
> > Q2: Overstated claims and conclusions
>
> A2: On the one hand, we added more experiments in Appendix A - D (please refer to the global rebuttal for more details), to stress test KANs’ ability to perform more challenging tasks, as well as conduct fairer comparisons between KANs and baselines. On the other hand, we have downplayed our claims and acknowledged our limitations. We acknowledge the slow training problem in the abstract and the PDE section.
>
> > Q3: interpretability claims and symbolic regression protocol
>
> A3: Thanks for the great question! We have deleted the discussion about symbolic regression in the Appendix but added a fairer discussion at the end of Section 3. As a network-based method, KANs’ strong capability (in fitting even non-symbolic functions) makes it unfavorable for standard symbolic regression benchmarks. For example, KAN ranks second-to-last in GEOBENCH (https://openreview.net/forum?id=TqzNI4v9DT), whereas the last-ranked one EQL is also a network-based model although EQL has been shown to be useful in many scientific applications despite its inability to recover exact symbolic formulas. On the one hand, we would like to explore ways to restrict KANs' hypothesis space so that KANs can achieve good performance on standard symbolic regression benchmarks. On the other hand, we want to point out that KANs have good features that are not reflected by existing benchmarks: (1) interactivity. It is very hard to ``debug'' evolutionary-based symbolic regression methods since the evolution process is long and random. However, it is relatively easier to visualize the training dynamics of KANs, which gives human users intuition on what could go wrong. (2) The ability to “discover” new functions. Since most symbolic regression methods require the input of the symbolic library, they cannot discover things they are not given. For example, if the ground truth formula contains a special function but is not given in the symbolic library, all SR methods will fail definitely. However, KANs are able to discover the need for a new function whose numerical behavior suggests maybe it is a Bessel function; see Figure 23 (d) for an example. In fact, Figure 4 (e) shows an example where we are very close to discovering a new equation, but after our mathematician collaborator dug into the literature for a week, they realized that this is a known fact in the literature. In the future, we hope to use KANs to make truly novel discoveries in science.

---

> ### Author Response · Authors · 2024-11-20
> **Response to Reviewer 56QY Part 2**
>
> > Q4: Unconventional training protocols for training MLPs, e..g, widths are too small, using LBFGS instead of Adam.
>
> A4: We agree that it is a standard protocol to use Adam to train reasonably wide MLPs. In our new experiments (Appendix A-C) for more challenging tasks, we use the Adam optimizer for training and choose MLP to have a width of 128. We find that KANs are comparable to or outperform baselines (for image fitting in Appendix A) in all tasks in terms of accuracy. However, we also show that KANs are significantly slower to train by reporting their wall time. In Appendix D, we also conduct to grid scan of hyperparameters to show the effect of different hyperparameter choices. We observe that KANs consistently achieve lower training losses than MLPs, although KANs might be overfitting when the training data size is small or they become too large (large depth or grid size). We are aware of the papers that conduct fair comparisons between MLPs and KANs. They are not contradictory to our paper, but indeed bring up KANs’ limitations: (1) slow training; (2) non-conventional protocols (LBFGS optimizer, grid update, grid extension) to achieve the best results; (3) applicability to other domains beyond function fitting and PDE solving.
>
> > Q5: Comparison against plain MLPs in PINNs.
>
> A5: We have included stronger baselines (MLPs with random Fourier features) in Appendix B, for solving the Poisson equation, Allen-Cahn equation, and the Darcy equation. KANs are comparable to the stronger baseline in terms of accuracy but have longer training. We understand that solving PDEs with neural networks is a big and well-developed field itself, so we do not expect to achieve SOTA results in this paper. Instead, we have a paragraph at the end of Section 4 to acknowledge the limitations of KANs. Said that, we do feel combining KANs with advanced techniques in PINN will give rise to fruitful directions that will we would like to explore in the future. For example, we have attempted to combine KANs with causal training in the Allen-Cahn example (Appendix B.2), and it will be interesting to see how the gating mechanism in PirateNets can be incorporated into KANs as well.
>
> > Q6: Baselines for symbolic regression.
>
> A6: Please refer to A3.
>
> > Q7: Baselines for function regression
>
> A7: In Appendix A-C, we now included MLPs with random Fourier features as a stronger baseline than MLPs. For image fitting specifically, we include SIREN as an even stronger baseline. In Appendix D, we conduct to grid scan of hyperparameters to show the effect of different hyperparameter choices (including optimization method, learning rate, width, and depth). We observe that KANs consistently achieve lower training losses than MLPs, although KANs might be overfitting when the training data size is small or KANs become too large (large depth or grid size).
>
> > Q8: Baseline for PDE solving.
>
> A8: We included the random Fourier features (inspired by the PirateNet) to obtain a stronger baseline (MLP_RFF, i.e., MLP with random Fourier features). We find that KANs are comparable to MLP_RFF and MLP in the examples we tried in Appendix B, when KANs are trained using conventional protocols. As we wrote in A5, we understand that solving PDEs with neural networks is a big and well-developed field itself, so we do not expect to achieve SOTA results in this paper. Instead, we have a paragraph at the end of Section 4 to acknowledge the limitations of KANs. Said that we do feel combining KANs with advanced techniques in PINN will give rise to fruitful directions that we would like to explore in the future. For example, we have attempted to combine KANs with causal training in the Allen-Cahn example (Appendix B.2), and it will be interesting to see how the gating mechanism in PirateNets can be incorporated into KANs as well.
>
> > Q9: Continual learning.
>
> A9: We want to provide updated results about deeper KANs in this continual learning example. We find that the continual learning feature is partially lost for a [1,5,5,1] KAN, especially when the grid size gets smaller. This is somewhat expected because the locality gets lost when the grid size becomes smaller or when KANs get deeper. We agree that this “continual learning” feature is not specific to KANs, but specific to the locality of activation functions. To add a caveat in Appendix Q “We want to mention that this toy example is somewhat trivial and is attributed to local activation functions rather than the KAN architecture. We simply feel this is a cute example to share in case anyone is inspired by it. However, this should not be interpreted as solving the continual learning problem. Indeed, when we try a deeper KAN [1,5,5,1], the continual learning feature is partially lost, depending on grid sizes.”

---

> > ### Author Response · Authors · 2024-11-20
> > **Response to Reviewer 56QY Part 3**
> >
> > > Q10: Appendix content and organization.
> >
> > A10: We apologize for the messy appendices as they were! We agree that the LAN approach is not that relevant to the current paper and have removed the whole section. We now add new experiments applying KANs to image fitting tasks in Appendix A, where KANs outperform all other baselines despite slow training. We would like to only keep those appendices that are relevant to the main paper, and order them according to where they are referred to in the main paper. We have removed a few appendices (e.g., LANs, visualization of KANs for Feynman problems, and special functions). We are happy to remove more sections if any reviewer deems them unnecessary.
> >
> > > Q11: Training details and data normalization.
> >
> > A11: We apologize for missing the details! KANs can update grids on the fly (as we now clarify at the beginning of Section 2.5) based on input statistics, so in principle, there is no need to normalize input data for KANs. However, to make it fair for MLPs (MLPs do need normalized data), we either obtain samples in bounded regions when we have control of data generation or normalize inputs to zero mean and unit variance.
> >
> > > Q12: Computational complexity of grid extension and refinement.
> >
> > A12: We apologize for missing these important details. We now add the analysis in Section 2.5 and Appendix L. (We use extension and refinement interchangeably; we guess the reviewer means grid update and extension/refinement).
> >
> > > Q13: Code availability
> >
> > A13: We have uploaded our codes as a zip file in the system. Hopefully, this can also help clarify some of the questions above.

---

> > > ### Comment · Reviewer_56QY · 2024-11-24
> > >
> > > Thank you for providing all the answers and new experiments! I have some follow-up comments and suggestions:
> > >
> > > 1. First, I commend your comprehensive response and the significant improvements made to the manuscript. The new experiments, particularly those testing KANs on more challenging tasks and comparing against stronger baselines, have strengthened the paper considerably.
> > >
> > > 2. Make sure you provide sufficient details on the setup of each experiment so others can reproduce your results (e.g., details on training and test data and whether reported PSNR values correspond to train or test locations for the image regression benchmarks).
> > >
> > > 3. While I appreciate the more measured claims in the revised version, some of the well-grounded conclusions currently in the appendix deserve more prominence in the main text.
> > >
> > > 4. The manuscript would benefit from reorganization to better highlight your strongest results, particularly those from the more challenging test cases. Currently, some of your most compelling findings are somewhat buried in the appendix.
> > >
> > > Generally speaking, I believe this new version of the paper presents a good improvement over the original, and I intend to increase my score.

---

> > > > ### Author Response · Authors · 2024-11-24
> > > > **Response to Reviewer 56QY**
> > > >
> > > > We want to thank the reviewer for their valuable time and kind reconsideration of the updated manuscript.
> > > >
> > > > > Q1: Make sure you provide sufficient details on the experimental setups.
> > > >
> > > > A1: We agree that this is important for reproductivity! We make sure that necessary details are included in the paper, and we’ll make codes (currently uploaded as a zip file) available publicly upon acceptance.
> > > >
> > > > > Q2: Some well-grounded conclusions in the appendix deserve more prominence in the main text.
> > > >
> > > > A2: Thanks for the suggestion! We have highlighted KAN’s slow training issue, brought up more details on the training algorithm (grid update/extension), etc. We are open to bringing up more conclusions/discussion from the appendices if reviewers deem it necessary.
> > > >
> > > > > Q3: The manuscript would benefit from reorganization to better highlight more challenging test cases, which are currently buried in the appendix.
> > > >
> > > > A3: Thanks for the suggestion! The image-fitting task is an example we want to highlight in the main paper. Due to the page limit, we moved the Anderson localization example (which might be less relevant for the machine learning community) to the appendix. By doing this, we hope that current contents strike a better balance between machine learning applications and scientific applications.

---

> > > > > ### Comment · Reviewer_56QY · 2024-11-24
> > > > >
> > > > > Thank you again for the clear response. I appreciate all the work you have put in addressing the comments and revising the manuscript which is in much better shape now. I have increased my score accordingly.

---

> ### Public Comment · ~Ali_Kashefi1 · 2024-11-24
> **KAN for complicated goemetries and PDEs**
>
> As a general comment, I would like to give a brief response to the following comment by the reviewer:
>
> **While I believe that this paper puts forth a creative idea that is worthwhile publishing, the current presentation makes it very difficult to assess whether KANs are indeed a good and competitive method to employ for all the different tasks presented in this work: symbolic regression, function regression, PDE solving, etc.**
>
> Please see the following manuscripts and journal papers. They show the applications of KAN for complex geometries and PDEs (e.g., the full Navier-Stokes equations):
>
> **1. A physics-informed deep learning framework for solving forward and inverse problems based on Kolmogorov–Arnold Networks**
>
> --> https://www.sciencedirect.com/science/article/pii/S0045782524007722
>
> **2. A comprehensive and FAIR comparison between MLP and KAN representations for differential equations and operator networks**
>
> --> https://www.sciencedirect.com/science/article/pii/S0045782524005462
>
> **3. Kolmogorov-Arnold PointNet: Deep learning for prediction of fluid fields on irregular geometries**
>
> --> https://arxiv.org/abs/2408.02950
>
> I hope that it helps.

---

### Official Review · Reviewer_5Dsi · 2024-11-03

**Soundness:** 3
**Presentation:** 4
**Contribution:** 4
**Rating:** 8
**Confidence:** 3

**Summary:**

This paper proposes a novel network architecture named KAN based on the Kolmogorov-Arnold representation theorem. Unlike MLPs, the KAN architecture is composed of univariate but learnable activation functions and the summation operation. It extends the Kolmogorov-Arnold representation of a function to deeper layers, which partially addresses the issue of non-smooth functions in the original two-layer Kolmogorov-Arnold representation. KAN shows good interpretability and performance in tasks that deal with symbolic formulae (e.g., applications in scientific discovery).

**Strengths:**

1.	The design of KAN is well motivated and rooted in previous literature, i.e., the Kolmogorov-Arnold representation theorem. Why this theorem was not widely applied for machine learning previously is also discussed.

2.	The presentation of the paper is clear and easy to follow.

3.	The paper covers comprehensive aspects of the proposed architecture, including its theoretical foundation, detailed architectural design, guarantee on approximation ability, implementation details and tricks, etc.

4.	The paper clearly states the “comfort zone” and limitations of KAN, and points out potential future directions.

5.	The view of external and internal degrees of freedom is interesting.

**Weaknesses:**

One major concern is that the selection of architectural hyperparameters of KAN (e.g., the number of layers, the number of nodes in each layer) is not discussed in detail, and it seems that we need some prior knowledge of the target function when selecting these hyperparameters. More detailed concerns and questions are listed below.

**1. About the theoretical guarantee of the approximation ability in Theorem 2.1.** Theorem 2.1 states that if the target function admits a structure of the composition of $L$ KAN layers with smooth activation functions, then we can use an $L$-layer KAN (with B-splines) to approximate it well. However, it does not discuss the scenario where the structure (in particular, the number of layers $L$) of the target function is unknown, which is common in real applications. I wonder if we use a shallower KAN (less than $L$ layers) or a deeper KAN (more than $L$ layers) to approximate the target function (which is assumed to have $L$ layers), how will the result in Theorem 2.1 change? It seems that a deeper KAN can still approximate the target function by setting higher layers as identity mappings, but things become more nuanced for a shallower KAN.

**2. About how to select the number of layers in experiments.** Do we start from training a deep and wide KAN, and then use the sparsification trick in Appendix E to prune redundant nodes or even layers? Or do we simply apply a grid search over different number of layers and then select the best performing combination?  It is encouraged to provide a step-by-step description of the process for determining KAN architectures in the experiments.

There are also some subquestions in this part:

(a)	If we are to first train a deep KAN whose number of layers is probably more than the number of layers of the target function, what will this KAN look like after being trained to convergence? Let us assume the target function $f$ is a composition of $L$ symbolic formulae, and the KAN we use has $M>L$ layers. Would the last $M-L$ layers behave as identity mappings, and the first $L$ layers capture exactly $f$’s structure?

(b)	(As a continuation of question (a)) Although a deeper KAN may have better performance, would the increasing depth degrade its interpretability?

(c)	If we use a KAN that is too deep, would there be difficulties in optimization? If so, how many layers would you recommend starting with?

(d)	Are there advanced and task-dependent strategies (maybe heuristics) to select these hyperparameters, especially for cases where prior knowledge of the target function is limited?

**3. About how to select the number of nodes per layer in experiments.** I find it surprising that in Line 300, a KAN with only one internal node can perform very well in the task of knot theory. That being said, it is unclear how to determine that we only need one node in the intermediate layer. Is it achieved by some prior knowledge of the task or by training a wide KAN and then pruning out redundant nodes?

---

Another concern is about the necessity of the sparsity loss introduced in Appendix E. Is there any ablation study of this sparsity loss? By ablation study, I refer not only to the analysis of performance changes with and without the sparsity loss but also to how the structure of the learned KAN changes. For example, if there is no sparsity loss, is it possible for a layer to learn multiple identical functions “distributed” over different nodes, such as $ x^{2}/3, x^{2}/3, x^{2}/3$, instead of a single $ x^{2}$?

**Questions:**

1.	About the experiment in Figure 5. It is known that standard MLPs (e.g., with ReLU activations) perform poorly in fitting periodic functions such as sine. Is it possible to also try MLPs with other activation functions (e.g., the sine function) to see if it can get better or even comparable performance with KAN?

2.	About Table 1. Does the line labeled by “Theory” refer to the ground-truth formula and the other lines the formulae predicted by KAN? If so, why does the ground-truth formula also have an “accuracy”?

---

> ### Author Response · Authors · 2024-11-20
> **Response to Reviewer 5Dsi Part I**
>
> We would like to thank the reviewer for their careful reading and constructive suggestions. We respond to each point below.
>
> > Q1: 1. About the theoretical guarantee of the approximation ability in Theorem 2.1.
>
> A1: Yes your understanding is absolutely correct. This also motivates partially our choice of making KANs deeper. We have added the discussion in the article.
>
> > Q2: How to select the number of layers in experiments?
>
> A2: Thanks for the question and apologize that we haven’t made this point clear. In the accuracy section (section 4), we employ a hyperparameter scan both for MLPs and for KANs and select the best-performing models. In the interpretability section (section 3), we start with a KAN as small as possible (e.g., we surprisingly find that one hidden layer & one hidden neuron are enough to solve the knot invariant prediction), and then gradually expand the network if performance is bad. We first expand the width. If expanding the width doesn’t help, we expand the depth. Grid size G does not appear to have a large impact as long as it is set to typical values, e.g.,  G = 3,5,10.
>
> > Q2(a)(b): What if the KAN has more layers than needed? Does increasing depth degrate interpretability?
>
> A2(a)(b): Thanks, this is a very great question! We have performed a simple study in Appendix E and Figure 14. The conjecture is correct that the rest layers will (roughly) become an identity map. There are some subtleties though: (1) they might not be exactly identify maps, due to the fact that $f\circ g = (f\circ h)\circ(h^{-1}\circ g)$ for any invertible h. (2) The identity map can appear in early layers or later layers, depending on random seeds. (3) In some cases, a deep network (4-layer in Figure 14) may fail to learn a sparse graph as expected, which we would like to look into in the future.
>
> > Q2(c): If we use a KAN that is too deep, would there be difficulties in optimization? If so, how many layers would you recommend starting with?
>
> A2(c): KAN with 5 layers can optimize just fine on regression problems with no sparsity loss. However, deep KANs can become bad in training in certain cases: (1) when the sparsity penalty is non-zero. (2) When high-order derivatives are involved, as in PDE solving. So we suggest trying KANs with 2-layer (or even 1-layer, if applicable, equivalent to additive models), and gradually increasing the depth if needed. We find that shallow KANs of depth 2 & 3 are already quite powerful in many of our test cases.
>
> > Q2(d): Are there strategies (maybe heuristics) to select hyperparameters?
>
> A2(d): We would like to suggest these heuristics (they should be tested more formally in the future, but they are useful at least in our cases): (1) When one cares about performance. If one doesn’t know what is reasonable shape for MLPs, a hyperparameter scan is unavoidable. However, if one has experience playing with MLPs for the task, using the same shape that the optimal MLP has for KAN is a safe choice (and G~3,5,10) because KANs are more expressive than MLPs given the same shape.  (2) When one cares about interpretability and doesn’t care that much about accuracy (e.g, 85% accuracy is enough), we suggest starting from a KAN as simple as possible (without sparsity penalty), and then gradually increasing width and depth (width first, depth second) until a reasonable performance is achieved. Finally, crank up the sparsify penalty to explore the sparsity vs performance trade-off. Hopefully, the performance is not lost (much) as the network becomes sparser, but this is probably task-dependent.
>
> > Q3: How to select the number of nodes in experiments.
>
> A3: When we care about performance, we usually choose the width that this comparable to or smaller than a typical MLP. When we care about interpretability, we start from a KAN network as simple as possible. The knot invariant example lies in the second category where we start from the minimalist shape (1 hidden layer, 1 hidden neuron) and surprisingly find it to be working already.

---

> > ### Author Response · Authors · 2024-11-20
> > **Response to Reviewer 5Dsi Part 2**
> >
> > > Q4: The necessity of sparsity loss? If there is no sparsity loss, it is possible for a layer to learn multiple identical functions?
> >
> > A4: Thanks for the great question, the hypothesis is correct! We show in the new Appendix F and Figure 15 that a zero or small sparsity penalty (0.001) may lead to multiple redundant functions, which can be avoided by a large penalty strength (0.01).
> >
> > > Q5: Try MLPs with other activation functions and compare with KANs.
> >
> > A5: Thanks for the great suggestion! In the new Appendix A, we have added the task of image fitting, where we compare KANs with SIREN (MLP with sine activations) and MLPs with random Fourier features. We show that KANs can outperform all baselines in terms of PSNR, however with significantly more training wall time.
> >
> > > Q6: What does the ‘theory’ line refer to?
> >
> > A6: We apologize for the confusion! We are referring to Theorem 2.1, where we obtain the scaling law $N^{-4}$ (with k = 3 and m=0). Since the theorem does not provide a way to estimate the constant factor C, only the slope of the theory line (i.e., -4 in the log-log plot) is meaningful.

---

> > ### Comment · Reviewer_5Dsi · 2024-11-24
> >
> > I would like to thank the authors for their detailed responses. Most of my concerns are addressed, and I find the revised manuscript better by incorporating other reviewers' suggestions on new experiments, limitations, etc. I will keep my score towards acceptance.

---

### Official Review · Reviewer_cGWE · 2024-11-03

**Soundness:** 2
**Presentation:** 3
**Contribution:** 3
**Rating:** 8
**Confidence:** 4

**Summary:**

The paper proposes a replacement for multilayer perceptrons based on the Arnold-Kolmogorov theorem which gives a two layer network for function approximation as a sum of univariate functions of individual features followed by a function of the sum. The Arnold-Kolmogorov theorem has previously been declared irrelevant for building learning machines since the involved functions can be nonsmooth [cite].
The paper reconsiders the Arnold-Kolmogorov theorem for building learning models and proposes that deep versions of networks using the Arnold-Kolmogorov theorem might not be limited by the non-smoothness that appears in shallow networks. The paper proposes Kolmogorov Arnold networks which are stacked layers of the univariate function+sum layers that appear in the 2-layer network.
The paper further gives an approximation theorem which bounds the error in the derivatives up to some order for a function approximated by a KAN network with B-splines.
The advantages of KANs over MLPs are said to be that KANs can be more interpretable than MLPs which makes them useful for symbolic regression tasks and are useful in data-driven scientific discovery.

**Strengths:**

The main contribution of the paper is a practical instantiation of the Arnold-Kolmogorov theorem as a deep network which can avoid the difficulties plaguing shallow versions of the theorem. There are a number of experiments validating the method ranging from replicating discovery results from knot theory, automatic and interactive symbolic regression and solution of partial differential equations. The paper is generally well-written and clear.

**Weaknesses:**

I have a few concerns, listed in the following.

1. In light of previous concerns in the literature with the Kolmogorov-Arnold theorem as a basis for learning machines, I think the paper should focus more on establishing whether the problem of non-smoothness is in fact avoided in deeper versions. Since this would establish whether the proposed model is general enough and can scale to high dimensions.

2. The paper presents Theorem 2.1 which gives a bound on the supremum of the error in the derivative. However, it is not clear whether the bound is actually useful since it depends on f and the network in an unknown way through the constant C.
If a general theoretical argument demonstrating smoothness is too difficult, perhaps the paper could investigate the smoothness of the learned functions (or the gradients during training) to provide empirical evidence that the model learns smooth functions.

3. Regarding the proof of Theorem 2.1 in Appendix B. It is not clear to me how the bound on the residual is obtained in line 894.
If I understand correctly, the bound on B-spline approximation in line 887 is for a single univariate function, phi_{l,i,j}, whose inputs are exact. However when the inputs from the layer below are also approximated and then summed, one would expect the bound of sup of error in the derivatives to also be summed? Whereas the error in line 887 for exact inputs and that in line 894 for approximate inputs is exactly the same. Could you clarify how the error for the residual is obtained?

4. I don’t quite understand how in the symbolic regression tasks the units are automatically converted into symbolic functions. I also couldn’t find the metric used for accuracy in the SR experiments.

5. I think the motivation described in lines 62-64 should be rephrased. Regardless of the aim of interpretability, smoothness is still quite essential for learning which is the point of Giros & Poggio [1989].

Minor corrections.

Line 16. Interpretabe -> interpretable

205. KAT -> KAN

473. pareton -> pareto

**Questions:**

Are the constants C in the proof in appendix B and the statement of Theorem 2.1 the same? If not, different notation would be better.

Did you compare the symbolic regression results with other methods? Some baselines would be useful.

As asked above, could you please clarify how the residual error bound in the proof of Theorem 2.1 is obtained?

---

> ### Author Response · Authors · 2024-11-20
> **Response to Review cGWE part 1**
>
> We would like to thank the reviewer for their careful reading and constructive suggestions. We respond to each point below.
>
> > Q1: The paper should focus on establishing whether the problem of non-smoothness is in fact avoided in deeper versions.
>
> A1: Thanks for the great suggestion! We have added a new Section 2.6 to expand on this point. We admit that providing theoretical proof can be challenging, so we instead use a concrete example to illustrate this point. We cooked up a four-dimensional smooth function. As shown in Figure 2, a three-layer KAN is able to represent this function smoothly, while a two-layer KAN learns highly oscillatory representations and has 200x higher RMSE than the three-layer KAN.
>
> > Q2: The paper presents Theorem 2.1 which gives a bound on the supremum of the error in the derivative. However, it is not clear whether the bound is actually useful since it depends on f and the network in an unknown way through the constant C. If a general theoretical argument demonstrating smoothness is too difficult, perhaps the paper could investigate the smoothness of the learned functions (or the gradients during training) to provide empirical evidence that the model learns smooth functions.
>
> A2: Thanks for the question. It is indeed a valid concern to verify the smoothness in practice. Thm 2.1 is stated in terms of best approximation, not necessarily implying what will happen in practice during the training dynamics. In the experiments in section 4, the loss curves indeed match the theoretical scaling law, thus providing empirical evidence that the model learns smooth functions.
>
> > Q3: Regarding the proof of Theorem 2.1 in Appendix B. It is not clear to me how the bound on the residual is obtained in line 894. If I understand correctly, the bound on B-spline approximation in line 887 is for a single univariate function, phi_{l,i,j}, whose inputs are exact. However when the inputs from the layer below are also approximated and then summed, one would expect the bound of sup of error in the derivatives to also be summed? Whereas the error in line 887 for exact inputs and that in line 894 for approximate inputs is exactly the same. Could you clarify how the error for the residual is obtained?
>
> A3: Thanks for the question. Your understanding is indeed correct. The constant C was indeed summed which varies from line to line, rendering the constant C to depend on the architecture and number of compositions. C is however independent of G, which highlights how we avoid the curse of dimensionality.
>
> > Q4: How are units automatically converted into symbolic functions? What are the metrics used for symbolic regression?
>
> A4: We apologize that we missed this detail in the main paper. We have added this sentence in Section 2.4 “The idea is to match learned spline functions with candidates in a symbolic function library specified by human users and replace the spline functions with the best-fitting ones.” Since both examples (knot theory and adnerson localization) are classification tasks, we used test accuracy as the metric to evaluate discovered formulas.
>
> > Q5: The motivation that “we care intrepretability rather than smoothness” should be rephrased.
>
> A5: Totally agree! We have deleted the sentence, but instead discussed the concrete example we have in A1 and the new Section 2.6.
>
> > Q6: Typos.
>
> A6: Thanks for reading so carefully! We have corrected the typos.
>
> > Q7: Are the constants C in the proof in appendix B and the statement of Theorem 2.1 the same? If not, different notation would be better.
>
> A7: See answer to Q3. We have adopted a different notation thanks to the suggestion.

---

> > ### Author Response · Authors · 2024-11-20
> > **Response to Review cGWE part 2**
> >
> > > Q8: Compare with other symbolic regression results
> >
> > A8: Thanks for the question!  As a network-based method, KANs’ strong capability (in fitting even non-symbolic functions) makes it unfavorable for standard symbolic regression benchmarks. For example, KAN ranks second-to-last in GEOBENCH (https://openreview.net/forum?id=TqzNI4v9DT), whereas the last-ranked one EQL is also a network-based model although EQL has been shown to be useful in many scientific applications despite its inability to recover exact symbolic formulas. On the one hand, we would like to explore ways to restrict KANs' hypothesis space so that KANs can achieve good performance on standard symbolic regression benchmarks. On the other hand, we want to point out that KANs have good features that are not reflected by existing benchmarks: (1) interactivity. It is very hard to ``debug'' evolutionary-based symbolic regression methods since the evolution process is long and random. However, it is relatively easier to visualize the training dynamics of KANs, which gives human users intuition on what could go wrong. (2) The ability to “discover” new functions. Since most symbolic regression methods require the input of the symbolic library, they cannot discover things they are not given. For example, if the ground truth formula contains a special function but is not given in the symbolic library, all SR methods will fail definitely. However, KANs are able to discover the need for a new function whose numerical behavior suggests maybe it is a Bessel function; see Figure 23 (d) for an example.
> >
> > > Q9: As asked above, could you please clarify how the residual error bound in the proof of Theorem 2.1 is obtained?
> >
> > A9: Thanks for the question. Due to the limitation of the presentation, it was not elaborated. As in the proof, we first obtain a bound on the univariate approximation with a constant (the constant should be understood as taking a supremum over all $\phi_{i,j,l}$). Then we use Lipschitz continuity to propagate on the bound on $R_l$, with another constant. Finally, notice that we can represent the error by a sum of $R_l$ and come up with our final constant $C$.

---

> > ### Comment · Reviewer_cGWE · 2024-11-25
> >
> > Thank you for the response. I agree the paper is improved.
> >
> > I do have a lingering concern regard theorem 2.1. The rebuttal says that the constant C is independent of G which is what allows avoiding the curse of dimensionality. The “curse of dimensionality”, however, only makes sense when considering a class of functions, say with varying n, whereas Theorem 2.1 is only valid for a single function. If it turns out that C is a quickly changing function, say exponential in n, then G would have to become very fine to maintain the same error.  Since the current version of the theorem doesn’t have a notion of family or class of functions, and does not consider the interplay of C and G required for reasonable error, I don’t think it is justified to present this as avoiding the curse of dimensionality.

---

> > > ### Author Response · Authors · 2024-11-26
> > >
> > > Thanks for the comment. We have now removed the "curse of dimensionality" claim and only maintained that it has better scaling laws. We have also specified the function class and made more precise the dependence of the constant C in appendix J.

---

> > > > ### Comment · Reviewer_cGWE · 2024-11-27
> > > >
> > > > Thank you for addressing the concerns. I will raise my score.

---

### Author Response · Authors · 2024-11-20
**Global Rebuttal**

We would like to thank all the reviewers for their valuable time and constructive suggestions. We have implemented most of them, leading to the new appendices A - H containing additional experimental results. We have rewritten the manuscript extensively to clarify statements, tone down our claims, and acknowledge the limitations, with major changes highlighted in blue. We also upload our codes as a zip file to facilitate reproducibility.

# Summary of Changes:

**1. Additional new experiments.** Reviewers share three major concerns about the experimental part of the paper: (a) The selected examples are very simple. It is advised by reviewers that more complex tasks should be included to stress test KANs; (b) The comparison between KANs and MLPs seems unfair. Reviewers suggest including stronger baselines and doing a more thorough hyperparameter scanning. (c) Ablation studies of KANs.

**To address (a), we conducted three families of experiments:**

**(i) Image Fitting (Appendix A)**, given that images usually contain sharp local features or high-frequency features. We show that KANs can outperform all baselines (including more advanced baselines, i.e., SIREN, and MLP with random Fourier features) in terms of PSNR. However, we also want to mention that KANs have 10x - 20x higher training wall time. Our chosen images include “Camera Man” (a natural image), “Turbulence” (from PDEBench), and “The Starry Night” (Van Gohn’s painting), to show KANs’ consistently high performance across different styles.

**(ii) PDE solving (Appendix B)**. In B.1, we use the Poisson equation with high-frequency solutions to test KANs’ ability to learn high-frequency modes. In B.2, we use the Allen-Cahn equation to test KAN’s ability to learn temporal PDEs and sharp boundaries. In B.3, we use Darcy flow to test KAN’s ability to learn unstructured solutions. We find that KANs can achieve comparable accuracy with MLPs and MLPs with random Fourier features (under “fair” comparison, i.e., using the Adam optimizer instead of LBFGS and without grid extension). However, we again note that KANs are significantly slow to train, as reflected in the report of wall time.

**(iii) MNIST classification (Appendix C)**. We test KANs’ applicability to high-dimensional data.  We find that KANs and MLPs have similar performance (arguably KANs are slightly better but not significantly), again with significantly more training time. We find an interesting phenomenon that the optimal MLP shape (i.e., depth & width) is the same as the optimal KAN shape, which might suggest something universal about MLPs and KANs.

**To address (b), we do a thorough scan of hyperparameters (Appendix D)**, including depths, widths, optimization methods, and learning rates. We find that KANs consistently outperform MLPs across these hyperparameter choices in terms of training losses, however, KANs overfit more significantly than MLPs when KANs are too large (either too deep or fine-grained) and training data size is small.

**To address (c), we add ablation studies in Appendix E - H.** We study the role of depth in Appendix E, the role of sparsity strength in Appendix F, the role of skip connections in Appendix G, and the role of grid extension in Appendix H.

**2. Texts have been changed to better (a) clarify motivations, (b) add in necessary details, and (c) reduce overclaims and acknowledge limitations. Major changes are highlighted in blue.**

(a) As suggested by Reviewer cGWE, we now add a new Section 2.6 to discuss how depth helps the non-smoothness problem. We use a concrete example to show that 2L-KAN learns highly oscillatory representations while 3L-KAN learns smooth representations.

(b) We added more details for the complexity of our method, and sketched the main idea of grid extension in the main paper, etc.

(c) We acknowledge our limitations. We acknowledge the slow training problem in the abstract and the PDE section. We acknowledge KANs’ inability to do well on standard symbolic regression benchmarks at the end of Section 3. We also downplayed our novelty claim in conclusions by acknowledging the possibility that MLPs and KANs might be equivalent in some aspects but different in others, which will need more formal study in the future.

---

### Author Response · Authors · 2024-11-24
**Global Rebuttal (2nd round)**

We would like to thank all reviewers again for their constructive suggestions which helped us greatly improve our manuscript. In response to reviewers’ new comments, we made some new changes and would like to summarize them here:

 * Reviewers suggest highlighting more challenging test cases currently buried in the appendix. We now move the image fitting task to the main paper, while (due to page limit) moving the Anderson Localization example to the appendix. By doing this, we hope that the current organization strikes a better balance between machine learning applications and scientific applications.
 * Reduced training time for PDE examples. We realized that a small trick (disabling the symbolic branch, by calling ``model.speed()``) can largely reduce training time in PDE cases. We have rerun these PDE examples and reported the new training time, which is typically 10 times faster than the previous results. This makes KAN, as a PDE solver, have comparable (or still slightly worse) efficiency with MLPs, but at least it is no longer the case “KANs are significantly slower than MLPs”. This is an important trick to optimize KAN’s efficiency, which we feel is important enough to highlight here for general readers to take notice.

---

### Public Comment · ~Yanbing_Mao1 · 2024-11-29
**Ignored Existing Work?**

Dear Authors, Reviewers, and Public:

I want to share three references below. At a high level, the three papers and the KAN work have some things in common. For example:
1)  Directly embed validated nonlinear features into NN rather than approximation by act(Wx + b) [1-3].
2) Link editing to remove/avoid spurious links [2].
3) The activation function can be removed and, of course, preserved [1,2].

I put the comment here because i) I noticed many submissions related to KAN, but I believe this submission is the original one of KAN, and ii) I am disappointed to see only comparisons with MLP but ignore the existing work [1-3]. Compared with MLP, the frameworks in [1-3] are the most related ones that KAN should compare (just my opinion). It would be great if readers could spend 10 seconds reading [1-3] to verify my opinion. Hopefully, my opinion is wrong.

**Referneces**

[1] Chrysos, G. G., Moschoglou, S., Bouritsas, G., Deng, J., Panagakis, Y., & Zafeiriou, S. (2021). Deep polynomial neural networks. IEEE transactions on pattern analysis and machine intelligence, 44(8), 4021-4034.

[2] Mao, Y., Gu, Y., Sha, L., Shao, H., Wang, Q., \& Abdelzaher, T. (2023). Phy-Taylor: Partially Physics-Knowledge-Enhanced Deep Neural Networks via NN Editing. IEEE Transactions on Neural Networks and Learning Systems

[3] Brunton, S. L., Proctor, J. L., \& Kutz, J. N. (2016). Discovering governing equations from data by sparse identification of nonlinear dynamical systems. Proceedings of the national academy of sciences, 113(15), 3932-3937.

---

### Meta-Review · Area_Chair_ppio · 2024-12-20

**Metareview:**

This paper proposes Kolmogorov-Arnold networks (KANs) as a more interpretable alternative to multi-layer perceptrons (MLPs), particularly for scientific applications. The authors have made significant improvements to the paper in response to reviewer feedback, including adding new experiments, toning down strong claims, and providing a more detailed analysis of computational complexity.

The reviewers have raised several points, including the need for more empirical evidence to support the claims, the similarity between KANs and MLPs, and the lack of evaluation on larger, more complex datasets. The authors have addressed these concerns by adding new experiments, including comparisons to MLPs on MNIST and image-fitting tasks, and providing a more detailed analysis of the computational complexity of KANs.

While the reviewers have noted that KANs are slower to train than MLPs and may not have a clear advantage over MLPs on all tasks, they have also acknowledged the potential benefits of KANs, including their interpretability and ability to discover new functions. The authors have been responsive to feedback and have made significant improvements to the paper.

Hence, after discussion with the authors and among themselves, the reviewers find the paper much improved, with all reviewers leaning towards acceptance. We are therefore happy to accept the paper. We would still like to encourage the authors to address all the reviewers' comments in the camera-ready version.

**Additional Comments On Reviewer Discussion:**

see above

---

### Decision · Program_Chairs · 2025-01-22

Accept (Oral)